EMBO
Molecular Medicine

# WNT inhibition creates a BRCA-like state in Wnt-addicted cancer

Amanpreet Kaur[1], Jun Yi Stanley Lim[1], Sugunavathi Sepramaniam[2], Siddhi Patnaik[1], Nathan Harmston[1,3], May Ann Lee[2], Enrico Petretto[4], David M Virshup[1,5] & Babita Madan[1,*]

## Abstract

Wnt signaling maintains diverse adult stem cell compartments and is implicated in chemotherapy resistance in cancer. PORCN inhibitors that block Wnt secretion have proven effective in Wnt-addicted preclinical cancer models and are in clinical trials. In a survey for potential combination therapies, we found that Wnt inhibition synergizes with the PARP inhibitor olaparib in Wnt-addicted cancers. Mechanistically, we find that multiple genes in the homologous recombination and Fanconi anemia repair pathways, including *BRCA1*, *FANCD2*, and *RAD51*, are dependent on Wnt/β-catenin signaling in Wnt-high cancers, and treatment with a PORCN inhibitor creates a BRCA-like state. This coherent regulation of DNA repair genes occurs in part via a Wnt/β-catenin/MYBL2 axis. Importantly, this pathway also functions in intestinal crypts, where high expression of BRCA and Fanconi anemia genes is seen in intestinal stem cells, with further upregulation in Wnt-high APC[min] mutant polyps. Our findings suggest a general paradigm that Wnt/β-catenin signaling enhances DNA repair in stem cells and cancers to maintain genomic integrity. Conversely, interventions that block Wnt signaling may sensitize cancers to radiation and other DNA damaging agents.

**Keywords** BRCA1; DNA repair; ETC-1922159; FANCD2; homologous recombination

**Subject Categories** Cancer; Stem Cells & Regenerative Medicine

See also: **S Angers** (April 2021)

## Introduction

Stem cells in normal tissues protect the integrity of their genome by expression of diverse proteins that accurately detect and repair mutations introduced by DNA replication and environmental mutagens (Hua *et al*, 2012). Inherited defects in DNA repair pathways, such as germline mutations in BRCA1 or Fanconi anemia pathway genes that normally repair double-strand breaks and DNA cross-links, lead to the accumulation of mutations that impair stem cell function and promote tumorigenesis (Dietlein *et al*, 2014; Ma *et al*, 2018). The FA pathway proteins serve as interstrand cross-link (ICL)-sensors and promote DNA repair in conjunction with homologous recombination and other DNA repair pathways including BRCA proteins (Deans & West, 2011; Ceccaldi *et al*, 2016). Individuals with Fanconi anemia who have mutations in Fanconi anemia (FA) pathway genes are exquisitely sensitive to DNA ICL-generating agents. These DNA repair pathways are often co-opted in cancers during the development of drug resistance (Jackson & Bartek, 2009).

Defects in specific DNA repair pathways also present an opportunity for targeted anti-cancer therapies (Dietlein *et al*, 2014; Ma *et al*, 2018; Ashworth & Lord, 2018). Breast and ovarian cancers with defects in the homologous recombination (HR) repair pathway, especially BRCA1 and BRCA2 mutant tumors, are sensitive to poly ADP-ribose polymerase (PARP) inhibitors such as olaparib (Farmer *et al*, 2005; Bryant *et al*, 2005; Ashworth & Lord, 2018). These inhibitors work by blocking the function of PARP proteins essential for single-strand break (SSB) repair. Upon PARP inhibitor treatment, the unrepaired SSBs can progress to double-strand breaks that are then repaired by homologous recombination. In HR-deficient cells, treatment with PARP inhibitors results in growth arrest by apoptosis or senescence due to the accumulation of DNA damage (Ashworth & Lord, 2018).

Wnt signaling is important for the maintenance of stem cell state, adult tissue homeostasis, and the prevention of differentiation (Clevers *et al*, 2014). Wnt signaling is also associated with the development of radioresistance in many cancers, but the underlying mechanisms are not well understood (Metcalfe *et al*, 2014; Jun *et al*, 2016; Emons *et al*, 2017; Zhao *et al*, 2018; Luo *et al*, 2019; Karimaian *et al*, 2017). Wnts are a family of 19 secreted palmitoleated glycoproteins that signal by binding to Frizzled (FZD) and additional co-receptors on the cell surface. The interaction of Wnts with their receptors stabilizes β-catenin, which then translocates to the nucleus to regulate gene expression. Aberrant stabilization of β-catenin can be caused by mutation of components of the Wnt

1 Program in Cancer and Stem Cell Biology, Duke-NUS Medical School, Singapore, Singapore
2 Experimental Drug Development Centre, A*Star, Singapore, Singapore
3 Science Division, Yale-NUS College, Singapore, Singapore
4 Center for Computational Biology and Program in Cardiovascular and Metabolic Disorders, Duke-NUS Medical School, Singapore, Singapore
5 Department of Pediatrics, Duke University School of Medicine, Durham, NC, USA
 *Corresponding author. Tel: +65 161790; E-mail: babita.madan@duke-nus.edu.sg

pathway such as APC, AXIN and CTNNB1 (β-catenin), as is frequently seen in colorectal, gastric and liver cancers (Nusse & Clevers, 2017). Additionally, a subset of pancreatic, mucinous ovarian, colorectal, gastric, adrenocortical, and endometrial cancers harbor loss-of-function mutations in E3-ubiquitin ligases RNF43 or its paralog ZNRF3 or gene fusions leading to activation of Wnt agonists RSPO2/3 (R-spondin 2/3) (Wu *et al*, 2011; Bailey *et al*, 2016; Ryland *et al*, 2013; Wang *et al*, 2014; Giannakis *et al*, 2014; Assié *et al*, 2014; Seshagiri *et al*, 2012). These mutations enhance the abundance of Frizzleds and cause cancers to be dependent on activated Wnt signaling. This subset of cancers is highly sensitive to upstream inhibitors of Wnt signaling pathway such as anti-FZD and anti-R-spondin antibodies as well as PORCN inhibitors such as ETC-159 (Gurney *et al*, 2012; Jiang *et al*, 2013; Madan *et al*, 2016). Several of these agents are currently in clinical trials. These inhibitors are also powerful tools to investigate the pathways that are regulated by Wnt signaling in cancer (Madan *et al*, 2016, 2018).

During a screen to identify approved drugs that synergize with PORCN inhibitors, we made the unexpected observation that the PARP inhibitor olaparib synergized with ETC-159 in Wnt-addicted cancers. Mechanistically, we found that inhibition of Wnt signaling in multiple cancer cell lines and normal intestinal crypts results in the suppression of multiple HR pathway genes. Wnt signaling through a β-catenin/MYBL2 pathway regulates the expression of *BRCA1*, *BRCA2*, *RAD51*, and *FANCD2*. This study uncovers a role for Wnt/β-catenin signaling in the regulation of homologous recombination DNA repair pathway in intestinal stem cells and in cancer and demonstrates that inhibition of Wnt signaling confers a BRCA-like phenotype, providing a novel therapeutic opportunity for Wnt high cancers.

# Results

## Wnt inhibition synergizes with PARP inhibitor Olaparib

The PORCN inhibitor ETC-159 has shown efficacy as a monotherapy in preclinical models of Wnt-addicted cancers (Madan *et al*, 2016). To identify potential combinatorial therapeutic options, we performed a synergy screen using the Chou-Talalay method with selected drugs that are either FDA approved or in clinical trials (Chou, 2010). Since Wnt-addicted cells are substantially more sensitive to PORCN inhibitors when grown in suspension, the screen was performed in the Wnt-addicted RNF43-mutant pancreatic cancer cell line HPAF-II using soft agar colony formation as the readout (Madan *et al*, 2016; Zhong *et al*, 2019). We observed that the combination of olaparib and ETC-159 was significantly more effective in inhibiting colony formation of HPAF-II cells than treatment with either drug individually (Fig 1A and Table EV1). Consistent with this, the drug combination index values for ETC-159 and olaparib, as determined using the Chou-Talalay CompuSyn algorithm, showed a synergistic effect (Combination Index < 1) for each of the dosages tested (Fig 1B). To test whether this combination was also efficacious *in vivo*, we used the HPAF-II pancreatic cancer xenograft model. Treatment with olaparib alone or low dose of ETC-159 alone was less effective compared to the combination of olaparib and ETC-159 in preventing HPAF-II tumor growth in mice. This was shown by changes in tumor volumes during the course of treatment (Fig 1C) as well as the tumor weights at the end of 21 days treatment (Fig 1D).

Next, we assessed the combination of PORCN inhibitor and olaparib in three additional Wnt-addicted cell lines from diverse cancer types with distinct Wnt pathway mutations. The cholangiocarcinoma cell line EGI-1 (Wnt-addicted due to an R-spondin translocation), the ovarian cancer cell line MCAS (Wnt-addicted due to an *RNF43* mutation), and the pancreatic cancer cell line CFPAC-1 (sensitive to Wnt inhibition, mechanism unknown) were used. Similar to what was observed with HPAF-II cells, the combination of ETC-159 and olaparib synergistically inhibited colony formation in all three cell lines in soft agar assay at all the doses tested (Figs 1E and F, and EV1A and B and Table EV1). Thus, the synergy of ETC-159 and olaparib is a general phenomenon. Taken together, the data indicate that blocking Wnt activity with a PORCN inhibitor sensitizes Wnt-addicted cells to a PARP inhibitor.

## Wnt inhibition reduces expression of homologous recombination (HR) and Fanconi anemia (FA) repair pathway genes

Olaparib and related PARP inhibitors are uniquely effective in BRCA-mutant and BRCA-like cancers that have dysfunctional homologous recombination (Armstrong & Clay, 2019). Diverse mechanisms can cause defective BRCA-like behavior, including inherited mutations in *BRCA1*, *BRCA2*, and Fanconi anemia complementation (*FANC*) group of genes (Lord & Ashworth, 2016). Epigenetic and transcriptional mechanisms that silence HR pathway genes can also cause a BRCA-like state (Ibrahim *et al*, 2012). To test whether Wnt inhibition induces a BRCA-like state, we examined the expression of genes involved in DNA repair using our transcriptome dataset from ETC-159-treated Wnt-addicted HPAF-II pancreatic cancer orthotopic xenografts (Fig EV1C) (Madan *et al*, 2018). Remarkably, among the genes downregulated following PORCN inhibition, i.e. *Wnt-activated* genes, there were three clusters of genes (C1, C5, and C12) that were significantly enriched for Gene Ontology (GO) annotated processes and pathways related to multiple components of the DNA damage repair pathway (Figs 2A and B and EV1C).

Further analysis of the genes involved in regulating each of the DNA repair pathways highlighted that Wnt inhibition robustly downregulated genes involved in homologous recombination (HR) and Fanconi anemia (FA) pathways (Fig 2C). By comparison, genes annotated to be involved in mismatch repair and in the repair of single strand breaks such as base excision repair (BER) or nucleotide excision repair (NER) were much less affected by changes in Wnt signaling. Most notably, the expression of *BRCA1*, *BRCA2*, *RAD51*, and multiple *FANC* genes was decreased by 70–80% compared to the vehicle group by 32 h after the start of treatment with the PORCN inhibitor (Fig 2C and D). Immunoblotting for BRCA1, BRCA2, FANCA, and FANCD2 in HPAF-II tumors harvested from ETC-159 treated mice confirmed the downregulation of these proteins by 56 h (Figs 2E and EV1D). This downregulation of genes involved in the HR and FA pathways following Wnt inhibition is consistent with the observed increase in sensitivity of Wnt-addicted tumors and cell lines to olaparib (Fig 1).

While ETC-159 treatment of HPAF-II cells *in vitro* for 48 h significantly reduced the expression of HR pathway and FA genes, we did not observe a significant change in the percentage of cells in S-phase as compared to control (Fig EV1E and F). These results suggest that the suppression of HR and FA pathway gene expression by ETC-159

treatment is not due to its effect on cell proliferation (Saleh-Gohari, 2004; Davidson & Niehrs, 2010).

To test whether PORCN inhibition causes a decrease in HR and FA genes in additional Wnt-addicted cancer models, we examined RNA-seq data from a colorectal cancer patient-derived xenograft (CRC PDX) with a *RSPO3* translocation that was treated with ETC-159 (Madan *et al*, 2016). In this CRC PDX model, similar to the HPAF-II xenografts, genes that were downregulated (FDR < 0.1, fold change > 1.5) also had significant enrichment for multiple processes associated with DNA damage repair (Fig 2B). Specifically, several key HR and FA pathway genes were downregulated upon Wnt inhibition (Fig 2F; Madan *et al*, 2016, 2018). We confirmed the marked

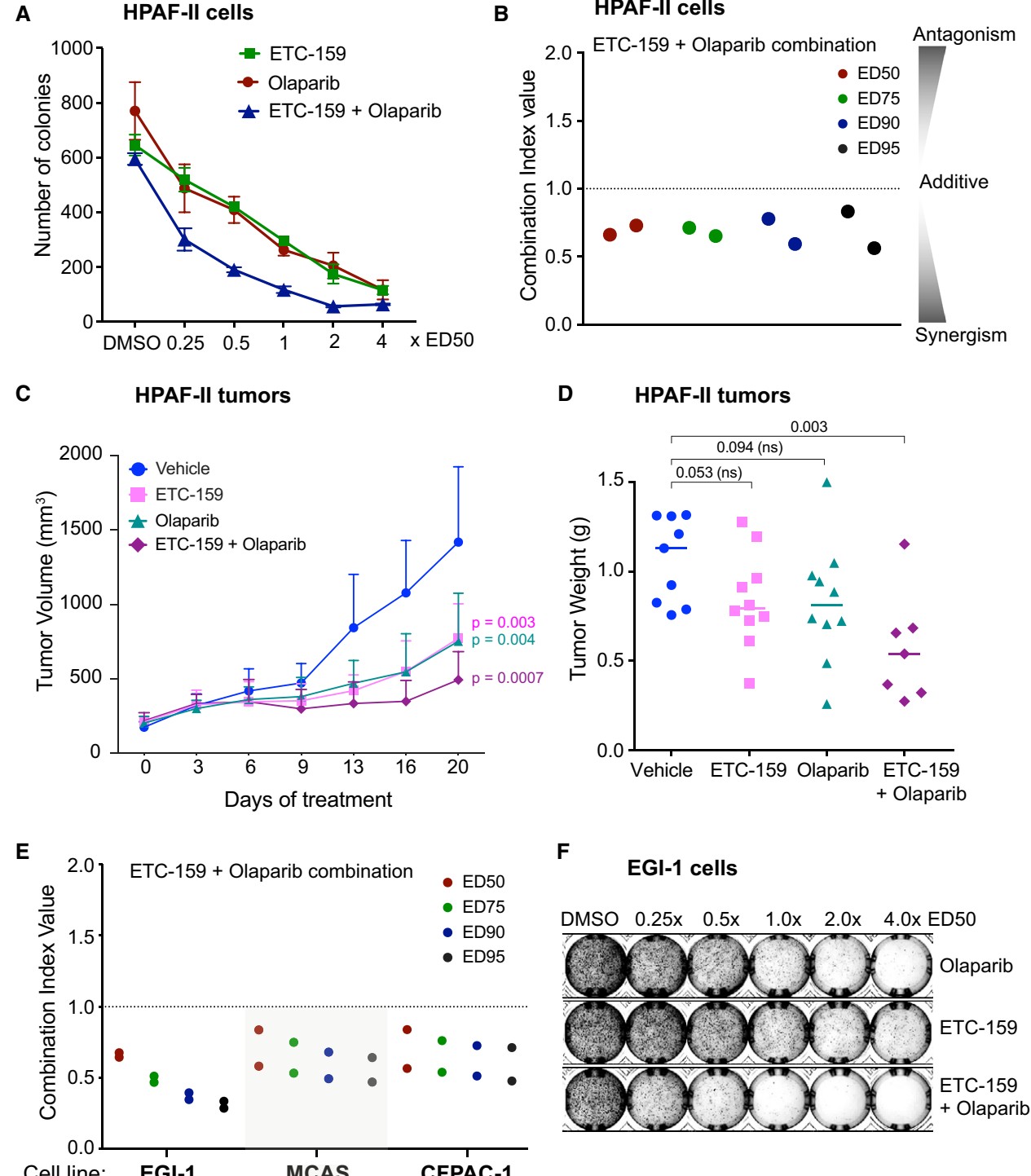

**Figure 1.**

◀

**Figure 1. ETC-159 synergizes with the PARP inhibitor Olaparib.**

A, B   PORCN inhibitor ETC-159 synergizes with PARP inhibitor olaparib in suppressing proliferation of HPAF-II pancreatic cancer cells in soft agar. The $ED_{50}$ (dose that reduced colony formation by 50% of the maximal inhibition) was determined (Table EV1) and cells were treated with ETC-159, olaparib or the combination at the indicated dose (for example 0.25 x $ED_{50}$ of Olaparib or ETC-159, or 0.25 x $ED_{50}$ of Olaparib + 0.25 × $ED_{50}$ of ETC-159, respectively). (A) The data is representative of two independent experiments with each point representing an average colony count ± SD of duplicates. (B) The Combination Index (CI) values of olaparib and ETC-159 calculated for the two independent experiments using the Chou-Talalay CompuSyn software. CI < 1, = 1, and > 1 indicate synergism, additive effect, and antagonism, respectively. The lower the CI, the stronger the synergism.

C, D   Olaparib and ETC-159 synergize to prevent the growth of HPAF-II xenografts in mice. NSG mice with established HPAF-II subcutaneous xenografts were randomized into four groups. Mice were gavaged daily with ETC-159 (10 mg/kg), Olaparib (50 mg/kg) or a combination of ETC-159 (10 mg/kg) and Olaparib (50 mg/kg). Treatment was initiated after HPAF-II tumors were established. (C) Tumor volumes were measured starting from day 0 and during the course of treatment as shown. Data points represent the mean ± SD. n = 7–8 tumors/group. P-values indicate significant difference compared to the vehicle group. (D) Tumor weights in the respective groups at the end of 21 days treatment are shown, each dot represents an individual tumor and line represents mean. P-values were calculated with Mann–Whitney U-test.

E, F   Olaparib and ETC-159 synergize in multiple Wnt-addicted cancer cells. Soft agar colony formation assays were performed as in Fig 1A with the indicated cell lines treated with varying concentrations of ETC-159, olaparib or a combination of both as indicated and the combination index calculated from two independent experiments using Chou-Talalay method are shown. PARP inhibitor olaparib and ETC-159 synergistically prevent colony formation of EGI-1, MCAS and CFPAC-1 cells in soft agar. (F) Representative image of soft agar colonies of EGI-1 cells is shown.

downregulation of FANCD2 and RAD51 protein by immunoblotting lysates from this CRC PDX (Fig 2G). Consistent with this being a general phenomenon in Wnt-addicted cancers, HR pathway gene expression as analyzed by qRT–PCR was also decreased upon ETC-159 treatment of an *RNF43* mutant (G371fs) pancreatic cancer patient-derived xenograft as well as in AsPC-1 pancreatic cancer and EGI-1 cholangiocarcinoma cells (Figs 2H and I, and EV1G).

To test the functional impact of reduced HR pathway gene expression on homologous recombination, we used an established assay for homology-directed repair, the Direct Repeats (DR)-GFP assay (Jasin & Rothstein, 2013). We compared the effect of Wnt inhibition on GFP recombination using AsPC-1 cells with transiently expressed or integrated DR-GFP reporter construct. DR-GFP contains a modified GFP gene disrupted by an I-SceI cleavage site and a donor GFP sequence that restores functional GFP upon HR-mediated DNA repair. In DR-GFP AsPC-1 cells, expression of the I-SceI endonuclease led to a four-fold increase in the percentage of GFP-positive cells. However, Wnt inhibition with ETC-159 for 48 h led to ~40% reduction in the percentage of cells undergoing homology-directed repair (GFP-positive) (Fig 2J). Similar to HPAF-II cells, ETC-159 treatment for 48 h did not significantly change the percentage of AsPC-1 cells in S phase compared to the control (Fig EV1H). As a control for off-target effects of ETC-159, we performed the same DR-GFP assay in Panc 08.13 cells that do not have pathologic

▶

**Figure 2. Homologous recombination (HR) and Fanconi anemia (FA) repair pathway genes are regulated by Wnt signaling.**

A   Heatmap of selected temporal clusters containing the Wnt-activated genes that are enriched for DNA repair pathways. Transcriptomic data from HPAF-II orthotopic pancreatic tumors (dataset originally reported in Madan *et al*, 2018) was assessed at multiple time points during treatment with the PORCN inhibitor 37.5 mg/kg *bid* ETC-159 (Fig EV1C). Genes that were differentially expressed over time (FDR < 10%) following PORCN inhibition were clustered based on their pattern of transcriptional response. Clusters 1, 5 and 12 refer to temporal clusters defined in (Madan *et al*, 2018) and $TI_{50}$ is the time (in hours) after start of therapy to achieve 50% inhibition in gene expression for each cluster.

B   Gene Ontology (GO) biological process enrichment of each cluster of *Wnt-activated* genes. Analysis of *Wnt-activated* genes in clusters 1, 5 and 12 from HPAF-II orthotopic pancreatic tumors (Fig 2A) and colorectal cancer (CRC) patient-derived xenograft (PDX) highlights enrichment of genes involved in multiple DNA repair pathways including interstrand cross-link repair and double strand break (DSB) repair via homologous recombination (HR).

C   DNA repair genes involved in HR and FA pathways were differentially expressed over time in response to PORCN inhibition in HPAF-II orthotopic xenografts. Comparison of log₂ fold change (FDR < 10%) in the expression of *Wnt-activated* genes over time across multiple DNA repair pathways shows that genes regulating HR and FA pathways have higher fold changes compared to genes regulating BER, NER or other pathways. BER: base excision repair; HR: homologous recombination; FA: Fanconi anemia; MMR: mismatch repair; NER: nucleotide excision repair; NHEJ: non-homologous end joining; TLS: translesion DNA synthesis. The horizontal line represents median of 4–6 replicates with the upper and the lower edges of the box representing the 75th and 25th percentile of the data respectively and the whiskers representing 1.5× interquartile range.

D, E   ETC-159 treatment of HPAF-II orthotopic xenografts reduces the expression and protein levels of multiple HR and FA pathway genes. (D) Temporal expression of selected *Wnt-activated* HR and FA pathway genes are shown. n = 7–10 tumors/group. (E) Tumor lysates from HPAF-II xenografts treated with vehicle or ETC-159 for 8 or 56 h were analyzed by SDS–PAGE and immunoblotted with the indicated antibodies. Each lane represents an individual tumor.

F, G   Wnt inhibition reduces the expression and protein levels of HR and FA pathway genes in a Wnt-addicted colorectal cancer patient-derived xenograft. Mice with CRC PDX driven by a *RSPO3* translocation were treated with vehicle or ETC-159 for 56 h before harvesting. (F) Gene expression was analyzed by RNA-seq (dataset reported in (Madan *et al*, 2016)). The graph shows the normalized expression of the indicated genes. The horizontal line represents mean of 4 tumors/group. (G) Tumor lysates were analyzed by SDS–PAGE and immunoblotted with the indicated antibodies. Each lane represents an individual tumor.

H, I   Wnt inhibition reduces the expression of HR and FA pathway genes in a *RNF43*-mutant pancreatic cancer patient-derived xenograft and AsPC-1 cells. (H) Pancreatic cancer PDX with G371fs *RNF43* mutation treated with vehicle or ETC-159 (30 mg/kg) for 21 days were analyzed for changes in the expression of the indicated genes measured by qRT–PCR. Each data point represents an individual tumor. The horizontal lines represent mean of 3 tumors/group. (I) AsPC-1 cells were seeded in low adherence plates, treated with DMSO or ETC-159 (100 nM) for 72 h and the expression of indicated genes was measured by qRT–PCR.

J   Wnt inhibition reduces homologous recombination. AsPC-1 or Panc 08.13 cells with a stable integration of HR reporter construct DR-GFP (encoding a modified GFP gene with the I-SceI site and a donor GFP sequence that restores GFP sequence upon HR-mediated DNA repair) were transfected with a plasmid expressing I-SceI endonuclease, followed by treatment with DMSO or ETC-159 (100 nM) for 24 h. The percentage of GFP-positive cells as measured by flow cytometry indicates that the extent of repair via HR pathway in the presence of ETC-159 treatment is reduced in Wnt high AsPC-1 cells or but not in Wnt-insensitive Panc 08.13 cells. Data show the values from three independent experiments with two to three replicates each. P-values were calculated using Mann–Whitney U-test.

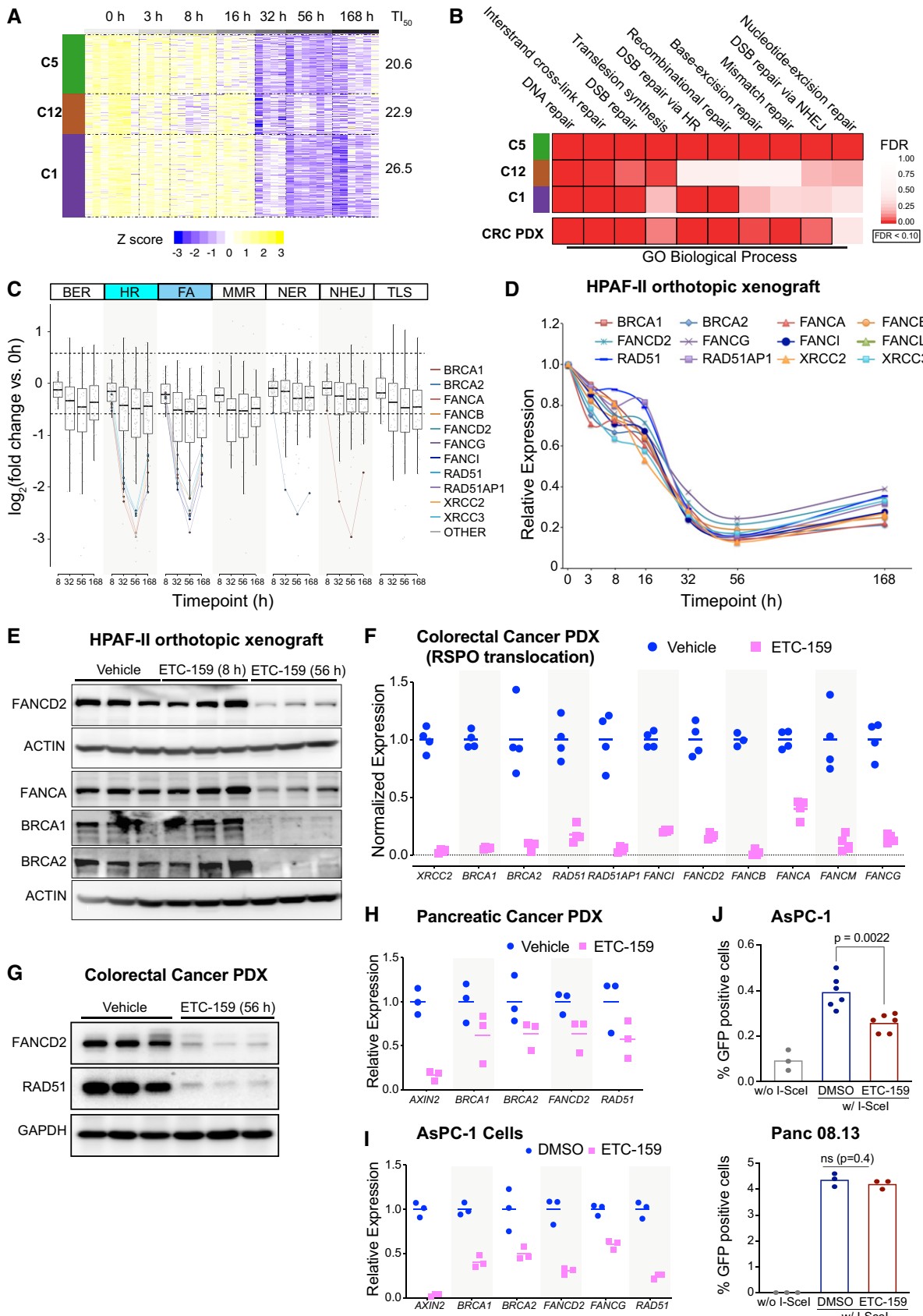

**Figure 2.**

activation of Wnt signaling. In these cells, ETC-159 treatment had no effect on the recombination repair efficiency as measured by percentage of GFP-positive cells (Fig 2J).

Taken together, the data from multiple models of Wnt-addicted cancers demonstrate that inhibition of Wnt signaling sensitizes cells to the PARP inhibitor olaparib by creating a BRCA-like state via downregulation of HR and FA pathways.

**Wnt high cells that are resistant to PORCN inhibition can be sensitized to olaparib**

There is a subset of cells with high Wnt signaling whose growth is not slowed down by PORCN inhibition alone (Jiang *et al*, 2013). PaTu8988T and PL45 cells are Wnt high owing to the loss-of-function RNF43 mutations, F69C and M18fs respectively, and have high abundance of Frizzleds on the cell surface (Fig 3A). Although PaTu8988T cells show reduced expression of Wnt target genes such as *AXIN2* upon knockdown of β-catenin (Fig 3B), their proliferation is relatively insensitive to Wnt inhibition alone, presumably due to a second downstream mutation (Jiang *et al*, 2013). Despite their continued proliferation, PaTu8988T cells showed downregulation of both HR and FA pathway genes in response to Wnt inhibition with either siβ-catenin or ETC-159 treatment (Fig 3B). A high dose of ~1.0 μM of either ETC-159 or olaparib reduced colony formation by only 50% (Fig 3C and Table EV1), with no further reduction even when the dose was increased up to 4.0 μM ($4.0\times$ $ED_{50}$). However, the combination of olaparib and ETC-159 was highly synergistic even at 250 nM ($0.25\times$ $ED_{50}$) (Combination Index < 0.2 at three dosages tested) (Fig 3C and D). The combination of ETC-159 and olaparib was also more effective in preventing the growth of PaTu8988T orthotopic xenografts compared to treatment with the two drugs alone (Fig 3E). Similarly, PL45 cells were also insensitive to single agent treatment with ETC-159 or olaparib, while the combination synergistically inhibited their growth (Fig 3F). This is consistent with ETC-159 creating a BRCA-like state via inhibition of the Wnt pathway rather than through an off-target effect.

On the other hand, Panc 08.13, a pancreatic cancer cell line with no RNF43 mutation, has low abundance of Frizzleds on the cell surface and hence has low Wnt signaling (Fig 3A). Consistent with low Wnt signaling, Panc 08.13 cells were not sensitive to Wnt inhibition and no synergy was observed between ETC-159 and olaparib in soft agar colony formation assay (Fig 3G). Panc 08.13 cells also did not show a reduction in the expression of HR and FA pathway genes upon ETC-159 treatment (Fig 3H). Taken together, these data suggest that the combination of olaparib and ETC-159 may be an effective treatment for some Wnt high cancers with inactivating RNF43 mutations that are resistant to monotherapy with Wnt pathway inhibitors.

**Regulation of DNA repair genes is β-catenin dependent in Wnt-addicted cells**

Wnt signaling regulates gene expression in large part through stabilization of β-catenin (Yu & Virshup, 2014). To test whether HR and FA pathway gene expression was β-catenin dependent, we knocked down β-catenin expression in HPAF-II cells with siRNA. Similar to Wnt inhibition with ETC-159, knockdown of β-catenin in HPAF-II cells caused a marked decrease in the expression of multiple HR and FA pathway genes (Fig 4A). Conversely, we tested whether degradation of β-catenin was required for the downregulation of these genes. We established xenograft tumors using either HPAF-II cells or HPAF-II cells that express a stabilized β-catenin that is insensitive to CK1α/GSK3 phosphorylation and subsequent proteasomal degradation. Tumor-bearing mice were treated with ETC-159 for 56 h, and then, the expression of Wnt target genes and HR and FA pathway genes was compared between the two groups. In the HPAF-II tumors with stabilized β-catenin, treatment with ETC-159 did not impact the expression of Wnt/β-catenin target genes such as *AXIN2*

**Figure 3. Wnt high cells that are resistant to PORCN inhibition can be sensitized by combination with olaparib.**

A   *RNF43* mutant cells have high cell surface abundance of FZDs. Endogenous cell surface FZD levels was assessed by flow cytometric analysis in PaTu8988T cells and PL45 cells (with *RNF43* inactivating mutations) and Panc 08.13 cells with wild-type *RNF43*. Cells were stained with pan-FZD antibody clone F2.A or an isotype control followed by anti-human Fc fragment APC-conjugated secondary antibody. MFI = median fluorescence intensity.

B   Wnt inhibition using siRNA against β-catenin or ETC-159 treatment reduces the expression of HR and FA pathway genes in PaTu8988T cells. PaTu8988T cells were transfected with indicated control or β-catenin siRNAs or treated with ETC-159 (1 μM) for 48 h, total RNA was isolated and expression of β-catenin (*CTNNB1*), *AXIN2* and DNA repair genes was measured by qRT–PCR. The horizontal lines represent mean of replicates.

C, D   ETC-159 and olaparib synergistically inhibit the growth of ETC-159 resistant pancreatic cancer cell line PaTu8988T. (C) PaTu8988T cells were seeded in soft agar and treated with ETC-159, olaparib or the combination of the two inhibitors at an equivalent dose as described in Fig 1A. Representative image of soft agar colonies from PaTu8988T combination study in a 48-well plate is shown. (D) The Combination Index (CI) values of olaparib and ETC-159 calculated from two independent experiments using the Chou-Talalay CompuSyn software.

E   Olaparib and ETC-159 combination is more effective in preventing the growth of PaTu8988T xenografts in mice. NSG mice with established PaTu8988T orthotopic xenografts were randomized into four groups. Mice were gavaged daily with ETC-159 (30 mg/kg), olaparib (50 mg/kg) or a combination of ETC-159 and olaparib. Treatment was initiated after the tumors were established. Tumor weights in the respective groups at the end of the 18 days treatment are shown. Each dot represents an individual tumor and the horizontal lines represent the mean. *P*-values were calculated with Mann–Whitney *U*-test.

F   ETC-159 and olaparib synergistically inhibit the growth of ETC-159 resistant pancreatic cancer cell line PL45. PL45 cells seeded in 48 well plates were treated with ETC-159, olaparib or the combination of the two inhibitors at an equivalent dose as described in Fig 1A. A representative image of the wells from the combination study is shown.

G   Wnt inhibition does not synergize with olaparib in Wnt-low Panc 08.13 cells. Panc 08.13 cells were seeded in soft agar and treated with increasing doses of olaparib alone or in combination with ETC-159 as indicated and allowed to form colonies for 2 weeks. The data is representative of two independent experiments with each point representing an average colony count ± SD of duplicates.

H   Wnt inhibition through ETC-159 does not change the expression of HR and FA pathway genes in Panc 08.13 cells. Panc 08.13 cells were cultured in low adherence plates and treated with ETC-159 for 72 h. Total RNA was then isolated and expression of indicated genes was measured by qRT–PCR. The horizontal lines represent mean of replicates.

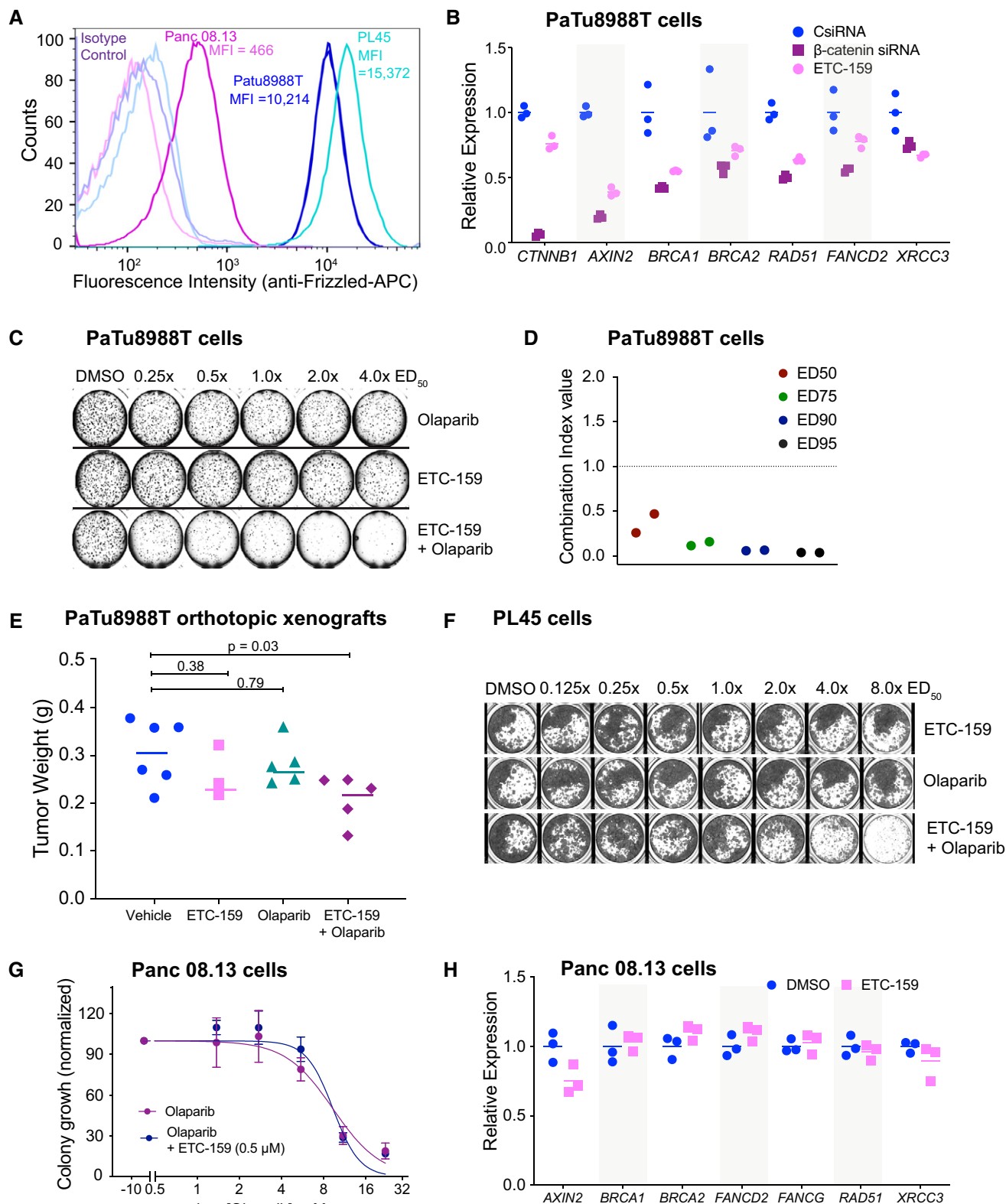

Figure 3.

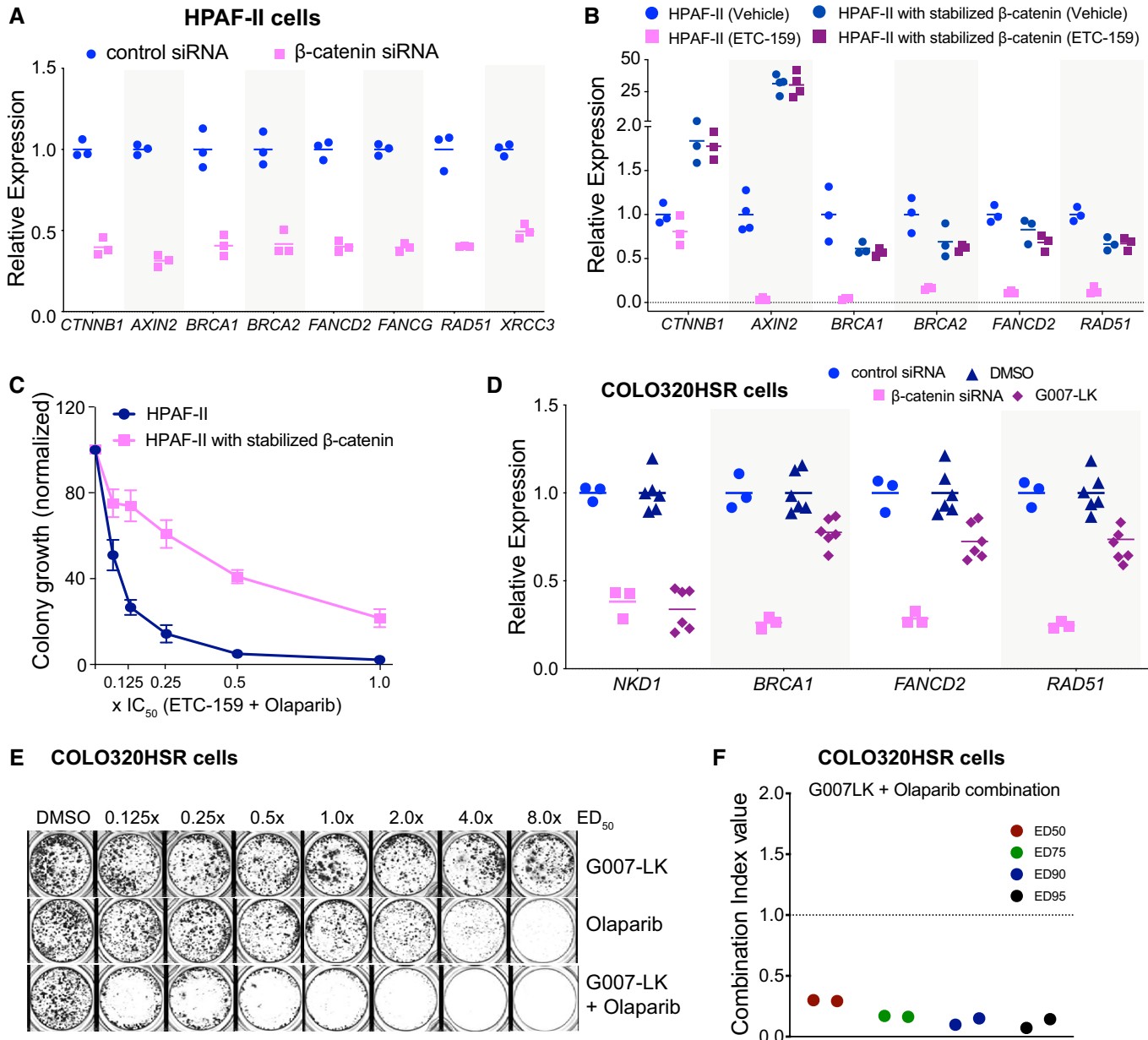

**Figure 4. β-catenin regulates the expression of HR and FA pathway genes.**

A   β-catenin regulates the expression of HR and FA pathway genes in HPAF-II cells. HPAF-II cells were transfected with siRNA against β-catenin for 48 h. The relative expression of genes as measured by qRT–PCR is shown. The horizontal lines represent mean of replicates.

B   Stabilized β-catenin prevents the downregulation of HR and FA pathway genes upon Wnt inhibition. Mice bearing HPAF-II xenografts without or with stabilized β-catenin were treated with ETC-159 (37.5mg/kg *bid*) for 56 h before tumors were harvested and the expression of indicated genes was measured by qRT–PCR. ETC-159 induced reduction in the expression of HR and FA pathway genes was blocked in xenografts with stabilized β-catenin. The horizontal lines represent mean of replicates.

C   Stabilized β-catenin reduces the sensitivity of HPAF-II cells to olaparib and ETC-159. HPAF-II cells without or with stabilized β-catenin were plated in soft agar in 48 well plates. The cells were treated with a combination of olaparib and ETC-159 as in Fig 1A and the total number of colonies were scored after 2 weeks. Data shown are average colony counts ± SD from three replicates.

D   Wnt inhibition reduces the expression of HR and FA pathway genes in *APC*-mutant COLO320HSR cells. COLO320HSR cells were seeded in 6 well plates, treated with control siRNA, siβ-catenin, DMSO or G007-LK for 72 h and then the expression of the indicated genes was measured by qRT–PCR. The horizontal lines represent mean of replicates.

E, F   G007-LK and olaparib synergistically inhibit the growth of COLO320HSR colorectal cancer cells with APC mutation. COLO320HSR cells were plated at a low density and treated with G007-LK, olaparib or the combination of the two inhibitors at an equivalent dose as described in Fig 1A. (E) Representative image from two independent experiments is shown. (F) The Combination Index (CI) values of olaparib and G007-LK calculated from two independent experiments using the Chou-Talalay CompuSyn software.

(Fig 4B). As expected, ETC-159 reduced the expression of HR pathway genes in the control HPAF-II tumors, but this effect was totally abrogated in the tumors with stabilized β-catenin (Fig 4B). Moreover, in a soft agar colony formation assay, we observed that unlike the HPAF-II cells (Fig 1A), co-treatment with ETC-159 failed to sensitize HPAF-II cells with stabilized β-catenin to olaparib (Fig 4C). These findings confirm the importance of β-catenin signaling in the regulation of DNA repair genes. This also serves as a control for demonstrating that the downstream effects are not due to off-target effects of the PORCN inhibitor.

Finally, we tested in a colorectal cancer cell line with an APC mutation if Wnt/β-catenin signaling also regulated the HR and FA pathway. We first confirmed that knockdown of β-catenin in the APC-mutant colorectal cancer cell line COLO320HSR markedly reduced the expression of *BRCA1*, *FANCD2*, and *RAD51* (Fig 4D). We then tested the tankyrase inhibitor G007-LK that inhibits Wnt/β-catenin signaling by an entirely different mechanism, stabilizing the destruction complex protein AXIN so as to enhance the degradation of β-catenin (Lau *et al*, 2013). G007-LK is selective for TNKS1/2 ($IC_{50} < 50$ nM) and does not inhibit PARP1/2 ($IC_{50} > 10$ μM) at therapeutic doses (Lau *et al*, 2013). COLO320HSR cells were treated with increasing concentrations of G007-LK alone or in combination with olaparib (Fig 4E). While G007-LK and olaparib alone were not particularly effective in preventing the growth of COLO320HSR cells, the combination of the two drugs was highly synergistic in inhibiting the growth of these cells (Fig 4E and F). Similar to knockdown of β-catenin, treatment with G007-LK inhibited the expression of DNA repair genes in the COLO320HSR cells (Fig 4D). Taken together with the data in Figs 1 and 3, these results demonstrate that multiple cancers with a diverse array of Wnt-activating mutations can be sensitized to olaparib by regulating the expression of DNA repair genes.

## Identification of a Wnt/β-catenin/MYBL2 pathway

To understand the mechanism of regulation of HR pathway genes by Wnt/β-catenin signaling, we analyzed the promoters of the HR and FA pathway genes that were coherently downregulated upon Wnt inhibition. Transcription factor binding site (TFBS) targets were obtained from the ENCODE, ReMAP, and Literature ChIP-seq datasets using the online tool CHEA3 (Keenan *et al*, 2019). This analysis identified significant enrichment for FOXM1 and MYBL2 binding events (Fig 5A) which is consistent with the established roles of these two transcription factors in the regulation of DNA repair pathways (Khongkow *et al*, 2014; Zona *et al*, 2014; Bayley *et al*, 2018). We next examined if either of these transcription factors was Wnt-activated. Indeed, in our pancreatic cancer transcriptomic dataset, both *FOXM1* and *MYBL2* were robustly Wnt-activated. Expression of both *MYBL2* and *FOXM1* was decreased by ~90% during PORCN blockade (Figs 5B and EV2A). Consistent with this, PORCN inhibition decreased MYBL2 protein abundance in HPAF-II pancreatic tumors (Fig 5C). The decrease in *MYBL2* transcripts as well as protein was abrogated by the expression of stabilized β-catenin, demonstrating that MYBL2 is regulated by β-catenin (Fig 5D). Furthermore, knockdown of β-catenin using an siRNA also reduced the expression of *MYBL2* (Fig 5E). Thus, both *FOXM1* and *MYBL2*, implicated in the regulation of DNA repair pathways, are Wnt-regulated.

We assessed the relative importance of *MYBL2* and *FOXM1* in the regulation of HR and FA genes in the Wnt-addicted cells. Knockdown of *FOXM1* with two independent siRNAs had no effect on HR gene expression (Fig EV2B). However, knockdown of *MYBL2* in HPAF-II cells in culture using two independent siRNAs reduced the expression of HR and FA pathway genes by 30–50% (Fig 5F). Supporting the role of MYBL2 in regulation of DNA repair genes, we

---

**Figure 5. Wnts regulate the expression of HR and FA pathway genes via MYBL2.**

A   Wnt-activated genes show TFBS enrichment for *FOXM1* and *MYBL2*. Promoters of *Wnt-activated* HR and FA pathway genes were scanned for transcription factor binding site (TFBS) motifs using ENCODE, ReMAP, and Literature ChIP-seq databases using the web-based tool, CHEA3.

B   Temporal regulation of *MYBL2* in HPAF-II orthotopic xenografts treated with ETC-159. Each data point represents an individual tumor. The horizontal lines represent median of 4–6 replicates with the upper and the lower edges of the boxes representing the 75th and the 25th percentile of the data respectively and the whiskers representing 1.5× interquartile range.

C   *MYBL2* protein levels are reduced upon ETC-159 treatment in HPAF-II xenografts. Tumor lysates from vehicle or ETC-159-treated (56 h) mice were resolved by SDS–PAGE followed by immunoblotting with the indicated antibodies. Each lane represents an individual tumor.

D   Expression of *MYBL2* is regulated by β-catenin. Expression of *MYBL2* (i) protein levels as measured by immunoblotting and (ii) transcripts as measured by qRT–PCR in HPAF-II tumors with or without stabilized β-catenin treated with vehicle or 100 nM ETC-159 for 56 h. Stabilized β-catenin prevents the ETC-159 induced decrease in both *MYBL2* transcripts and protein levels. The horizontal lines represent mean of replicates.

E   β-catenin regulates the expression of *MYBL2* in HPAF-II cells. HPAF-II cells were either treated with ETC-159 (100 nM) or transfected with siRNA against β-catenin alone or in the presence of ETC-159 for 48 h. The relative expression of *MYBL2* as measured by qRT–PCR is shown. The horizontal lines represent mean of replicates.

F   Expression of DNA repair genes in Wnt-addicted cells is regulated by *MYBL2*. HPAF-II cells were transfected with two independent siRNAs against *MYBL2* or treated with ETC-159 (100 nM) for 48 h. Total RNA was isolated and expression of *MYBL2* and DNA repair genes was measured by qRT–PCR. Data are representative of three independent experiments. The horizontal lines represent mean of replicates.

G   Enrichment of *MYBL2* binding on the promoters of DNA repair genes. HPAF-II cells were treated with DMSO or ETC-159 (100 nM) for 48 h. Chromatin immunoprecipitated with IgG or MYBL2 antibodies was analyzed using primers specific for the DNA binding sites in the indicated DNA repair genes. Data presented are fold enrichment normalized to input. Data are representative of three independent experiments.

H–K   Expression of *MYBL2* is downregulated upon ETC-159 or G007-LK treatment in multiple Wnt-high tumors. Expression of *MYBL2* in AsPC-1 xenograft tumors and colorectal cancer PDX (as measured by RNA-seq) or in pancreatic cancer PDX with *RNF43* mutation (measured by qRT–PCR) is reduced with ETC-159 treatment. G007-LK reduces MYBL2 expression in COL320HSR cells (measured by qRT–PCR). *P*-values were calculated with Mann–Whitney U-test. Each data point represents an individual tumor or replicate. *n* = 4–6 samples/group. The horizontal lines represent median of 4–6 replicates with the upper and the lower edges of the boxes representing the 75th and the 25th percentile of the data respectively and the whiskers representing 1.5x interquartile range.

L   PORCN inhibitor ETC-159 creates a BRCA-like state: Wnt ligand binding to the cell surface receptors Frizzled (Fzd) and LRP5/6 leads to the activation of Wnt/β-catenin target genes such as *MYBL2* that in turn activates the HR and FA pathway genes. ETC-159 treatment inhibits this HR and FA pathway and consequently enhances sensitivity to olaparib.

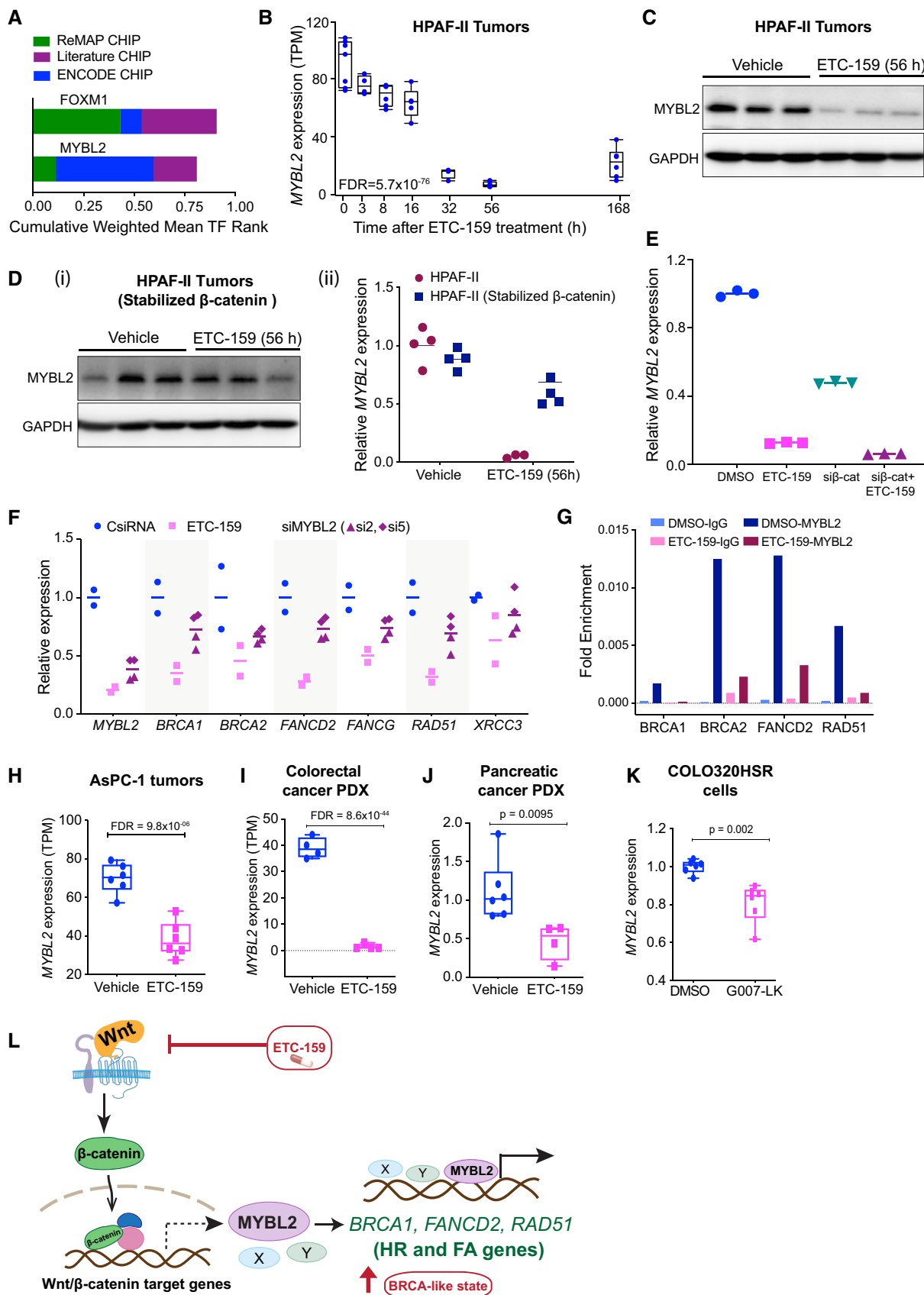

**Figure 5.**

observed that binding sites for MYBL2 were enriched in the promoters of DNA repair genes, but not on the promoters of direct Wnt target genes such as *AXIN2* (Fig EV2D; ENCODE Project Consortium, 2012). MYBL2 binding to the promoters was also confirmed in HPAF-II cells by chromatin immunoprecipitation (ChIP) assay. This enrichment of MYBL2 binding to the promoters of DNA repair genes in HPAF-II cells was reduced upon Wnt inhibition with ETC-159 (Figs 5G and EV2E).

To determine whether the Wnt/β-catenin/MYBL2 axis was a general phenomenon, we examined our previously published datasets (Madan *et al*, 2016, 2018). We observed that, similar to the regulation of DNA repair genes, *MYBL2* expression was regulated by Wnt signaling in AsPC-1 orthotopic xenografts, as well as in a pancreatic cancer PDX with an RNF43 mutation, and a colorectal cancer PDX with an R-spondin translocation (Fig 5H–J). Treatment of COLO320HSR cells with G007-LK also reduced the expression of MYBL2 (Fig 5K). Taken together, these data indicate that the expression of HR and FA genes is in part regulated by a Wnt/β-catenin/MYBL2 axis in multiple cancers (Fig 5L).

Finally, we examined if the Wnt target gene *MYC* played a role in the regulation of the HR and FA genes in Wnt high tumors (Karlsson *et al*, 2003; Guerra *et al*, 2010; Campaner & Amati, 2012). We previously examined HPAF-II tumors containing ectopically expressed stabilized MYC (T58A) where PORCN inhibition did not decrease MYC protein abundance (Madan *et al*, 2018). Both the basal expression of HR and FA pathway genes and their response to ETC-159 treatment was only modestly influenced by the presence of stabilized MYC (Fig EV2C) suggesting that the regulation of HR and FA pathways is not via a Wnt/MYC axis.

**Olaparib and ETC-159 combination enhances DNA damage and senescence**

We examined the mechanism by which combination of Wnt and PARP inhibition caused tumor growth inhibition. PARP inhibition in BRCA1-deficient cancers has been reported to cause apoptosis as well as senescence (Santarosa *et al*, 2009; Sedic *et al*, 2015). We analyzed the HPAF-II xenografts described in Fig 1C and D, following treatment with the combination of the PORCN and PARP inhibitors. Olaparib alone had no effect on the abundance of HR and FA

pathway proteins, while ETC-159 alone and in combination with olaparib reduced their protein levels (Fig 6A). Unexpectedly, neither drug alone, nor in combination, caused an increase in markers of apoptotic cell death. There was no increase in cleaved caspase 3 or cleaved PARP in tumors treated with either ETC-159 or olaparib alone or in tumors treated with a combination of these two inhibitors, showing that synergistic growth inhibition was not working via enhancing cell death by apoptosis (Fig 6B).

As DNA damaging agents can also induce a senescence response, we next analyzed the expression of genes associated with senescence (Sharpless & Sherr, 2015). Among the well-established markers of senescence, *CDKN2A* (encoding p16[INK4a] and p14[ARF]) is inactive in HPAF-II cells due to a large in-frame deletion (COSMIC database). *CDKN2B* (encoding p15[INK4b]) acts as a tumor suppressor by promoting cell cycle arrest and senescence in the absence of *CDKN2A*, especially in pancreatic cancer (Krimpenfort *et al*, 2007; Tu *et al*, 2017). The expression of *CDKN2B* was significantly enhanced in tumors from mice treated with ETC-159 alone and was further increased in tumors treated with the combination containing olaparib (Fig 6C). In addition, expression of the senescence markers *IL32* and *CCN2* and protein levels of p21 (Staal *et al*, 2001; Jun & Lau, 2017; Hsu *et al*, 2019) were also higher in the tumors from the ETC-159 and olaparib combination group compared to the tumors treated with ETC-159 or olaparib alone (Fig 6C and D). Loss of *LMNB1* (lamin B1) is also associated with senescence (Freund *et al*, 2012; Wang *et al*, 2017), and we observed a marked reduction in its expression in the tumors treated with ETC-159 either alone or in combination with olaparib (Fig 6E). Consistent with these changes in the expression of genes associated with senescence, the tumors from the ETC-159-treated group stained positive for senescence-associated β-galactosidase (SA-β-gal), with a significant increase in SA-β-gal staining in the tumors from the ETC-159 and olaparib combination group (Fig 6F and G).

We also tested if Wnt inhibition increased the amount of DNA damage induced by olaparib treatment. We treated the HPAF-II cells with ETC-159 and olaparib alone or in combination for 7 days and analyzed the formation of 53BP1 foci. 53BP1 is a DNA damage response factor that accumulates at double-strand DNA (dsDNA) breaks, forming nuclear foci (Schultz *et al*, 2000). Treatment with olaparib alone resulted in formation of 53BP1 foci and this was

---

**Figure 6.  Combination of Wnt inhibition and Olaparib enhances senescence.**

A       Protein levels of HR and FA pathway genes are reduced upon Wnt inhibition in both ETC-159 treated and combination groups. Tumor lysates from mice treated with vehicle, ETC-159, olaparib or combination of ETC-159 and olaparib (as shown in Fig 1C and D) for 56 h were resolved on a 10% SDS–PAGE gel. The indicated HR and FA pathway proteins were detected by immunoblotting. Each lane represents an individual tumor.

B       Wnt inhibition alone or in combination with olaparib does not induce apoptosis. Tumor lysates from mice treated as above for 56 h or 21 days (Fig 1C and D) were resolved on 12% SDS–PAGE gel and analyzed by immunoblotting with the indicated antibodies. Each lane represents an individual tumor.

C–E   Co-treatment with olaparib and Wnt inhibitor regulates expression of senescence associated genes. (C and E) Expression of senescence-associated genes was analyzed in the tumors from all four treatment groups (treated for 56 h) by qRT–PCR. Each data point represents an individual tumor and horizontal lines represent mean. *P*-values for the significant changes as calculated by Mann–Whitney *U*-test are shown. (D) Tumor lysates from indicated groups were immunoblotted for p21. Each lane represents an individual tumor.

F, G   Wnt inhibition induces senescence, which is further enhanced by co-treatment with olaparib. (F) Representative images of tumor sections from the four treatment groups (treated for 21 days) stained for SA-β-galactosidase, a senescence marker, and counterstained with nuclear fast red. Blue color indicates positive staining for senescent cells. (G) The percentage of SA-β-galactosidase positively stained area (blue) in the representative images from each of the groups is shown. Multiple areas were imaged from three tumor samples per group (indicated as mouse 1, 2, and 3). The horizontal lines represent mean of replicates. *P*-values were calculated by Mann–Whitney *U*-test.

H, I    Wnt inhibition potentiates DNA double strand breaks induced by olaparib treatment. (H) HPAF-II cells were treated with DMSO, ETC-159 (50 nM), olaparib (20 µM), or both for 7 days. 53BP1 foci per cell were then assessed by immunofluorescence. The horizontal lines represent mean of replicates. *P*-values were calculated by Mann–Whitney *U*-test. (I) Representative immunofluorescence images of 53BP1 foci (red) and nuclei counterstained with DAPI (blue) are shown for each group.

---

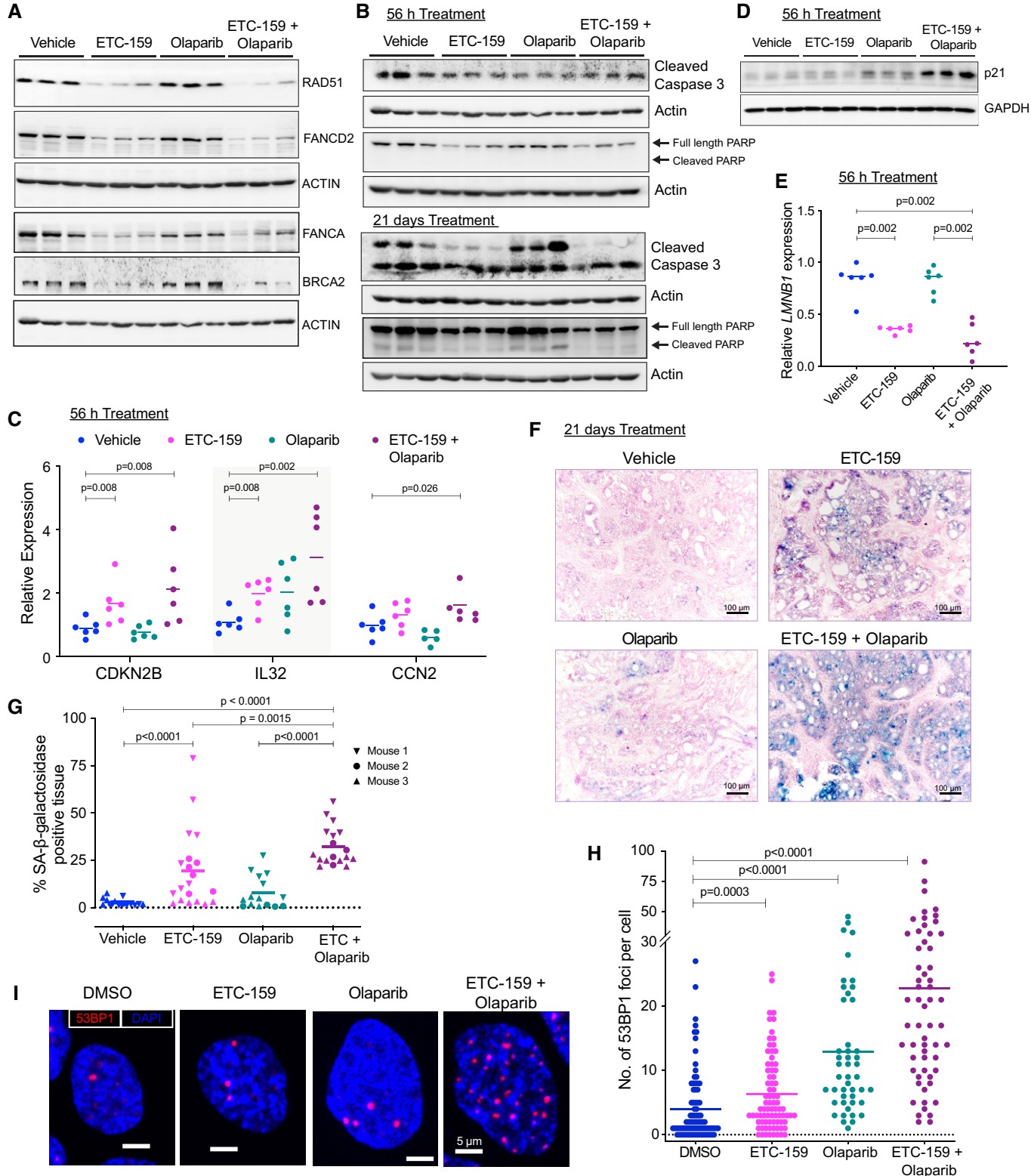

Figure 6.

markedly enhanced in the cells treated with a combination of ETC-159 and olaparib (Figs 6H–I and EV3A). Senescent cells have also been found to have an increase in heterochromatin foci, which can be scored in DAPI-stained cells (Zhang et al, 2007a). We confirmed that treatment with ETC-159 alone and in combination of ETC-159 and olaparib markedly increased the heterochromatin foci, further

supporting the enhanced senescence as observed by SA-β-gal staining (Fig EV3A and B).

Overall, our results suggest that combination therapy works in two steps. First, Wnt inhibition alone induces premature senescence through RAS-MAPK activation (Harmston *et al*, 2020). Second, the reduced expression of DNA repair genes following ETC-159 treatment causes HR deficiency that, when combined with blockage of PARP-mediated ssDNA repair by olaparib, leads to an accumulation of dsDNA breaks that accelerates the development of senescence.

## Wnt signaling drives expression of HR and FA pathway genes in the stem cell compartment of the intestines

The above studies show that inhibition of Wnt signaling in diverse Wnt/β-catenin-high cancers decreased the expression of HR and FA pathway genes, leading to a BRCA-like state and acquisition of PARP inhibitor sensitivity in tumors. To determine whether physiological Wnt signaling also regulates DNA repair pathways *in vivo*, we examined the expression of the HR and FA pathway genes in intestinal crypts. High Wnt/β-catenin signaling in the intestinal crypts is required for maintenance of the stem cell niche with progressive reduction in Wnt signaling as the cells move to the transit-amplifying compartment and then migrate toward the tips of the villi (van der Flier & Clevers, 2009). In agreement with the presence of a Wnt gradient in the intestine, we observed positive nuclear staining for BRCA1, FANCD2, and RAD51 throughout the crypts. Conversely, these proteins were absent in Wnt-low villus epithelial cells (Figs 7A and EV4) consistent with the model that high Wnt signaling promotes the expression of HR and FA pathway genes.

As an independent test of the Wnt/β-catenin/DNA repair axis, we examined two recently published gene expression datasets. *Lgr5* is a robust Wnt/β-catenin target gene in intestinal crypts (Barker *et al*, 2007). Jarde et al. sorted mouse intestinal epithelial cells into *Lgr5* high, mid, low and negative populations and reported RNA-seq data for each pool (Jardé *et al*, 2020). Consistent with our model, the expression of DNA repair genes and *Mybl2* tracked with expression of the Wnt target genes *Lgr5* and *Axin2* (Fig 7B). Similarly in a human intestine single-cell RNA-seq dataset (Wang *et al*, 2020), the expression of *LGR5*, *MYBL2*, and HR/FA genes was highly enriched in the stem, progenitor and transit-amplifying cell clusters (Fig EV5A).

We asked whether enhanced Wnt signaling increases expression of HR and FA genes in intestinal crypts. First, we analyzed tissue sections from the intestines of APC$^{min/+}$ mice where the Wnt/β-catenin pathway is activated by loss of APC (adenomatous polyposis

coli) function (Su *et al*, 1992). The resulting sporadic intestinal adenomas have robust immunostaining for BRCA1, FANCD2, and RAD51 (Fig 7C). Second, analysis of a published dataset of an intestinal organoid model (Dow *et al*, 2015) with inducible activation of Wnt signaling confirmed that there too, knockdown of APC caused an increase in the expression of *Mybl2* and HR/FA DNA repair genes (Fig EV5B).

Finally, to test if the expression of HR/FA pathway genes in intestinal stem cells is Wnt-dependent, we treated mice for 3 days with ETC-159. Therapeutically effective doses of ETC-159 are normally well tolerated by mice and do not impact Wnt target genes in the intestine due to the expression of drug pumps by the stromal Wnt-producing cells in the gut (Chee *et al*, 2018). To overcome this intrinsic drug resistance of the gut stroma, we used a high dose (60 mg/kg) of ETC-159. This treatment significantly reduced the expression of *Mybl2* and the HR and FA pathway genes *Brca1*, *Brca2*, *Fanca*, *Fancd2*, and *Rad51* in the intestine (Fig 7D). The protein levels of FANCD2, RAD51, BRCA1, and phosphorylated-BRCA1 in the intestinal crypts were similarly reduced as visualized by immunostaining (Fig 7E). Thus, Wnt signaling is necessary for the expression of these genes. Taken together, these data demonstrate that Wnt/ β-catenin signaling regulates DNA damage repair gene expression, in part via MYBL2, not only in tumors but also under normal physiological conditions in an adult stem cell compartment.

# Discussion

In this study, we report that Wnt inhibition, via either PORCN or tankyrase inhibition, sensitizes diverse cancers to the PARP inhibitor olaparib. This occurs in part via a Wnt/β-catenin pathway acting in part through MYBL2, that enhances the expression of genes involved in homologous recombination and Fanconi anemia pathways, including *BRCA1*, *BRCA2* and multiple *FANC* genes. Wnt/β-catenin signaling regulates the same genes in human and mouse intestines and mouse colon organoids, suggesting this is a general mechanism by which Wnt/β-catenin signaling can protect the genomes of both Wnt high cancers and adult stem cells.

A genetically defined subset of cancers with high Wnt signaling are vulnerable to Wnt secretion inhibitors. Hence, several small molecules targeting the O-acyltransferase PORCN required for Wnt secretion have advanced to clinical trials. However, cancers inevitably acquire resistance to single agent therapies (Zhong & Virshup,

**Figure 7. HR and FA pathway genes are expressed in the Wnt high compartment of the small intestine.**

A   Immunohistochemical staining of sections of small intestine from C57BL/6J mice for DNA repair proteins shows that the expression of BRCA1, FANCD2 and RAD51 was high in the crypts (Wnt high compartment) whereas the villi did not show any staining.

B   Expression of DNA repair genes correlates with the Lgr5 expression in mouse intestine: Gene expression analysis of sorted Lgr5 populations from the mouse intestine (dataset GSE149311) shows that increased expression of known Wnt target genes *Lgr5* and *Axin2* correlates with *Mybl2*, as well as with *Brca1/2*, *Rad51*, and *Fanc* genes. The horizontal lines represent mean of replicates.

C   Small intestinal adenomas from APC$^{min/+}$ mice displayed positive nuclear immunostaining for HR and FA pathway proteins BRCA1, FANCD2, and RAD51. The bottom panels are higher magnification images of the areas marked in the top panels.

D   PORCN inhibition reduces the expression of HR and FA pathway genes and Mybl2 in the normal mouse intestine. Total RNA was isolated from small intestines of C57BL/6J mice treated with vehicle or ETC-159 (60 mg/kg) for 3 days. Expression of DNA repair genes and *Mybl2* was measured by qRT–PCR. Each data point represents an individual mouse. *n* = 3/group. The horizontal lines represent mean of replicates.

E   PORCN inhibition reduces the expression of HR and FA pathway proteins in the normal mouse intestine. C57BL/6J mice were treated with vehicle or ETC-159 (60 mg/kg) for 3 days. Small intestines were sectioned and analyzed by immunostaining for BRCA1, FANCD2, RAD51 and phosphorylated-BRCA1.

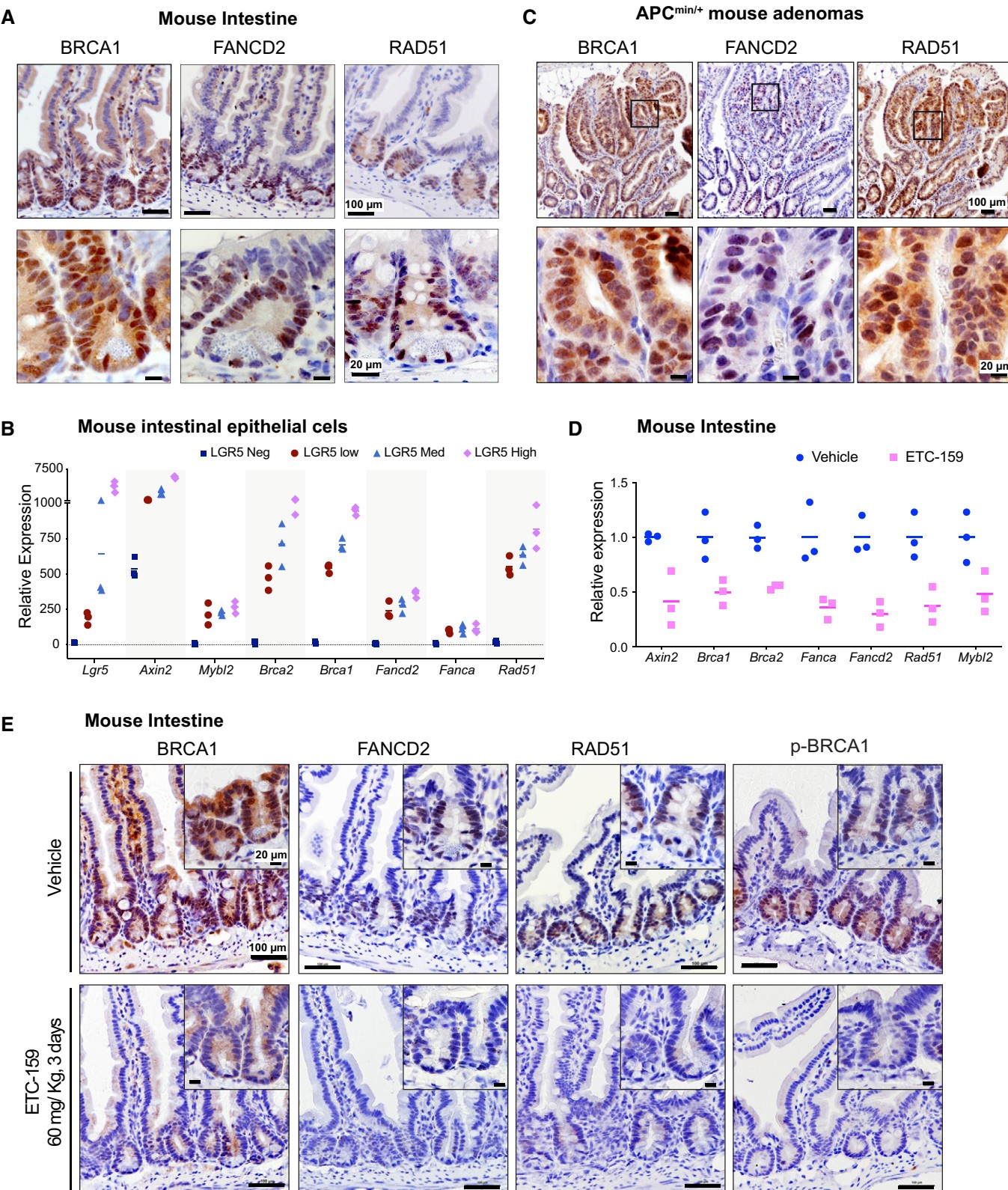

**Figure 7.**

2020). One strategy to overcome drug resistance and reduce adverse effects is synergistic drug combinations. PARP inhibitors like olaparib have successfully moved to the clinics and are now approved treatments of BRCA-mutant ovarian, breast, and pancreatic cancers (Robson *et al*, 2017; Moore *et al*, 2018; Golan *et al*, 2019). The clinical utility of PARP inhibitors is based on the concept of synthetic lethality (Ashworth & Lord, 2018). The enzymes PARP1 and PARP2 are critical for repairing single strand breaks. PARP inhibitors therefore lead to increased unrepaired ssDNA breaks that convert to dsDNA breaks during DNA replication. In the presence of faulty dsDNA break repair, as is seen in BRCA-like states, the accumulated dsDNA breaks cause cell death or senescence (Jackson & Bartek, 2009; Dietlein *et al*, 2014). Therefore, cancers with loss-of-function mutations in HR pathway components such as *BRCA1*, *ATM*, *ATR*, *RAD51*, and *FANC* genes are susceptible to treatment with PARP inhibitors (Farmer *et al*, 2005; Bryant *et al*, 2005; McCabe *et al*, 2006; Lord & Ashworth, 2016; Ashworth & Lord, 2018). One key finding of our study was that inhibition of Wnt signaling leads to a coherent downregulation of the expression of *BRCA1/2*, *RAD51*, *FANCD2*, and multiple other *FANC* genes, inducing a BRCA-like state in the cells and therefore sensitizing them to the PARP inhibitor olaparib.

Here we find that Wnt/β-catenin signaling regulates the expression of multiple HR and FA pathway genes in part via activation of MYBL2. Overexpression of *MYBL2* is observed in several cancers and is associated with poor patient prognosis (Musa *et al*, 2017). In p53 mutant cancer cells, *MYBL2* hyperactivation can prevent DNA damage-induced cell cycle arrest (Mannefeld *et al*, 2009). MYBL2 is known to regulate cell proliferation, differentiation and tumorigenesis (Tarasov *et al*, 2008; Baker *et al*, 2014). However, its role in regulating the HR and FA pathways has only recently been reported. In hematopoietic stem cells (HSCs), *MYBL2* expression was shown to correlate with the levels of DNA repair genes including RAD51, BRCA1/2, and FANCD2 (Bayley *et al*, 2018). Low levels of MYBL2 in myelodysplastic syndrome (MDS) patients also preceded transcriptional deregulation of DNA repair genes (Yang *et al*, 2019). Our study for the first time shows that Wnt/β-catenin signaling regulates the expression of MYBL2 and this axis regulates the expression of HR and FA pathway genes.

Induction of apoptotic cell death is a well-established tumor suppressive response initiated by DNA damaging agents. We did not find any evidence for apoptosis in tumors treated with ETC-159 alone or in combination with olaparib. It has increasingly become clear that several other mechanisms operate to prevent tumor progression. Interestingly, senescence is one of the major outcomes following treatment with agents causing DNA cross-links (Santarosa *et al*, 2009), double-strand breaks (Robles & Adami, 1998), and PARP inhibition (Fleury *et al*, 2019). Moreover, depletion of *BRCA1*, *FANCD2*, and other *FANC* genes is also linked with premature induction of senescence in response to DNA damaging insults (Zhang *et al*, 2007b; Santarosa *et al*, 2009; Sedic *et al*, 2015; Helbling-Leclerc *et al*, 2019). Based on these reports and our results, we propose that combination therapy works in two steps. First, ETC-159 alone induces premature senescence. Second, ETC-159 treatment inhibits Wnt/β-catenin/MYBL2 pathway leading to HR deficiency, and this combined with blockage of PARP-mediated ssDNA repair in proliferating cells leads to accumulation of dsDNA breaks. This is further enhanced by sustained defective DNA damage response caused by

olaparib treatment. Consistent with this, MYBL2 has been shown to inhibit senescence in fibroblasts and HeLa cells (Masselink *et al*, 2001; Martinez *et al*, 2011; Mowla *et al*, 2014).

A role for Wnt signaling in the control of DNA repair is additionally supported by the high expression of DNA repair genes in the intestinal stem and transit-amplifying cell compartments where Wnt-regulated gene expression is high (van der Flier & Clevers, 2009). Prior studies have also suggested Wnt signaling drives enhanced DNA repair. In normal intestine, $Lgr5^+$ cells are radioresistant (Metcalfe *et al*, 2014) (Hua *et al*, 2012; Sato *et al*, 2020), although there is a dissenting view using indirect measures of DNA repair (Tao *et al*, 2015). In cancer, WNT16B is a driver of resistance to cytotoxic chemotherapy in prostate cancer (Sun *et al*, 2012), PARP1 is high in colorectal cancer (Dörsam *et al*, 2018), and *WNT3A* expression is a cause of *in vitro* olaparib resistance in an ovarian cancer cell line (Yamamoto *et al*, 2019). Most recently, mutational inactivation of RNF43 in gastric cancers was found to induce resistance to DNA damage (Neumeyer *et al*, 2020). Further, we find that treatment of mice with Wnt secretion inhibitors reduced the expression of both *Mybl2* and DNA repair genes in the intestinal crypts. Thus, the role of Wnts in regulating HR and FA pathway genes extends beyond cancers. Wnt signaling in normal stem cell compartments may serve to prevent the mutations arising from normal genotoxic stresses that the intestinal epithelium is exposed to. Notably, ETC-159 is well tolerated at therapeutically effective doses without gut toxicity. This is due to the selective expression of drug-efflux pumps in the Wnt-producing intestinal myofibroblasts (Chee *et al*, 2018). Therefore, a low dose of ETC-159 used in combination treatment with olaparib results in tumor-selective inhibition of DNA repair pathways while sparing the intestinal stem cells.

Cells maintain genomic integrity by repairing DNA lesions arising from intracellular processes, such as DNA replication, and extrinsic factors, such as radiation and chemotherapeutic drugs (Jackson & Bartek, 2009). Activation of Wnt signaling is also one of the mechanisms that promotes radioresistance in cancers such as glioblastoma, colorectal, esophageal, and nasopharyngeal cancers. This was mainly attributed to the ability of Wnts in promoting cellular proliferation and maintenance of stem cells (Kim *et al*, 2012; Emons *et al*, 2017; Jun *et al*, 2016; Zhao *et al*, 2018). Our study suggests that activation of Wnt signaling in these cancers may enhance their ability to repair DNA damage and hence induce resistance and maintenance of stem cells.

Our findings that inhibition of Wnt signaling regulates DNA repair pathways, specifically the high-fidelity homologous recombination pathway for repairing double-strand breaks, has several implications. It improves our understanding of how high Wnt signaling maintains stemness, provides options for the treatment of Wnt high cancers by a combination of Wnt inhibitors with PARP inhibitors, and suggests options for overcoming resistance to radiation and chemotherapies by inhibiting Wnt signaling.

# Materials and Methods

### Mice, reagents, and cell lines

C57BL/6J and NSG mice were purchased from InVivos (Singapore) or Jackson Laboratories (Bar Harbor, ME, USA). The Duke-NUS

Institutional Animal Care and Use Committee approved all animal studies. Both male and female mice of ages between 8 and 16 weeks were used for the experiments. Animals were housed in standard cages and were allowed access *ad libitum* to food and water. Mice were treated with ETC-159 or olaparib daily by oral gavage.

ETC-159 was manufactured by Experimental Therapeutics Centre, A*STAR, Singapore. Olaparib (Cat #O-9201) was purchased from LC Laboratories, Woburn MA, 01801. HPAF-II, CFPAC-1, AsPC-1, and Panc 08.13 and COLO320HSR were obtained from ATCC, PaTu8988T and EGI-1 from DSMZ. HPAF-II and EGI-1 cell lines were cultured in MEM media with Earle's salts and non-essential amino acids (Gibco #10370021), AsPC-1 and COLO320HSR cells in RPMI (Nacalai Tesque #0517625), CFPAC-1 cells in Iscove's modified Dulbecco's medium (StemCell Technologies #36150) and PaTu8988T cells in DMEM media (Nacalai Tesque #0845935). All media were supplemented with 1 mM sodium pyruvate (Lonza #BW13-115E), 2 mM L-glutamine (Gibco #25030081), 10% FBS (Hyclone SV30160.03), and 1% penicillin/streptomycin antibiotics (Gibco #15140122) and maintained on adherent cell culture flasks in a humidified incubator at 37°C and 5% $CO_2$. Panc 08.13 cells were cultured in RPMI media with 15% FBS, 4 µg/ml insulin (Gibco #12585014), and the above-mentioned supplements. All cell lines were routinely tested to be negative for mycoplasma.

For suspension cell culture, optimized number of cells were seeded per well into 24-well ultra-low attachment plates (Corning Costar #3473). Cell clusters and spheroids suspended in the media were collected by centrifugation and rinsed with 1× PBS before RNA isolation.

## Soft agar and low-density colony formation assay

HPAF-II, EGI-1, MCAS, CFPAC-1, PaTu8988T, or Panc 08.13 cells were plated in 48-well suspension cell culture plates. For each well, optimized number of cells (2,000–5,000 cells per well) mixed with 0.3% agar (Sigma-Aldrich, A1296) or agarose (Sigma-Aldrich, A9045) in 250 µl complete culture media were layered on top of 300 µl of base layer containing 0.6% agar or agarose in complete culture media. After solidification of the agar layers, 250 µl complete culture media containing small-molecule inhibitors or DMSO control was added to each well. The cells were allowed to form colonies for 2–3 weeks depending on the colony forming status of each of the cell lines. The colonies were then stained with MTT (3-(4,5-dimethylthiazol-2-yl)-2,5-diphenyltetrazolium bromide) and counted by GelCount (Oxford Optronix, Abingdon, UK). The $ED_{50}$ (the dose that leads to 50% of the maximum colony formation suppressive effect in soft agar) of the inhibitors was determined for each of the cell lines and is listed in Table EV1. For determining synergy, cells were treated with either ETC-159, olaparib or a concentration gradient of the combination of the two inhibitors (two replicates per condition). Averages of the total number of colonies obtained as a fraction of the control were then used to determine the Combination Index values using the Chou-Talalay CompuSyn Software (http://www.combosyn.com/; Chou, 2010).

For low density plating studies, 2,000 cells were plated per well in 48-well cell culture plates coated with poly-L-lysine (Sigma-Aldrich, P4707). PL45 or COLO320HSR cells were treated with single drug or combinations of indicated drugs and DMSO control, and cells were allowed to grow for 1–2 weeks depending on the

confluency status. The cells were then stained with 0.5% crystal violet solution (Sigma-Aldrich, V5265) in 25% methanol and counted by GelCount. $ED_{50}$ and combination index values were determined using CompuSyn as above.

## Tumor implantation and treatment of mice

Mouse xenografts were established by subcutaneous injection of HPAF-II cells or HPAF-II cells with stabilized β-catenin in NSG mice. $5 \times 10^6$ HPAF-II cells resuspended in 50% Matrigel were used for injection. Patu8988T cells were injected orthotopically into the pancreas of the mice. Mice were treated with ETC-159 or olaparib after establishment of tumors. ETC-159 was formulated in 50% PEG 400 (vol/vol) in water and administered by oral gavage at a dosing volume of 10 µl/g body weight. Olaparib formulated in 50% PEG 400 (vol/vol) in water was administered intraperitoneally at a dosing volume of 10 µl/g body weight. Tumor dimensions were measured with a caliper routinely, and the tumor volumes were calculated as $0.5 \times \text{length} \times \text{width}^2$. All mice were sacrificed 8 h after the last dosing.

## Western blot

Protein lysates were prepared by homogenizing the tumor tissues in 4% SDS. Protein concentration was determined using Pierce BCA protein assay kit (Thermo Scientific 23225). Equal amount of protein (~40 µg) was loaded and resolved on a 10% or 12% SDS polyacrylamide gel and then transferred to PVDF membranes. The membranes were probed with antibodies against BRCA1 (Cell Signaling Technology 9010, 1:1,000 dilution and Santa Cruz Biotechnology sc-6954, 1:500 dilution), BRCA2 (Cell Signaling Technology #10741, 1:1,000 dilution), RAD51 (Abcam ab133534, 1:1,000 dilution), FANCD2 (Abcam ab108928, 1:1,000 dilution), FANCA (Proteintech 11975-1-AP, 1:1,000 dilution), MYBL2 (Santa Cruz Biotechnology sc-81192, 1:1,000 dilution), cleaved Caspase-3 (Cell Signaling Technology 9661, 1:500 dilution), PARP (BD Pharmingen 556494, 1:500 dilution), and p21 (BD Pharmingen 556430, 1:1,000 dilution). Actin (Abcam ab3280, 1:3,000 dilution) or GAPDH (Abcam ab8245, 1:2,000 dilution) antibodies were used as protein loading controls. HRP conjugated anti-rabbit IgG (Bio-Rad 1706515) or anti-mouse IgG (Bio-Rad 1721011) secondary antibodies were used at 1:10,000 dilution. Blots were visualized on ImageQuant LAS 4000 imager (GE Healthcare Life Sciences).

## Flow cytometric analysis

For detecting endogenous Frizzled levels, PaTu8988T, PL45, or Panc 08.13 cells were seeded into 6-well plates. After 48 h, cells were stained with 1:100 diluted pan-FZD antibody clone F2.A (Steinhart *et al*, 2017). Anti-human Fc fragment APC-conjugated (Jackson Laboratory, #109-135-098) was used as a secondary antibody. The cells were acquired on BD LSRFortessa and analyzed using FlowJo v10 software.

For cell cycle analysis, HPAF-II or AsPC-1 cells were seeded in 6-well plates and treated with ETC-159 as indicated. Cells were harvested by trypsinization, washed with PBS, and pelleted by centrifugation at 300 $g$ for 7 min. The cells were fixed in prechilled 70% ethanol, treated with RNase (5 µg/50 µl), and stained with propidium iodide (15 µg in 300 µl) at 37°C for 30 min in dark. The cells were acquired on BD FACScan and analyzed using FlowJo software.

## Immunohistochemistry

Small intestines from C57BL/6J mice were flushed with PBS, longitudinally cut open, fixed with 10% neutral-buffered formalin for 24 h, and embedded as swiss-rolls in paraffin. For immunohistochemistry, 5 μm paraffin sections were deparaffinized, rehydrated, and boiled in 10 mM sodium citrate, pH 6.0 solution for 25 min. The sections were then rinsed in TBS and blocked with TBS containing 3% BSA and 0.1% Tween-20 for 1 h at room temperature followed by incubation with primary antibody diluted in the blocking buffer overnight at 4°C. Next day, sections were rinsed in TBS, incubated with 3% $H_2O_2$ for 15 min at room temperature and then incubated with goat anti-rabbit IgG-HRP (Bio-Rad 170-6515) secondary antibody diluted 1:300 in the blocking buffer for 1 h at room temperature. Staining was visualized with the Liquid DAB$^+$ Substrate Chromogen System (Dako K3468). Primary antibodies used were RAD51 (Abcam ab133534, 1:400 dilution), FANCD2 (Abcam ab108928, 1:80 dilution), BRCA1 (Proteintech 20649-1-AP, 1:100 dilution), and phosphorylated-BRCA1 (Ser1524) (Thermo Fisher Scientific PA5-36627, 1:100 dilution). Images were acquired using Nikon Ni-E microscope with DS-Ri2 camera.

## Immunostaining and quantification

HPAF-II cells were seeded on glass coverslips and treated with the indicated drugs for 7 days. Following fixation in 4% paraformaldehyde, cells were permeabilized with 0.3% Triton X-100 for 15 min, blocked for 1 h with 2% BSA and 0.1% Triton X-100 in TBS, and incubated with anti-53BP1 antibody (Novus NB100-304, 1:1,000 dilution) at 4°C overnight. Cells were stained with anti-rabbit IgG, Alexa Fluor 594 antibody (Invitrogen # A11012, 1:2,000 dilution) and DAPI (0.5 μg/ml) for 1 h at room temperature and mounted in ProLong Diamond Antifade Mountant (Invitrogen #P36961). Cells were imaged at 60× magnification, LSM710 confocal microscope, and the images were quantified using ImageJ (*Rasband*, *W.S.*, *ImageJ*, *U. S. National Institutes of Health*, *Bethesda*, *Maryland*, *USA*, https://imagej.nih.gov/ij/, *1997–2018*). The number of foci within each nucleus was counted using the "Find Maxima" function in ImageJ with a prominence value (pixel intensity cut off) of 50. Approximately 50–80 cells were counted for each treatment group. The significance was calculated by Mann–Whitney *U*-test.

## Senescence associated (SA)-β-galactosidase staining

Fresh frozen OCT-embedded tumor tissues were cut into 8 μm thick sections. The sections were fixed in 2% paraformaldehyde with 0.125% glutaraldehyde solution at room temperature for 5 min and rinsed with 1× PBS containing 2 mM $MgCl_2$. Thereafter, sections were incubated with X-gal containing staining solution from senescence detection kit (Abcam ab65351) according to manufacturer's instructions. The sections were counterstained with nuclear fast red (NFR) for 5 min. Slides were mounted in mounting media containing gelatin and glycerol. Brightfield images were acquired at 10× magnification using Nikon Ni-E microscope with DS-Ri2 camera. Multiple fields were imaged per tissue section from three tumor samples per treatment group. Quantifications were performed using Fiji ImageJ software. Briefly, color deconvolution was performed to segregate the SA-β-galactosidase (blue stain)-

positive tissue. Image threshold values were calculated and set to measure the percentage area covered by the entire tissue (NFR stained pink regions) and SA-β-galactosidase-positive (blue stained) tissue. Percentage of SA-β-galactosidase-positive tissue area was calculated, and individual values were plotted with GraphPad Prism. Statistical significance was calculated by Mann–Whitney unpaired 2-tailed *t*-test, and *P*-value of < 0.05 was considered a significant difference.

## Chromatin immunoprecipitation

$2 \times 10^7$ HPAF-II cells treated with DMSO or 100 nM ETC-159 for 48 h were fixed in 1% formaldehyde. After quenching with 125 mM glycine, cross-linked cells were washed and lysed sequentially with Buffer A (10 mM HEPES pH 8.0, 10 mM EDTA, 0.5 mM EGTA, 0.25% Triton X-100), Buffer B (10 mM HEPES pH 8.0, 200 mM NaCl, 1 mM EDTA, 0.5 mM EGTA, 0.01% Triton X-100) and resuspended in Buffer C (25 mM Tris–HCl pH 8.0, 150 mM NaCl, 2 mM EDTA, 1% Triton X-100, 1% SDS). Sonication was performed at 30% amplitude for 12 cycles of 30 s pulses using Vibra-Cell CV18 (Sonics & Materials, Inc.). The sonicated chromatin was immunoprecipitated overnight at 4°C with Protein G-Dynabeads (Invitrogen) loaded with anti-MYBL2 (Bethyl Laboratories #A301-654A) or IgG antibody. The immunoprecipitated chromatin was eluted and reverse-cross-linked at 65°C. DNA was purified and enrichment quantitated using qRT–PCR using the PCR primers listed in Table EV2. PCR products were resolved on a 12% acrylamide gel in Tris-Borate-EDTA buffer and stained with ethidium bromide.

## RNA analysis

Total RNA from small intestines, cell lines or tumors were extracted using the RNeasy kit (Qiagen #74106). For real-time quantitative PCR (qRT–PCR) analysis, RNA was converted to cDNA using iScript cDNA synthesis kit (Bio-Rad #1708891), and SsoFast Eva Green Supermix (Bio-Rad #1725205) was used for qRT–PCR. Gene expression was normalized to *EPN1* and *ACTB* for human cell lines, and *Epn1* and *Pgk1* for mouse tissues. The qRT–PCR primers are listed in Table EV2.

## DR-GFP homologous recombination assay

AsPC-1 and Panc 08.13 cells were transfected with pDR-GFP plasmid (a gift from Maria Jasin; Addgene #26475) (Pierce *et al*, 1999). Cells were selected with puromycin (1.2 μg/ml), and single-cell clones with stably integrated DR-GFP plasmid were generated. Genomic DNA was isolated from the single-cell clones using QIAamp DNA mini kit (Qiagen #51304), and qRT–PCR was performed using DR-GFP-CN1 and DR-GFP-CN2 primers (Table EV2) to assess the pDR-GFP integration. The expression of DR-GFP inserts was normalized to *ZNF80* and *GPR15* genes (Table EV2). A HeLa-DR-GFP clone (Parvin *et al*, 2011) with a single copy of pDR-GFP was used as a control to determine the copy number in AsPC-1 and Panc 08.13 DR-GFP single-cell clones.

AsPC-1 and Panc 08.13 DR-GFP clones were seeded at optimal density in 24- or 12-well plates and treated with DMSO or 100 nM ETC-159 for 48 h. The cells were transfected with pCBASceI (a gift

**The paper explained**

**Problem**
Resistance to DNA damage in both cancers and adult stem cells is often associated with activation of the Wnt signaling pathway. Understanding how Wnt signaling drives chemotherapy resistance is of major interest and may assist in developing novel anti-cancer therapies.

**Results**
We found that Wnt signaling turns on a broad set of genes involved in repairing DNA double-strand breaks and inhibiting Wnt signaling can reverse this. These DNA repair genes, including BRCA1, normally maintain genomic integrity in stem cells and cancers. This helps explain how activated Wnt signaling can drive chemotherapy resistance in both stem cells and cancer.

**Impact**
We find that Wnt pathway inhibition synergizes with PARP inhibition in multiple Wnt-addicted cancers. Wnt pathway inhibitors may be a useful intervention to reverse drug resistance in Wnt-high cancers.

from Maria Jasin; Addgene #26477) (Richardson *et al*, 1998) or pcDNA3.2/V5-DEST (control) plasmid using Lipofectamine2000 (Thermo Fisher Scientific, #11668500) After 6 h, cells were again treated with DMSO or ETC-159. After 2 days, cells were harvested by trypsinization, acquired on BD LSRFortessa, and analyzed using FlowJo v10 software for expression of GFP. The percentage of GFP-positive cells was used as a measure of cells undergoing homologous recombination.

**RNA-seq analysis**

RNA-seq datasets were analyzed and clustered as described in Madan *et al* (2018). Briefly, reads originating from mouse (mm10) were removed using Xenome, and the remaining human reads aligned against hg38 (Ensembl version 79) using STAR v2.5.2 (Dobin *et al*, 2013) and quantified using RSEM v1.2.31 (Li & Dewey, 2011). Differential expression analysis was performed using DESeq2 (Love *et al*, 2014). The same pipeline was used to process and analyze RNA-seq reads from a CRC PDX model treated with ETC-159 (Madan *et al*, 2016). For the clustering of gene expression changes in HPAF-II orthotopic model, all genes differentially expressed over time (DESeq2, false discovery rate (FDR) < 10%) were clustered using GPClust (Hensman *et al*, 2013).

**Functional enrichment analysis**

For the analysis of the *Wnt-activated* genes, Gene Ontology (GO) enrichments were performed using GOStats (Falcon & Gentleman, 2007) using all genes differentially expressed (FDR < 10%) as background. Terms with an FDR < 10% were defined as significantly enriched.

# Data availability

This study includes no data deposited in external repositories.

**Expanded View** for this article is available online.

## Acknowledgements

We acknowledge the assistance of members of the Virshup lab and members of Experimental Therapeutics Centre. We thank Dr. David Hsu at Duke University for useful discussions. We acknowledge the assistance of the vivarium staff including Hock Lee. We thank Ray Dunn for sections of APC$^{min/+}$ mouse intestines. We thank Dr Stephane Angers, Dr. Sachdev Sidhu and University of Toronto for the pan-FZD antibody. This research is supported in part by the National Research Foundation Singapore and administered by the Singapore Ministry of Health's National Medical Research Council NMRC/STAR/0017/2013 (to DMV under the STAR Award Program) and NMRC/OFIRG/0055/2017 (to BM under OF-IRG). BM also acknowledges the support of Duke/Duke-NUS collaborative Research Grant 2017/0040.

## Author contributions

BM, EP, and DMV designed the study. BM and AK performed the animal studies. BM, AK, JYSL, SP, and SS performed the biochemical analysis. NH performed the bioinformatics analysis. BM, MAL, EP, and DMV supervised the study. BM, AK, EP, and DMV wrote the manuscript.

## Conflict of interest

Babita Madan and David M. Virshup have a financial interest in ETC-159. The authors declare that they have no conflict of interest.

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
