## [Review Process File · EMBO Molecular Medicine]

WNT inhibition creates a BRCA-like state in Wnt-addicted cancer

Amanpreet Kaur, Jun Yi Stanley Lim, Sugunavathi Sepramaniam, Siddhi Patnaik, Nathan Harmston, May Ann Lee, Enrico Petretto, David Virshup, and Babita Madan

DOI: [10.15252/emmm.202013349](https://doi.org/10.15252/emmm.202013349)

Corresponding authors: Babita Madan (babita.madan@duke-nus.edu.sg)

Review Timeline:

Transfer from Review Commons:	27th Aug 20
Editorial Decision:	31st Aug 20
Revision Received:	30th Nov 20
Editorial Decision:	22nd Dec 20
Author Correspondence:	12th Jan 21
Editor Correspondence:	19th Jan 21
Revision Received:	25th Jan 21
Accepted:	1st Feb 21

Editor: Lise Roth

Transaction Report:

This manuscript was transferred to EMBO Molecular Medicine following peer review at Review Commons.

Review #1

1. How much time do you estimate the authors will need to complete the suggested revisions:

Estimated time to Complete Revisions (Required)

(Decision Recommendation)

Between 1 and 3 months

2. Evidence, reproducibility and clarity:

Evidence, reproducibility and clarity (Required)

In this manuscript Kaur et al uncover the Wnt-bcatenin-MYBL2 signaling axis that appear to be important for the regulation of expression of several genes involved in DNA damage pathways. This pathway is not only active in cancer cells exhibiting high Wnt signaling activity but also in endogenous contexts where high Wnt signaling is required such as in tissue stem cell self-renewal. Given the shared properties of radio and chemo resistance of tumor cells and stem cells this pathway may constitute a general mechanism to protect cells from genomic insults. The authors also exploit their findings as a therapeutic vulnerability by modulating Wnt-bcatenin signaling activity and creating a "BRCA-like state" that render the cells susceptible to treatment with the PARP inhibitor Olaparib. Considering that the stem cell environment is "protected" from therapeutic levels of Wnt inhibitors by drug efflux pumps, this combinatorial treatment is proposed to selectively harm tumor cells and be more efficient than the individual treatments. Together this is a very original piece of work revealing a novel signaling pathway downstream of B-catenin activity that uncovers the molecular basis underlying radioresistance of tumour cells and stem cells. Importantly, this pathway also provides a rational for combinatorial treatment with Wnt and PARP inhibitors for cancer treatment and possibly preventing the emergence of resistance. Below, I list major and minor points that are meant as constructive criticisms of the study to potentially strengthen the findings. **Major points:** 1. MYBL2 expression is found to be induced in Wnt-bcatenin activated cells and to be inhibited by PORCN inhibitors. Is MYBL2 a bcatenin target gene? Can the authors perform CHIP with bcatenin or LEF/TCF on the MYBL2 promoter or identify LEF/TCF elements within the MYBL2 promoter. 2. As stated by the authors, PARP inhibitors such as Olaparib, are effective only in the context of HR pathway mutations such as BRCA1. It is therefore surprising that HPAF cells (which presumably do not contain HR pathway mutation) appear to be quite sensitive to Olaparib treatment, both in vitro and in vivo. What is the explanation for this? 3. The authors assess the reduction in HR upon treatment with Wnt inhibitors with the DR-GFP assay and state that the direct observation of DNA damage with quantification of foci was impossible. It is understood that the expression of several DNA repair genes normally present at foci is decreased but can a marker such as gH2AX be used? 4. In figure 1C, presumably the decrease in tumor volume in ETC-159 and Olaparib is

statistically significant? This should be indicated. Also, is the combination treatment statistically different from the individual treatments? 5. The findings that inhibiting tankyrase also synergized with Olaparib represent an important extension of the work given the more frequent mutations of distal Wnt pathway components. It would therefore be important to show that tankyrase treatment in APC mutant tumors also lead to decreased expression of DNA repair genes to validate their mechanism in this context. **Minor Points:** 1. On line 147 the authors state that the genes were decreased by 8-10 folds. Figure 2D however does not show fold and is expressed as fraction of expression compared to no treatment. 2. In figure 2D, is there an explanation for the absence of difference in tumor weight in the ETC-159 condition? Is this due to differentiation of tumors by ETC-159 leading to mucus loading and hence heavier tumors? 3. Why the authors performed the DR-GFP assay in Aspc-1 cells and not in HPAF-II in which most of the work was performed? 4. The legend for Figure 3 needs to be adjusted to correspond to the panels.

3. Significance:

Significance (Required)

This is a significant study revealing molecular understanding underlying the contribution of Wnt-bcatenin signaling to radioresistance of tumor cells and stem cells. The findings provide an interesting therapeutic opportunity for combinatorial treatment using Wnt and PARP inhibitors for treatment of cancers.

Review #2

1. How much time do you estimate the authors will need to complete the suggested revisions:

Estimated time to Complete Revisions (Required)

(Decision Recommendation)

Between 3 and 6 months

2. Evidence, reproducibility and clarity:

Evidence, reproducibility and clarity (Required)

Summary: The manuscript by Kaur et al. provides evidence that Wnt-signalling drives a DNA repair phenotype. Using a combination of genetic and chemical inhibitors of the specific pathways they narrow this finding down to the HR and FA repair pathways. They further show that this seems to be mediated via a Wnt/ β -catenin/Mybl2 axis. These findings are replicated in a number of Wnt-high cancer models, and in an in vivo setting in the intestinal crypt. This is an interesting and important finding, but I

have a number of significant comments, as detailed below. ****Major comments:**** *- Are the key conclusions convincing?* -The key conclusions are largely convincing. In most cases, multiple experiments are carried out in order to arrive at a conclusion (with some caveats mentioned below). There are a few additional things I would like to see, but generally this seems to be a well carried out study. I have particular reservations regarding the senescence and intestinal data, as the authors do not show that the same mechanisms are at play. *- Should the authors qualify some of their claims as preliminary or speculative, or remove them altogether?* -My main issue with this manuscript is the inconsistent use of different cell lines. At various points HPAF-II, EGI-1, MCAS, CFPAC-1 and COLO320-HSR are used, without any explanation of why specific lines were used for different experiments. It has a feeling of cherry-picking, and to be truly convinced I would like to see consistency in this aspect. For example, in Fig 4D the COLO320-HSR line is used to show that inhibition of Wnt with G007-LK phenocopies inhibition with ETC-159. However, this is the first time the cell line appears in the manuscript and the effect of ETC-159 on this line is not shown, making it difficult to compare with other experiments. -The authors state that the staining for BRCA1, FANCD2 and RAD51 is higher in the stem cells compared to the more proliferative cells higher in the crypt. The images supplied don't really show this however. RAD51 in particular looks to be expressed at a higher level in the upper crypt rather than the lower crypt. I would need to see better evidence to agree with the authors' claim. Lgr5 positive cells have been extensively transcriptomically characterized, so if they are transcriptionally regulated, as the previous data would suggest, then this will be available in the literature. These datasets will also allow the analysis of Mybl2 in the crypt base. Furthermore, it is now thought that the Lgr5 cells at the base of the crypt are a homogenous population, the majority of which are radiosensitive rather than radioresistant (DOI: 10.1038/s41598-020-64987-1). This needs to be acknowledged in the manuscript and the conclusions adjusted accordingly. -The data regarding the induction of senescence also needs improvement. Xenograft experiments in Fig 1 are used to show an increase in senescence, which is nicely demonstrated using various markers. The authors are missing a measure of proliferation within these tumours (Ki67, pHH3 or BrdU if they have it) in order to show a functional loss of growth potential. The authors further suggest that an accumulation of double strand breaks accelerates the senescence but provide no evidence to show this. Additionally, as above, it would be good to see similar effects in additional cell lines in order to gauge the universality of the finding. Indeed, senescence is an extremely rare event in the intestine, so it is likely that an alternative mechanism is at play in the data presented in Fig 7. This should be acknowledged (or shown otherwise) and the conclusion drawn should reflect this. -The evidence that Mybl2 is binding to the promoters is based on previously published data. It would strengthen the manuscript to that ChIPSeq data from the lines used here, potentially with/without Wnt-inhibition. *-Would additional experiments be essential to support the claims of the paper? Request additional experiments only where necessary for the paper as it is, and do not ask authors to open new lines of experimentation.* -The authors are not consistent with their use of cell lines, using different ones at different points in the manuscript. I have addressed elsewhere in this review. -The use of EGI-1, MCAS and CFPAC-1 cell lines is used to show that this is a general phenomenon. Considering that colon tissue is the focus of the latter part of the paper, it would make sense to include a colon cancer line here. Indeed, such a line is used (as mentioned elsewhere) but not included here. -The authors show altered HR and FA pathways after Apc deletion in the intestine (and in Wnt-high normal

physiology), but do not show that this leads to increased sensitivity to Parp inhibition. Treatment of animals would be needed to support the claim that this is functionally relevant. *- Are the suggested experiments realistic in terms of time and resources? It would help if you could add an estimated cost and time investment for substantial experiments.* -The additional experiments are realistic and would serve to significantly improve the reliability of the results. *- Are the data and the methods presented in such a way that they can be reproduced?* -Yes *- Are the experiments adequately replicated and statistical analysis adequate?* -Yes, with the caveats mentioned in this review. **Minor comments:** *- Specific experimental issues that are easily addressable.* - There is variability in group size that is strange. In Fig 1D for example, ETC159 and Olaparib groups have 10 animals each, whereas the combination has 7. What is the reason for this? -In Figure 1, 3 cell lines are used to show the generality of this finding. However, the figure only shows the images for the EGI-1 cell line. The others need to be included (in a supplementary figure perhaps) to prove the finding. -Along the same lines, the immunoblot shown in Fig 2E should be repeated on additional cell lines. -Fig 4D shows images of a colony formation assay. In the text, these images are quantified, but this quantification is not included in the figure. That should be added. *- Are prior studies referenced appropriately?* -A previous study from the Rudolph lab (DOI: 10.15252/emj.201490700), which was focused on DNA damage and Wnt signalling in the intestinal stem cell, is not in agreement with the results in this manuscript. That paper detailed a well carried out study and is the strongest data we have at present in this regard. The authors really should reference this paper, and comment on the disagreement between their findings. -A study by Dorsam et al. (DOI: 10.1073/pnas.1712345115) address the expression of Parp1 in intestinal cancer. This study needs to be cited and commented on. *- Are the text and figures clear and accurate?* -The figures are well presented, and the legends are thorough. The text is written well. *- Do you have suggestions that would help the authors improve the presentation of their data and conclusions?* -No

3. Significance:

Significance (Required)

-Describe the nature and significance of the advance (e.g. conceptual, technical, clinical) for the field. -The idea that Wnt-inhibition produces a synthetic lethal state with Parp inhibitors is an interesting one. However, the authors stretch the universality of this finding, and would need a lot more evidence to confirm that. There are several studies in the intestine in particular (some of which are referenced above) that indicate a far more complex picture than that presented here. While it is always tempting to think about the clinical significance of findings such as those presented here, I am not convinced that these will translate to the clinic. *-Place the work in the context of the existing literature (provide references, where appropriate).* -This has been done above. *-State what audience might be interested in and influenced by the reported findings.* - This will be of interest to the Wnt-signalling community. There is a lot of data already available that can be mined to assess the relevance of the findings presented here. *- Define your field of expertise with a few keywords to help the authors contextualize your point of view. Indicate if there are any parts of the paper that you do not have sufficient expertise to evaluate.* -Wnt, intestine, stem cell

Review #3

1. How much time do you estimate the authors will need to complete the suggested revisions:

Estimated time to Complete Revisions (Required)

(Decision Recommendation)

Between 1 and 3 months

2. Evidence, reproducibility and clarity:

Evidence, reproducibility and clarity (Required)

The authors found that Wnt/ β -catenin signal regulates the expression of genes involved in the homologous recombination (HR) and Fanconi anemia (FA) repair pathways in cancer cells in which Wnt signaling is activated, and showed that inhibition of Wnt signaling caused susceptibility to PARP inhibitors. This tumor-suppressive effect of combined Wnt and PARP inhibition is common in multiple cancer cell types and has been observed not only in Wnt signal-enhanced cancer cells sensitive to PORCN inhibitors but also in colorectal cancer cells insensitive to PORCN inhibitors. In these Wnt signal-dependent cancer cells, the β -catenin signal positively regulated the expression of MYBL2, which was involved in the expression of the HR/FA genes. The combination of Wnt signaling and PARP inhibitors induced a phenotype of cell growth inhibition and cell senescence. Furthermore, the authors demonstrated that the Wnt-MYBL2-HR/FA axis functions not only in cancer cells but also in Wnt-signal-dependent normal cells of intestinal crypts. **Major comments:** (1) In relation to Fig 1, phenotypic analysis with chemical inhibitors is always concerned with off-target effects. It is better to confirm a synergistic antitumor effect with a combination of at least two independent inhibitors. Synergistic antitumor effects should be confirmed using other compounds. (2) In relation to Fig 2E, Immunoblotting of BRCA1, BRCA2, FANCA and FANCD2 in HPAF-II tumors collected 8 and 56 hours after the start of ETC-159 treatment confirmed the down-regulation of these proteins. However, the reduced expression at 8 hours after treatment is unacceptable. The results should be accurately described. In addition, the BRCA1 blot is dirty and unappealing. They should be replaced. (3) It should be determined whether the reduction of HR and FA pathway genes by PORCN inhibitors is dependent on the reduction of Wnt/ β -catenin signaling rather than an indirect effect such as cell proliferation. For example, it is possible to test whether the decrease in gene expression by ETC-159 can be reversed by treatment with a GSK3 inhibitor for a short period. (4) The authors should show direct experimental evidence be provided whether the potentiation of synergistic antitumor effects in combination with PARP inhibitors is dependent on the impairment of the double-stranded DNA repair pathway by PORCN inhibitors. (5) The results for the synergistic antitumor effect in Fig 3 are impressive. It would be more informative that the authors show one more additional case of Wnt high cells resistant to PORCN inhibition other

than PaTu8988T cells. Further, as shown in Fig 1C and D, data from xenograft experiments would be more convincing. (6) For Fig 4B, the authors need to explain the reasons why there is no increased expression of HR/FA genes due to stabilized β -catenin expression compared to control cells. As a control experiment, the authors should show the expression of AXIN2 by stabilized β -catenin expression. (7) Related to Fig 4D, in addition to PORCN inhibitors in Wnt ligand-dependent cancer cells, it is interesting to note the synergistic antitumor effect of PARP inhibitors on G007-LK in the APC mutant colorectal cancer cell line COLO320-HSR cells. The expression of the AXIN2 and HR/FA pathway genes should be confirmed in COLO 320 HSR cells treated with G007-LK. (8) In the APC mutant colon cancer cell line COLO 320 HSR cells, it would be more persuasive if the combined effects of G007-LK and PARP inhibitor could be confirmed in a xenograft experiment. (9) The Wnt- β -catenin-MYBL2 pathway is an important point in the analysis of the mechanism in the current paper, but the data are somewhat superficial. The following experiments are absolutely required. Determine whether MYBL2 is a direct target gene for the β -catenin/TCF pathway. Does β -catenin knockdown reduce MYBL2 expression in HPAFII cells? Is there consensus binding sequence for TCF4 in the regulatory region of the MYBL2 gene? The binding of β -catenin/TCF4 to the regulatory element of the MYBL2 gene should be confirmed by ChIP analysis in HPAFII cells. (10) Is reduced expression of MYBL2 also observed with G007-LK treatment in COLO 320 HSR cells? (11) The authors confirmed the binding of MYBL2 to the promoters of HR/FA pathway genes using a public ChIP sequence data set. However, these data are from human chronic myeloid leukemia cells (K562) cells. The authors should confirm the binding of MYBL2 to the promoter region of the HR/FA pathway genes using Wnt-dependent cells used in this paper. (12) The public ChIP sequence data should also specify the location of the transcription start site (TTS), the location of the peak from TTS, and the presence or absence of the consensus motif for MYBL2. (13) The importance of MYBL2 as a downstream mediator of the Wnt/ β -catenin pathway is not clear. Is suppression of PORCN inhibitor-dependent HR/FA pathway genes expression rescued by MYBL2 expression? In addition, can MYBL2 expression rescue the synergistic antitumor effects by a combination of PORCN and PARP inhibitors? These are important data that can also be used as responses to review comments (4). (14) Knockdown of MYBL2 and the combination of knockdown of MYBL2 with PARP inhibitors should be tested for their ability to mimic the phenotype of PORCN inhibitor. (15) Referring to Fig 6B, there is no description concerning the results of the 21-day treatment. (16) It seems that there is no consistent trend for the expression of p 21 in Fig 6D. The results should be accurately described or replaced with consistent data. (17) It would be nice to show that cellular senescence occurs by combined MYBL2 KD and PARP inhibitor treatment. Can MYBL2 expression also rescue cell senescence induced by PORCN and PARP inhibitors treatment? These experiments strengthen the authors' conclusion that the regulation of HR/FA pathway by the Wnt/ β -catenin/MYBL2 axis is involved in cell senescence due to the combination therapy with PORCN and PARP inhibitors. (18) Enlarged pictures of the villi should be shown for comparison with crypts in Fig 7A. (19) The authors should show the expression of MYBL2 in normal intestine and intestinal tumor regions of APC min mouse immunohistochemically.

3. Significance:

Significance (Required)

The authors suggest a significant finding and mechanism for the synergistic antitumor effects of PORCN inhibitor in combination with PARP inhibitor. However, some of the authors' results concerning the molecular mechanisms show parallel phenomena and are obscure in terms of causality. Although there is novelty in the identification of Wnt/ β -catenin/MYBL2-HR/FA pathway in this study, the analysis on MYBL2 is superficial and not sufficient. The additional experiments that we requested are necessary to support the authors' conclusion.

27th August, 2020

Dear Editors

EMBO Molecular Medicine

We thank the reviewers for their favorable comments. We are pleased that they find our study interesting, well carried out and our data convincing. The reviewers also appreciate the utility of the studies for therapeutic applications and improved understanding of the Wnt signaling pathway.

A few of the key comments from the reviewers:

Reviewer 1: *Together this is a very original piece of work revealing a novel signaling pathway downstream of β -catenin activity that uncovers the molecular basis underlying radioresistance of tumour cells and stem cells. Importantly, this pathway also provides a rationale for combinatorial treatment with Wnt and PARP inhibitors for cancer treatment and possibly preventing the emergence of resistance.*

Reviewer 2: *This is an interesting and important finding, The key conclusions are largely convincing. In most cases, multiple experiments are carried out in order to arrive at a conclusion, generally this seems to be a well carried out study. This will be of interest to the Wnt-signalling community.*

Reviewer 3: *The authors suggest a significant finding and mechanism for the synergistic antitumor effects of PORCN inhibitor in combination with PARP inhibitor. There is novelty in the identification of Wnt/ β -catenin/MYBL2-HR/FA pathway in this study.*

Based on the reviewers' comments we have revised some sections of the manuscript and also included additional new data as requested. We propose to perform additional studies to further address other concerns of the reviewers. Please find the revised version of the manuscript and the response to reviewers' comments below.

Reviewer #1 (Evidence, reproducibility and clarity (Required)):

In this manuscript Kaur et al uncover the Wnt-bcatenin-MYBL2 signaling axis that appear to be important for the regulation of expression of several genes involved in DNA damage pathways. This pathway is not only active in cancer cells exhibiting high Wnt signaling activity but also in endogenous contexts where high Wnt signaling is required such as in tissue stem cell self-renewal. Given the shared properties of radio and chemo resistance of tumor cells and stem cells this pathway may constitute a general mechanism to protect cells from genomic insults.

The authors also exploit their findings as a therapeutic vulnerability by modulating Wnt-bcatenin signaling activity and creating a "BRCA-like state" that render the cells susceptible to treatment with the PARP inhibitor Olaparib. Considering that the stem cell environment is "protected" from therapeutic levels of Wnt inhibitors by drug efflux pumps, this combinatorial

treatment is proposed to selectively harm tumor cells and be more efficient than the individual treatments.

Together this is a very original piece of work revealing a novel signaling pathway downstream of B-catenin activity that uncovers the molecular basis underlying radioresistance of tumour cells and stem cells. Importantly, this pathway also provides a rationale for combinatorial treatment with Wnt and PARP inhibitors for cancer treatment and possibly preventing the emergence of resistance. Below, I list major and minor points that are meant as constructive criticisms of the study to potentially strengthen the findings.

****Major points:****

1. MYBL2 expression is found to be induced in Wnt-b-catenin activated cells and to be inhibited by PORCN inhibitors. Is MYBL2 a b-catenin target gene?

Yes. In response to the reviews we have now examined the expression of *MYBL2* before and after β -catenin knockdown. The data, now included in new figure 5E, shows that β -catenin siRNA knockdown reduced the expression of *MYBL2*. This data, taken together with the data from tumors with stabilized β -catenin treated with the PORCN inhibitor in Figure 5, all support the conclusion that *MYBL2* is regulated by β -catenin.

Further supporting this, in addition to using Wnt pathway inhibitors and manipulation of β -catenin stability, we have now analyzed three published RNAseq datasets of from mouse and human intestine. In the dataset from Scott Lowe's group of an intestinal organoid model with inducible activation of Wnt signaling, activation of Wnt signaling by loss of APC led to an increase in the expression of both *Mybl2* and DNA repair genes (Dow *et al*, 2015). In a study from the Abud lab (GSE149311) (Jardé *et al*, 2020) (online July 2020) intestinal epithelial cells were sorted into LGR5 (a Wnt target gene and stem cell marker) high, medium, low and negative populations and analyzed by RNAseq. Looking at their dataset, we find that the expression of both *Mybl2* and DNA HR/FA repair genes increases as LGR5 expression increases, also consistent with our model. Similarly in a single cell RNAseq dataset from human intestine (Wang *et al*, 2020) the Wnt-high stem and transit-amplifying cells expressed *MYBL2* and DNA HR/FA repair genes. These findings provide independent support for the Wnt/ β -catenin/*MYBL2*/DNA repair axis (new Figures S3A-C).

Can the authors perform CHIP with b-catenin or LEF/TCF on the MYBL2 promoter or identify LEF/TCF elements within the MYBL2 promoter.

As the reviewer notes, our data demonstrate that *MYBL2* expression is regulated by Wnt/ β -catenin signaling. However, as the reviewer (and also reviewer 3) notes, we did not establish here which sequence elements in the genome are responsible for this. It's somewhat problematic, as β -catenin regulation of genes is very often through distant enhancers rather than through traditional LEF/TCF binding sites in proximal promoters, c.f., (Nakamura *et al*, 2016; Guenther *et al*, 2014). For example, Guenther *et al.*, showed the important TCF4 binding site determining brown versus blond hair is 350 kb distant from the transcriptional start site of the relevant gene. In addition, β -catenin ChIP is significantly more challenging than standard transcription factor ChIP because β -catenin is not a direct DNA binding protein, and LEF/TCF occupancy alone does not predict regulation well (Nakamura *et al*, 2016).

So we pose this query to the editors: We can predict β -catenin response elements by bioinformatic analysis, and/or do Chip-PCR and promoter bashing **if the editors think this is**

important. However, a negative result would not alter our conclusion and a positive result would not add much to our story while taking a lot of work. A full-blown β -catenin CHIP-seq experiment is, we feel, beyond the scope of this manuscript.

2. *As stated by the authors, PARP inhibitors such as Olaparib, are effective only in the context of HR pathway mutations such as BRCA1. It is therefore surprising that HPAF cells (which presumably do not contain HR pathway mutation) appear to be quite sensitive to Olaparib treatment, both in vitro and in vivo. What is the explanation for this?*

As the reviewer rightly states, PARP inhibitors are effective in the context of loss-of-function BRCA1 and HR pathway mutations. We find that treatment with PORCN inhibitors reduces the expression of HR pathway genes including BRCA1 and BRCA2 (Figure 2). Indeed, this is one of the key findings of our manuscript. Wnt inhibition induces a BRCA-deficient state. This coordinate reduction in the expression of DNA repair genes upon Wnt inhibition makes the cells deficient in the HR pathway. Therefore, Wnt inhibition sensitizes cells to the PARP inhibitors. We note that HPAF-II cells before Wnt inhibition are not intrinsically sensitive to Olaparib as the IC₅₀ of Olaparib for *in vitro* analysis is 40 μ M (Table S1) and the dose for mouse studies is also high (50 mg/kg).

3. *The authors assess the reduction in HR upon treatment with Wnt inhibitors with the DR-GFP assay and state that the direct observation of DNA damage with quantification of foci was impossible. It is understood that the expression of several DNA repair genes normally present at foci is decreased but can a marker such as γ H2AX be used?*

We agree that γ H2AX foci formation is the most commonly used method for assessing DNA damage. As the reviewer rightly noted, we are unable to use this marker as the expression of γ H2AX was also reduced upon Wnt inhibition. In response to the reviewer's query, we propose to use alternate assays such as comet assays or the formation of 53BP1 foci to assess DNA damage.

4. *In figure 1C, presumably the decrease in tumor volume in ETC-159 and Olaparib is statistically significant? This should be indicated. Also, is the combination treatment statistically different from the individual treatments?*

As per the reviewer's suggestion, we have now included the p values to indicate the significance of the differences between various treatment groups.

5. *The findings that inhibiting tankyrase also synergized with Olaparib represent an important extension of the work given the more frequent mutations of distal Wnt pathway components. It would therefore be important to show that tankyrase treatment in APC mutant tumors also lead to decreased expression of DNA repair genes to validate the mechanism in this context.*

We agree with the reviewer that this would be an important extension of the study. We will analyze the response of DNA repair genes in the APC mutant colon cancer models treated with the tankyrase inhibitor. To do this, we are now collaborating with the group of Hector Palmer in Barcelona to assess the expression of DNA repair genes in tankyrase inhibitor-treated patient-derived xenografts with APC loss of function mutations (Arques *et al*, 2016).

****Minor Points:****

1. On line 147 the authors state that the genes were decreased by 8-10 folds. Figure 2D however does not show fold and is expressed as fraction of expression compared to no treatment.

As suggested by the reviewer we have modified the text to improve clarity.

2. In figure 1D, is there an explanation for the absence of difference in tumor weight in the ETC-159 condition? Is this due to differentiation of tumors by ETC-159 leading to mucus loading and hence heavier tumors?

In Figure 1D the mice were treated with a low dose of ETC-159 (10 mg/kg) so the tumor weight reduction was not as robust as when we use higher (30-75 mg/kg/day) doses.

3. Why the authors performed the DR-GFP assay in *Aspc-1* cells and not in *HPAF-II* in which most of the work was performed?

We agree that performing the DR-GFP assay in *HPAF-II* cells would have been ideal. DR-GFP assay requires transient transfection of the I-Sce I expressing plasmid to assess the HR rate. We were limited by the ability to transfect *HPAF-II* cells using transient transfection methods as the efficiency is less than 1%.

4. The legend for Figure 3 needs to be adjusted to correspond to the panels.

Thank you for pointing this out. We have adjusted the figure legend according to the panels.

Reviewer #1 (Significance (Required)):

This is a significant study revealing molecular understanding underlying the contribution of Wnt-bcatenin signaling to radioresistance of tumor cells and stem cells. The findings provide an interesting therapeutic opportunity for combinatorial treatment using Wnt and PARP inhibitors for treatment of cancers.

Reviewer #2 (Evidence, reproducibility and clarity (Required)):

****Summary:****

The manuscript by Kaur et al. provides evidence that Wnt-signalling drives a DNA repair phenotype. Using a combination of genetic and chemical inhibitors of the specific pathways they narrow this finding down to the HR and FA repair pathways. They further show that this seems to be mediated via a Wnt/b-catenin/Mybl2 axis. These findings are replicated in a number of Wnt-high cancer models, and in an in vivo setting in the intestinal crypt.

This is an interesting and important finding, but I have a number of significant comments, as detailed below.

****Major comments:****

- Are the key conclusions convincing?

-The key conclusions are largely convincing. In most cases, multiple experiments are carried out in order to arrive at a conclusion (with some caveats mentioned below). There are a few additional things I would like to see, but generally this seems to be a well carried out study. I have particular reservations regarding the senescence and intestinal data, as the authors do not show that the same mechanisms are at play.

- Should the authors qualify some of their claims as preliminary or speculative, or remove them altogether?

-My main issue with this manuscript is the inconsistent use of different cell lines. At various points HPAF-II, EGI-1, MCAS, CFPAC-1 and COLO320-HSR are used, without any explanation of why specific lines were used for different experiments. It has a feeling of cherry-picking, and to be truly convinced I would like to see consistency in this aspect. For example, in Fig 4D the COLO320-HSR line is used to show that inhibition of Wnt with G007-LK phenocopies inhibition with ETC-159. However, this is the first time the cell line appears in the manuscript and the effect of ETC-159 on this line is not shown, making it difficult to compare with other experiments.

We apologize if the rationale for selecting various cell lines was not made clear in the manuscript. We used RNF43 mutant HPAF-II cells for most of our initial discovery studies (in vivo and in vitro analysis) as these cells are Wnt-addicted due to loss-of-function RNF43 mutations. We used additional Wnt-addicted cell lines from different tissue of origin and with distinct driver mutations to test generality, i.e., to determine if the reduction of DNA repair gene expression upon Wnt inhibition and the combination of Wnt inhibitor with olaparib is a common phenomenon. For this purpose, we used a cholangiocarcinoma cell line EGI-1 and colorectal patient-derived xenografts (Wnt-addicted due to an R-spondin translocation), the ovarian and pancreatic cancer cell lines MCAS and AsPC-1 respectively (Wnt-addicted due to an RNF43 mutation), and the pancreatic cancer cell line CFPAC-1 (sensitive to Wnt inhibition, mechanism unknown). We used AsPC-1 cells for the DR-GFP assay because these cells are readily transfectable.

To test if other Wnt pathway inhibitor drugs had a similar effect as ETC-159, we used COLO320-HSR cells that have an APC loss-of-function mutation. Hence we used a tankyrase inhibitor, G007LK that prevents Wnt signaling by stabilizing AXIN. This showed that the synergy of Wnt pathway inhibitors with olaparib is a general phenomenon.

-The authors state that the staining for BRCA1, FANCD2 and RAD51 is higher in the stem cells compared to the more proliferative cells higher in the crypt. The images supplied don't really show this however. RAD51 in particular looks to be expressed at a higher level in the upper crypt rather than the lower crypt I would need to see better evidence to agree with the authors' claim. Lgr5 positive cells have been extensively transcriptomically characterized, so if they are transcriptionally regulated, as the previous data would suggest, then this will be available in the literature. These datasets will also allow the analysis of Mybl2 in the crypt base. Furthermore, it is now thought that the Lgr5 cells at the base of the crypt are a homogenous population, the majority of which are radiosensitive rather than radioresistant (DOI: 10.1038/s41598-020-64987-1). This needs to be acknowledged in the manuscript and the conclusions adjusted accordingly.

We apologize for our misstatement regarding the localization of *BRCA1*, *FANCD2*, and *RAD51* observed in IHC staining and we thank the reviewer for pointing this out. We have modified the text to state: “In agreement with the presence of a Wnt gradient in the intestine, we observed positive nuclear staining for BRCA1, FANCD2 and RAD51 throughout the crypts. Conversely, Wnt-low villus epithelial cells did not show any staining for these proteins (Figure 7A and S4)”.

Based on the reviewer’s suggestion, we also looked for published RNAseq datasets of intestinal stem cells. Two recent studies were very informative as noted in our response to reviewer 1 as well. A study from the Abud lab (GSE149311) (Jardé *et al*, 2020) online July 2020 performed RNAseq on sorted LGR5 high, mid, low and negative populations from intestinal epithelial cell pools. Looking into their dataset, we find that expression of both *Mybl2* and DNA repair genes increases as LGR5 expression increases, consistent with our model. We have now included this data in our new Figure S3A.

The immunofluorescence and sequencing studies show that the expression of HR/FA genes is high in mouse Wnt-high cells. Additionally, we have now examined single cell RNA seq data from the human intestine, also very recently published (Wang *et al*, 2020). There, we also see co-expression of HR and FA genes with *MYBL2* in both stem and transit-amplifying cell clusters (new Figure S3B). Taken together, this additional data further support the model that the Wnt/ β -catenin/*MYBL2* axis regulates the expression of HR and FA genes in the human and mouse intestine.

-The data regarding the induction of senescence also needs improvement. Xenograft experiments in Fig 1 are used to show an increase in senescence, which is nicely demonstrated using various markers. The authors are missing a measure of proliferation within these tumours (Ki67, pHH3 or BrdU if they have it) in order to show a functional loss of growth potential. The authors further suggest that an accumulation of double strand breaks accelerates the senescence but provide no evidence to show this. Additionally, as above, it would be good to see similar effects in additional cell lines in order to gauge the universality of the finding. Indeed, senescence is an extremely rare event in the intestine, so it is likely that an alternative mechanism is at play in the data presented in Fig 7. This should be acknowledged (or shown otherwise) and the conclusion drawn should reflect this.

As per the reviewers’ suggestion, we have now performed the analysis of multiple cell cycle genes. The expression of the cell cycle genes *MKI67*, *CDKN1*, *CCNE2*, and *AURKB* is reduced upon treatment with ETC-159 and further reduced in tumors treated with a combination of olaparib and ETC-159 (New Figure 6H).

We agree that we did not provide evidence for the accumulation of double-strand DNA breaks. As suggested by the reviewer, we will assess the accumulation of DNA damage e.g. using the comet assay or 53BP1 foci formation.

Question to the editor: To provide evidence of increased senescence in additional models, we would need to test additional xenograft models, since we only see robust senescence in tumors, not in cultured cells in plastic. We don’t feel that these experiments will add a lot and will require a lot of mice. But if the editors feel strongly, we can test this in an additional mouse model (e.g. AsPC-1 xenograft). This will take quite some time and many mice, as it will need at a minimum four different treatment groups regimes.

Our data suggest senescence is occurring in the tumors, not the intestine. We agree with the reviewer and will modify the manuscript as suggested.

-The evidence that Mybl2 is binding to the promoters is based on previously published data. It would strengthen the manuscript to that ChipSeq data from the lines used here, potentially with/without Wnt-inhibition.

As noted above, identifying specific TF binding sites for gene regulation is interesting but we feel this is beyond the scope of the current manuscript. **We can perform MYB12 CHIP as suggested by the reviewers if the editors think this is important.**

-Would additional experiments be essential to support the claims of the paper? Request additional experiments only where necessary for the paper as it is, and do not ask authors to open new lines of experimentation.

-The authors are not consistent with their use of cell lines, using different ones at different points in the manuscript. I have addressed elsewhere in this review.

We have provided an explanation in response to the first question raised by the reviewer.

-The use of EGI-1, MCAS and CFPAC-1 cell lines is used to show that this is a general phenomenon. Considering that colon tissue is the focus of the latter part of the paper, it would make sense to include a colon cancer line here. Indeed, such a line is used (as mentioned elsewhere) but not included here.

As mentioned above we have used several cell lines with loss-of-function mutations in RNF43 or R-spondin that are Wnt addicted and show synergistic growth inhibition with a combination of ETC-159 and olaparib treatment.

The colorectal cell line COLO320-HSR with an APC mutation is sensitive to tankyrase inhibitor G007LK, but not PORCN inhibitors, that act upstream of the mutation. We have therefore tested the combination of PARP inhibitor and G007LK in COLO320-HSR cells. Please see the data from the Lowe lab (new figure S3C) that also shows the regulation of Mybl2 and HR/FA genes in mouse organoids after APC knockdown.

-The authors show altered HR and FA pathways after Apc deletion in the intestine (and in Wnt-high normal physiology), but do not show that this leads to increased sensitivity to Parp inhibition. Treatment of animals would be needed to support the claim that this is functionally relevant.

We showed (Figure 7B) that APC mutation further increases HR/FA pathway activity in the intestine. This is consistent with the re-analysis of gene expression in organoids after APC knockdown (new Suppl Fig S3C) as well where knockdown of APC in the intestinal organoids enhances the expression of DNA repair genes. Therefore Wnt-high tumors (APC adenomas included) should be less, not more sensitive to DNA damage. Therefore, we do not think that additional mouse studies will be useful here.

- Are the suggested experiments realistic in terms of time and resources? It would help if you could add an estimated cost and time investment for substantial experiments.

-The additional experiments are realistic and would serve to significantly improve the reliability of the results.

- Are the data and the methods presented in such a way that they can be reproduced?

-Yes

- Are the experiments adequately replicated and statistical analysis adequate?

-Yes, with the caveats mentioned in this review.

****Minor comments:****

- Specific experimental issues that are easily addressable.

-There is variability in group size that is strange. In Fig 1D for example, ETC159 and Olaparib groups have 10 animals each, whereas the combination has 7. What is the reason for this?

Usually we start with a group of mice injected with cancer cells, and then assign them to treatment arms to get equal size tumors in each group. Sometimes tumors don't grow, sometimes mice kill each other, sometimes we may make mistakes in dosing. That is the reason for the variability in group size.

-In Figure 1, 3 cell lines are used to show the generality of this finding. However, the figure only shows the images for the EGI-1 cell line. The others need to be included (in a supplementary figure perhaps) to prove the finding.

As per the reviewer's suggestion we have included the images from additional cell lines new supplemental Figure 1A-B.

-Along the same lines, the immunoblot shown in Fig 2E should be repeated on additional cell lines.

As suggested by the reviewer we have now included the immunoblot for DNA repair genes in colorectal PDX to substantiate the gene expression analysis data (new Figure 2G).

-Fig 4D shows images of a colony formation assay. In the text, these images are quantified, but this quantification is not included in the figure. That should be added.

As per the reviewer's suggestion we have now included the graph showing the combination index values for the COLO320 HSR and Olaparib combination (new Figure 4E).

- Are prior studies referenced appropriately?

-A previous study from the Rudolph lab (DOI: 10.15252/embj.201490700), which was focused on DNA damage and Wnt signalling in the intestinal stem cell, is not in agreement with the results in this manuscript. That paper detailed a well carried out study and is the strongest data we have at present in this regard. The authors really should reference this paper, and comment on the disagreement between their findings.

-A study by Dorsam et al. (DOI: 10.1073/pnas.1712345115) addresses the expression of Parp1 in intestinal cancer. This study needs to be cited and commented on.

We will modify the manuscript to address these concerns.

- Are the text and figures clear and accurate?

-The figures are well presented, and the legends are thorough. The text is written well.

- Do you have suggestions that would help the authors improve the presentation of their data and conclusions?**

-No

Reviewer #2 (Significance (Required)):

-Describe the nature and significance of the advance (e.g. conceptual, technical, clinical) for the field.

-The idea that Wnt-inhibition produces a synthetic lethal state with Parp inhibitors is an interesting one. However, the authors stretch the universality of this finding, and would need a lot more evidence to confirm that. There are several studies in the intestine in particular (some of which are referenced above) that indicate a far more complex picture than that presented here. While it is always tempting to think about the clinical significance of findings such as those presented here, I am not convinced that these will translate to the clinic.

We appreciate the reviewer's comment that the combination of Wnt-inhibition with Parp inhibitors is interesting. Wnt signaling has long been associated with drug resistance, but the mechanism we describe here is novel and has both basic and potentially translational impact. Of course additional studies will be needed to test if this will translate to the clinic. Clinical testing is feasible, as olaparib is an FDA approved drug for treatment of BRCA deficient tumors and PORCN inhibitors have advanced to clinical trials, [ClinicalTrials.gov-NCT01351103](https://clinicaltrials.gov/ct2/show/study/NCT01351103), [NCT02278133](https://clinicaltrials.gov/ct2/show/study/NCT02278133), [NCT02521844](https://clinicaltrials.gov/ct2/show/study/NCT02521844), [NCT03447470](https://clinicaltrials.gov/ct2/show/study/NCT03447470), [NCT02675946](https://clinicaltrials.gov/ct2/show/study/NCT02675946), and [NCT03507998](https://clinicaltrials.gov/ct2/show/study/NCT03507998).

-Place the work in the context of the existing literature (provide references, where appropriate).

-This has been done above.

-State what audience might be interested in and influenced by the reported findings.

-This will be of interest to the Wnt-signalling community. There is a lot of data already available that can be mined to assess the relevance of the findings presented here.

-Define your field of expertise with a few keywords to help the authors contextualize your point of view. Indicate if there are any parts of the paper that you do not have sufficient expertise to evaluate.

-Wnt, intestine, stem cell

Reviewer #3 (Evidence, reproducibility and clarity (Required)):

The authors found that Wnt/ β -catenin signal regulates the expression of genes involved in the homologous recombination (HR) and Fanconi anemia (FA) repair pathways in cancer cells in which Wnt signaling is activated, and showed that inhibition of Wnt signaling caused susceptibility to PARP inhibitors. This tumor-suppressive effect of combined Wnt and PARP inhibition is common in multiple cancer cell types and has been observed not only in Wnt signal-enhanced cancer cells sensitive to PORCN inhibitors but also in colorectal cancer cells insensitive to PORCN inhibitors. In these Wnt signal-dependent cancer cells, the β -catenin signal positively regulated the expression of MYBL2, which was involved in the expression of the HR/FA genes. The combination of Wnt signaling and PARP inhibitors induced a phenotype of cell growth inhibition and cell senescence. Furthermore, the authors demonstrated that the Wnt-MYBL2-HR/FA axis functions not only in cancer cells but also in Wnt-signal-dependent normal cells of intestinal crypts.

****Major comments:****

(1) In relation to Fig 1, phenotypic analysis with chemical inhibitors is always concerned with off-target effects. It is better to confirm a synergistic antitumor effect with a combination of at least two independent inhibitors. Synergistic antitumor effects should be confirmed using other compounds.

We would like to draw the attention of the reviewer to Figure 4 where we have used a different Wnt pathway inhibitor, G007LK, in combination with Olaparib. G007LK has a different mechanism of action than ETC-159. It stabilizes AXIN, hence increasing β -catenin degradation. G007LK and ETC-159 therefore function to inhibit the Wnt pathway by independent mechanisms and we show that both drugs synergize with Olaparib to prevent growth of Wnt high cancers.

(2) In relation to Fig 2E, Immunoblotting of BRCA1, BRCA2, FANCA and FANCD2 in HPAF-II tumors collected 8 and 56 hours after the start of ETC-159 treatment confirmed the down-regulation of these proteins. However, the reduced expression at 8 hours after treatment is unacceptable. The results should be accurately described. In addition, the BRCA1 blot is dirty and unappealing. They should be replaced.

We apologize for any confusion in our description of the immunoblot. We have modified the text to improve clarity as suggested by the reviewer. We are sorry the reviewer finds our blot unappealing, but that is what real data from mouse tumors looks like sometimes. We can repeat the BRCA1 blot in Fig 2E as suggested **if the editors think that repeating the blot would alter the conclusions.**

(3) It should be determined whether the reduction of HR and FA pathway genes by PORCN inhibitors is dependent on the reduction of Wnt/ β -catenin signaling rather than an indirect effect such as cell proliferation. For example, it is possible to test whether the decrease in gene expression by ETC-159 can be reversed by treatment with a GSK3 inhibitor for a short period.

Our data show clearly that the changes in HR/FA genes are dependent on changes in β -catenin signaling (Figure 4A-B). How much of this change in gene expression is contributed to by changes in cell proliferation is an important question. To address this concern, we would like to draw the reviewers' attention to Suppl Figures S1D and E. We show that while ETC-

159 treatment of HPAF-II cells in vitro for 48 h did not significantly reduce the percentage of cells in S-phase, there already was a significant reduction in the expression of HR-pathway and FA genes.

Additionally, in the DR-GFP assays in AsPC-1 cells (Figure 2I), treatment with ETC-159 for 24 h led to a ~40% reduction HR-activity upon Wnt inhibition. However, in newly added data (Figure S1G), we show that the changes in the S phase cells in ETC-159 treated cells after 48 h treatment were not significant, Taken together these data show that the reduction of HR and FA pathway genes by PORCN inhibitors is not due to an effect on cell proliferation.

(4) The authors should show direct experimental evidence be provided whether the potentiation of synergistic antitumor effects in combination with PARP inhibitors is dependent on the impairment of the double-stranded DNA repair pathway by PORCN inhibitors.

We are not sure what experiment the reviewer is asking for here. We showed that stabilized β -catenin reverses the synergistic antitumor effect (Figure 4C). β -catenin signaling regulates many genes in the HR and FA pathways, so re-expression of a single gene (e.g. FANCD2 or BRCA1) will not reverse the synergy. Even MYBL2 is not the only way β -catenin works, as the knockdown of MYBL2 gave half the effect that ETC-159 did (Figure 5E). While re-expressing MYBL2 may be an attractive idea initially, however, the outcome of that experiment is very uncertain, since we showed (Figure 5F) that MYBL2 is only part of the story of how β -catenin regulates the many HR and FA genes. Hence, the re-expression of MYBL2 will at best only partially rescue the sensitivity to PARP inhibitors.

(5) The results for the synergistic antitumor effect in Fig 3 are impressive. It would be more informative that the authors show one more additional case of Wnt high cells resistant to PORCN inhibition other than PaTu8988T cells. Further, as shown in Fig 1C and D, data from xenograft experiments would be more convincing.

Per the reviewer's suggestion we have tested an additional Wnt high cell line PL-45 that is resistant to ETC-159 and now show that co-treatment with olaparib sensitizes these cells (new Figure 3E). Regarding a xenograft model, we agree it would be nice to have, **but we request the editor weigh in on if this is an essential experiment.**

(6) For Fig 4B, the authors need to explain the reasons why there is no increased expression of HR/FA genes due to stabilized β -catenin expression compared to control cells. As a control experiment, the authors should show the expression of AXIN2 by stabilized β -catenin expression.

As per the reviewer's suggestion we have included the data showing the expression of AXIN2 in cells with stabilized β -catenin (modified Figure 4B). We agree with the reviewer that it is interesting that the basal levels of the DNA repair genes do not change in cells with stabilized β -catenin.

To address the reviewer's question on why expression of stabilized β -catenin did not increase the expression of HR/FA genes, we went back to our comprehensive analysis of gene expression in response to Wnt inhibition (Figure S1C) (Madan *et al*, 2018). We find that there is a class of Wnt/ β -catenin target genes whose expression is robustly upregulated by stabilized β -catenin. These include AXIN2, NKD1 and RNF43. However, there is a second class of Wnt/ β -catenin target genes including MYC, CCND1 and MYBL2 whose expression is NOT increased by stabilized β -catenin. We speculate that expression of this latter class of genes is dependent

on multiple upstream factors and that an increase of β -catenin alone is not sufficient to alter their expression.

(7) *Related to Fig 4D, in addition to PORCN inhibitors in Wnt ligand-dependent cancer cells, it is interesting to note the synergistic antitumor effect of PARP inhibitors on G007-LK in the APC mutant colorectal cancer cell line COLO320-HSR cells. The expression of the AXIN2 and HR/FA pathway genes should be confirmed in COLO 320 HSR cells treated with G007-LK.*

We agree with the reviewer's suggestions and we will perform the requested analysis.

(8) *In the APC mutant colon cancer cell line COLO 320 HSR cells, it would be more persuasive if the combined effects of G007-LK and PARP inhibitor could be confirmed in a xenograft experiment.*

We have already shown synergy with the PORCN inhibitor in multiple models, and in addition used the tankyrase inhibitor as a confirmation that it is not a PORCN inhibitor-specific effect. In addition, we will analyze tankyrase inhibitor-treated colorectal cancer for regulation of HR/FA genes as noted above (Reviewer 1, question 5). We think an additional xenograft synergy model is time and mouse intensive and will not substantially add to the core conclusion that Wnt signaling regulates BRCA and FA genes. **We seek the editor's judgement here on the question - nice to have but not essential, versus must have.**

(9) *The Wnt- β -catenin-MYBL2 pathway is an important point in the analysis of the mechanism in the current paper, but the data are somewhat superficial. The following experiments are absolutely required. Determine whether MYBL2 is a direct target gene for the β -catenin/TCF pathway. Does β -catenin knockdown reduce MYBL2 expression in HPAF II cells?*

As per the reviewers' suggestion we have tested and now show that *MYBL2* is a direct target of β -catenin and its levels are reduced in cells where β -catenin is knocked down by siRNA (new Figure 5E). We have also added new data from published datasets showing a) APC knockdown in organoids increases *Mybl2* (new figure S3C) and b) *Mybl2* expression positively correlates with *Axin2* and *Lgr5* in the mouse intestine (new Figures S3A-B). Taken together, these data support the existence of a Wnt/ β -catenin/*Mybl2* pathway.

Is there a consensus binding sequence for TCF4 in the regulatory region of the MYBL2 gene? The binding of β -catenin/TCF4 to the regulatory element of the MYBL2 gene should be confirmed by ChIP analysis in HPAF II cells.

As the reviewer notes and similar to our response to Reviewer 1, our data demonstrate that *MYBL2* expression is regulated by Wnt/ β -catenin signaling. However, as the reviewer also notes, we did not establish here which sequence elements in the genome are responsible for this. It's somewhat problematic, as β -catenin regulation of genes is very often through distant enhancers rather than through traditional LEF/TCF binding sites in proximal promoters. (Nakamura *et al*, 2016; Guenther *et al*, 2014). For example, Guenther *et al.*, showed the important TCF4 binding site determining brown versus blond hair is 350 kb distant from the transcriptional start site of the relevant gene. In addition, β -catenin ChIP is more difficult than standard ChIP because β -catenin is not a DNA binding protein.

So we pose this **question to the editors**: We can certainly identify potential TCF response elements by bioinformatic analysis, and/or do Chip-PCR and promoter bashing **if the editors think this is important**. However, a negative result would not alter our conclusion and a

positive result would not add much. A full-blown β -catenin ChIP-seq experiment is, we feel, beyond the scope of this manuscript.

(10) Is reduced expression of MYBL2 also observed with G007-LK treatment in COLO320 HSR cells?

As suggested by the reviewer we will analyze the expression of MYBL2 in G007LK treated COLO320 cells.

(11) The authors confirmed the binding of MYBL2 to the promoters of HR/FA pathway genes using a public ChIP sequence data set. However, these data are from human chronic myeloid leukemia cells (K562) cells. The authors should confirm the binding of MYBL2 to the promoter region of the HR/FA pathway genes using Wnt-dependent cells used in this paper.

As noted above, identifying specific TF binding sites for gene regulation is interesting but laborious and we feel is beyond the scope of the manuscript. We note that MYBL2 has recently been reported to regulate double-strand break repair, supporting our model (Bayley *et al*, 2018). We can perform MYBL2 CHIP as suggested by the reviewers **if the editors think this is important for this manuscript.**

(12) The public ChIP sequence data should also specify the location of the transcription start site (TTS), the location of the peak from TTS, and the presence or absence of the consensus motif for MYBL2.

Based on the reviewers' suggestion, we have modified the figure to add the location of the MYBL2 peaks with respect to the TSS. We will analyze the presence and absence of MYBL2 consensus motif and include the information

(13) The importance of MYBL2 as a downstream mediator of the Wnt/ β -catenin pathway is not clear. Is suppression of PORCN inhibitor-dependent HR/FA pathway genes expression rescued by MYBL2 expression? In addition, can MYBL2 expression rescue the synergistic antitumor effects by a combination of PORCN and PARP inhibitors? These are important data that can also be used as responses to review comments (4).

We appreciate the great interest the reviewer has in further characterization of MYBL2. It is important to point out that suppression of Wnt/ β -catenin signaling leads to changes in many thousands of genes including many transcription factors (Madan *et al*, 2018). We found that MYBL2 is one of the TFs that regulates HR/FA genes, but we do not claim that it is the only TF regulating HR/FA genes. In Fig 5F, MYBL2 knockdown was ~50% as effective as ETC-159 treatment. Other transcription factors likely make up the other 50% effect. We already showed that introduction of stabilized β -catenin rescued the MYBL2 expression and the synergistic antitumor effect (Figure 4B,C and 5D). **Hence, we are unenthusiastic about doing the MYBL2 rescue experiment suggested and we seek the editor's input.**

(14) Knockdown of MYBL2 and the combination of knockdown of MYBL2 with PARP inhibitors should be tested for their ability to mimic the phenotype of PORCN inhibitors.

As noted above, MYBL2 is part of the story but not the complete story for regulation of HR/FA genes. In addition, haploinsufficiency of Mybl2 in mice has been shown by others to

impair dsDNA break repair ((Bayley *et al*, 2018). **We seek the editor's guidance**, as to do the requested MYBL2 knockout experiment would require long-term assays that we don't think would add significantly to our conclusions.

(15) Referring to Fig 6B, there is no description concerning the results of the 21-day treatment.

We thank the reviewer for pointing this out. We have now included the information in the Figure legend that the tumors used for analysis of cleaved caspase and PARP (Figure 6B) are from the study shown in Figure 1C-D.

(16) It seems that there is no consistent trend for the expression of p21 in Fig 6D. The results should be accurately described or replaced with consistent data.

We agree with the reviewer and we have now repeated the western blot. Please see the revised Figure 6D.

(17) It would be nice to show that cellular senescence occurs by combined MYBL2 KD and PARP inhibitor treatment. Can MYBL2 expression also rescue cell senescence induced by PORCN and PARP inhibitors treatment? These experiments strengthen the authors' conclusion that the regulation of HR/FA pathway by the Wnt/ β -catenin/MYBL2 axis is involved in cell senescence due to the combination therapy with PORCN and PARP inhibitors.

Again, we appreciate the great interest the reviewer has in further characterization of MYBL2. In several models MYBL2 has been shown to inhibit senescence (Masselink *et al*, 2001; Martinez *et al*, 2011; Mowla *et al*, 2014). It is plausible that the senescence induction upon ETC-159 treatment is also mediated via MYBL2 inhibition.

Our study does not claim that targeting MYBL2 in combination with PARP inhibitors is a treatment option. Rather, we are showing that the combination of Wnt and PARP inhibitors regulates the expression of DNA repair genes. MYBL2 is one of the downstream regulators of the Wnt/ β -catenin pathway.

(18) Enlarged pictures of the villi should be shown for comparison with crypts in Fig 7A.

We thank the reviewer for the suggestion and as suggested we have included the enlarged images of the villi and additional images to show the difference in expression of DNA repair genes in crypts vs. villi (new Figure S4).

(19) The authors should show the expression of MYBL2 in normal intestine and intestinal tumor regions of APC min mouse immunohistochemically.

As per our response to the reviewers' comment #9, above we have tested and now show that MYBL2 is a direct target of β -catenin and its levels are reduced in cells where β -catenin is knocked down using a siRNA (new Figure 5E). As noted above, we have also added new data from published datasets showing a) APC knockdown in intestinal organoids increases *Mybl2* (new Figure S3C). and b) *Mybl2* expression positively correlates with *Lgr5* expression. Taken together, these data support the existence of a Wnt/ β -catenin/*Mybl2* pathway. It's not clear to us that MYBL2 IHC would add much (except a nice visual) to the existing data.

Reviewer #3 (Significance (Required)):

The authors suggest a significant finding and mechanism for the synergistic antitumor effects of PORCN inhibitor in combination with PARP inhibitor. However, some of the authors' results concerning the molecular mechanisms show parallel phenomena and are obscure in terms of causality. Although there is novelty in the identification of Wnt/ β -catenin/MYBL2-HR/FA pathway in this study, the analysis on MYBL2 is superficial and not sufficient. The additional experiments that we requested are necessary to support the authors' conclusion.

REFERENCES:

- Arques O, Chicote I, Puig I, Tenbaum SP, Argiles G, Dienstmann R, Fernandez N, Caratu G, Matito J, Silberschmidt D, Rodon J, Landolfi S, Prat A, Espin E, Charco R, Nuciforo P, Vivancos A, Shao W, Tabernero J & Palmer HG (2016) Tankyrase Inhibition Blocks Wnt/ - Catenin Pathway and Reverts Resistance to PI3K and AKT Inhibitors in the Treatment of Colorectal Cancer. *Clinical Cancer Research* **22**: 644–656
- Bayley R, Blakemore D, Cancian L, Dumon S, Volpe G, Ward C, Almaghrabi R, Gujar J, Reeve N, Raghavan M, Higgs MR, Stewart GS, Petermann E & García P (2018) MYBL2 Supports DNA Double Strand Break Repair in Hematopoietic Stem Cells. *Cancer Res.* **78**: 5767–5779
- Dow LE, O'Rourke KP, Simon J, Tschaharganeh DF, van Es JH, Clevers H & Lowe SW (2015) Apc Restoration Promotes Cellular Differentiation and Reestablishes Crypt Homeostasis in Colorectal Cancer. *Cell* **161**: 1539–1552
- Guenther CA, Tasic B, Luo L, Bedell MA & Kingsley DM (2014) A molecular basis for classic blond hair color in Europeans. *Nat Genet* **46**: 748–752
- Jardé T, Chan WH, Rossello FJ, Kaur Kahlon T, Theocharous M, Kurian Arackal T, Flores T, Giraud M, Richards E, Chan E, Kerr G, Engel RM, Prasko M, Donoghue JF, Abe S-I, Pheffe TJ, Nefzger CM, McMurrick PJ, Powell DR, Daly RJ, et al (2020) Mesenchymal Niche-Derived Neuregulin-1 Drives Intestinal Stem Cell Proliferation and Regeneration of Damaged Epithelium. *Cell Stem Cell*
- Madan B, Harmston N, Nallan G, Montoya A, Faull P, Petretto E & Virshup DM (2018) Temporal dynamics of Wnt-dependent transcriptome reveal an oncogenic Wnt/MYC/ribosome axis. *J. Clin. Invest.* **128**: 5620–5633
- Martinez I, Cazalla D, Almstead LL, Steitz JA & DiMaio D (2011) miR-29 and miR-30 regulate B-Myb expression during cellular senescence. *Proc. Natl. Acad. Sci. U.S.A.* **108**: 522–527
- Masselink H, Vastenhouw N & Bernards R (2001) B-myb rescues ras-induced premature senescence, which requires its transactivation domain. *Cancer Letters* **171**: 87–101
- Mowla SN, Lam EW-F & Jat PS (2014) Cellular senescence and aging: the role of B-MYB. *Aging Cell* **13**: 773–779

Nakamura Y, de Paiva Alves E, Veenstra GJC & Hoppler S (2016) Tissue- and stage-specific Wnt target gene expression is controlled subsequent to β -catenin recruitment to cis-regulatory modules. *Development* **143**: 1914–1925

Wang Y, Song W, Wang J, Wang T, Xiong X, Qi Z, Fu W, Yang X & Chen Y-G (2020) Single-cell transcriptome analysis reveals differential nutrient absorption functions in human intestine. *J Exp Med* **217**: 357

31st Aug 2020

Dear Dr. Madan,

Thank you for the submission of your research manuscript to our editorial offices. I have now had the opportunity to read your manuscript, as well as the referees' reports and your rebuttal letter, and to discuss them with the other members of our editorial team.

The reviewers find that the question addressed by the study is of potential interest, but however remain unconvinced that some of the major conclusions are sufficiently supported by the data. In your rebuttal letter, you indicated how you have/will address these concerns, and after discussion, we further clarified the following points:

- MYBL2 ChIP experiments should be performed from relevant cell lines (referees #1, #2 and #3)
- The BRCA1 blot from Fig. 2E should be repeated (ref #3, point 2)
- Combination of PORCN inhibitor and olaparib should be tested in vivo (ref #3, point 5)

However,

- We do not ask to provide evidence of increased senescence in additional xenograft models (ref #2)
- We do not ask to test the combination of G007-LK and PARP inhibitor in vivo (ref #3, point 8)
- Regarding the MYBL2 rescue experiments, we would like you to perform the in vitro experiments, but no in vivo experiment will be required (ref #3, points 13/14)

If you feel you can satisfactorily address these points and those listed in your rebuttal letter, you may wish to submit a revised version of your manuscript. Please attach a covering letter giving details of the way in which you have handled each of the points raised by the referees. A revised manuscript will once again be subject to review and we cannot guarantee at this stage that the eventual outcome will be favorable.

When submitting your revised manuscript, please carefully review the instructions that follow below. Failure to include requested items will delay the evaluation of your revision:

- 1) A .docx formatted version of the manuscript text (including legends for main figures, EV figures and tables). Please make sure that the changes are highlighted to be clearly visible.
- 2) Individual production quality figure files as .eps, .tif, .jpg (one file per figure).
- 3) A .docx formatted letter INCLUDING the reviewers' reports and your detailed point-by-point responses to their comments. As part of the EMBO Press transparent editorial process, the point-by-point response is part of the Review Process File (RPF), which will be published alongside your paper.
- 4) A complete author checklist, which you can download from our author guidelines

(<https://www.embopress.org/page/journal/17574684/authorguide#submissionofrevisions>). Please insert information in the checklist that is also reflected in the manuscript. The completed author checklist will also be part of the RPF.

5) Before submitting your revision, primary datasets produced in this study need to be deposited in an appropriate public database (see <https://www.embopress.org/page/journal/17574684/authorguide#dataavailability>). Please remember to provide a reviewer password if the datasets are not yet public. The accession numbers and database should be listed in a formal "Data Availability " section (placed after Materials & Method). Please note that the Data Availability Section is restricted to new primary data that are part of this study.

6) We would also encourage you to include the source data for figure panels that show essential data. Numerical data should be provided as individual .xls or .csv files (including a tab describing the data). For blots or microscopy, uncropped images should be submitted (using a zip archive if multiple images need to be supplied for one panel). Additional information on source data and instruction on how to label the files are available at .

7) Our journal encourages inclusion of *data citations in the reference list* to directly cite datasets that were re-used and obtained from public databases. Data citations in the article text are distinct from normal bibliographical citations and should directly link to the database records from which the data can be accessed. In the main text, data citations are formatted as follows: "Data ref: Smith et al, 2001" or "Data ref: NCBI Sequence Read Archive PRJNA342805, 2017". In the Reference list, data citations must be labeled with "[DATASET]". A data reference must provide the database name, accession number/identifiers and a resolvable link to the landing page from which the data can be accessed at the end of the reference. Further instructions are available at .

8) We replaced Supplementary Information with Expanded View (EV) Figures and Tables that are collapsible/expandable online. A maximum of 5 EV Figures can be typeset. EV Figures should be cited as 'Figure EV1, Figure EV2" etc... in the text and their respective legends should be included in the main text after the legends of regular figures.

- Additional Tables/Datasets should be labeled and referred to as Table EV1, Dataset EV1, etc. Legends have to be provided in a separate tab in case of .xls files. Alternatively, the legend can be supplied as a separate text file (README) and zipped together with the Table/Dataset file. See detailed instructions here:

9) The paper explained: EMBO Molecular Medicine articles are accompanied by a summary of the articles to emphasize the major findings in the paper and their medical implications for the non-specialist reader. Please provide a draft summary of your article highlighting

- the medical issue you are addressing,

- the results obtained and
- their clinical impact.

10) For more information: There is space at the end of each article to list relevant web links for further consultation by our readers. Could you identify some relevant ones and provide such information as well? Some examples are patient associations, relevant databases, OMIM/proteins/genes links, author's websites, etc...

11) Every published paper now includes a 'Synopsis' to further enhance discoverability. Synopses are displayed on the journal webpage and are freely accessible to all readers. They include a short stand first (maximum of 300 characters, including space) as well as 2-5 one-sentences bullet points that summarizes the paper. Please write the bullet points to summarize the key NEW findings. They should be designed to be complementary to the abstract - i.e. not repeat the same text. We encourage inclusion of key acronyms and quantitative information (maximum of 30 words / bullet point). Please use the passive voice. Please attach these in a separate file or send them by email, we will incorporate them accordingly.

Please also suggest a striking image or visual abstract to illustrate your article. If you do please provide a png file 550 px-wide x 400-px high.

12) As part of the EMBO Publications transparent editorial process initiative (see our Editorial at <http://embomolmed.embopress.org/content/2/9/329>), EMBO Molecular Medicine will publish online a Review Process File (RPF) to accompany accepted manuscripts.

In the event of acceptance, this file will be published in conjunction with your paper and will include the anonymous referee reports, your point-by-point response and all pertinent correspondence relating to the manuscript. Let us know whether you agree with the publication of the RPF and as here, if you want to remove or not any figures from it prior to publication.

I look forward to receiving your revised manuscript.

Yours sincerely,

Lise Roth

Lise Roth, PhD
Editor

30 November, 2020

EMBO Molecular Medicine

Dear Dr. Roth,

We thank the reviewers for their favorable comments and the editors for providing guidance on which experiments were considered essential.

In your response to our letter, you further clarified the following points:

- *MYBL2 ChIP experiments should be performed from relevant cell lines (referees #1, #2 and #3)*

As noted below, these studies have been performed, new figures 5G and EV2E.

- *The BRCA1 blot from Fig. 2E should be repeated (ref #3, point 2)*

As noted below we have now repeated this blot, new figure EV1D.

- *Combination of PORCN inhibitor and olaparib should be tested in vivo (ref #3, point 5)*

As noted below we have now performed this study, new figure 3E.

However,

- *We do not ask to provide evidence of increased senescence in additional xenograft models (ref #2)*

- *We do not ask to test the combination of G007-LK and PARP inhibitor in vivo (ref #3, point 8)*

- *Regarding the MYBL2 rescue experiments, we would like you to perform the in vitro experiments, but no in vivo experiment will be required (ref #3, points 13/14)*

We were pleased that the reviewers find our study interesting, well carried out and our data convincing. The reviewers also appreciated the utility of the studies for both therapeutic applications as well as improved understanding of the Wnt signaling pathway.

All told, in response to the reviews, we have now added more than ten new biological experiments and added analysis of three public datasets, resulting in the addition of nineteen new panels in main and supplemental figures.

A few of the key comments from the reviewers:

Reviewer 1: *Together this is a very original piece of work revealing a novel signaling pathway downstream of β -catenin activity that uncovers the molecular basis underlying radioresistance of tumour cells and stem cells. Importantly, this pathway also provides a rationale for combinatorial treatment with Wnt and PARP inhibitors for cancer treatment and possibly preventing the emergence of resistance. This is a significant study revealing molecular understanding underlying the contribution of Wnt-bcatenin signaling to radioresistance of tumor cells and stem cells. The findings provide an interesting therapeutic opportunity for combinatorial treatment using Wnt and PARP inhibitors for treatment of cancers.*

Reviewer 2: *This is an interesting and important finding, The key conclusions are largely convincing. In most cases, multiple experiments are carried out in order to arrive at a*

conclusion, generally this seems to be a well carried out study. This will be of interest to the Wnt-signalling community.

Reviewer 3: *The authors suggest a significant finding and mechanism for the synergistic antitumor effects of PORCN inhibitor in combination with PARP inhibitor. There is novelty in the identification of Wnt/ β -catenin/MYBL2-HR/FA pathway in this study.*

Based on the reviewers' comments and editors' guidance we have revised several sections of the manuscript and also included additional new data as requested. Please find the revised version of the manuscript and the response to reviewers' comments below.

Reviewer #1 (Evidence, reproducibility and clarity (Required)):

****Major points:****

1. *MYBL2 expression is found to be induced in Wnt-b-catenin activated cells and to be inhibited by PORCN inhibitors. Is MYBL2 a b-catenin target gene?*

Yes. In response to the reviews we have now examined the expression of MYBL2 before and after β -catenin knockdown. The data, now included in new figure 5E, shows that knockdown of β -catenin with siRNA reduced the expression of MYBL2. These data, taken together with the data from tumors with stabilized β -catenin treated with the PORCN inhibitor in Figure 5, support the conclusion that MYBL2 is regulated by β -catenin.

Further supporting this, in addition to using Wnt pathway inhibitors and manipulation of β -catenin stability, we have now analyzed three published RNAseq datasets of from mouse and human intestine. In the dataset from Scott Lowe's group of an intestinal organoid model with inducible activation of Wnt signaling, knockout of APC led to an increase in the expression of both Mybl2 and DNA repair genes (Dow et al, 2015). In a study from the Abud lab (GSE149311) (Jardé et al, 2020) (online July 2020) intestinal epithelial cells were sorted into LGR5 (a Wnt target gene and stem cell marker) high, medium, low and negative populations and analyzed by RNAseq. Looking at their dataset, we find that the expression of both Mybl2 and DNA HR/FA repair genes increases as LGR5 expression increases, also consistent with our model. Similarly in a single cell RNAseq dataset from human intestine (Wang et al, 2020) the Wnt-high stem and transit-amplifying cells expressed both MYBL2 and DNA HR/FA repair genes. These findings provide independent support for a Wnt/ β -catenin/MYBL2/DNA repair axis (new Figures 7B and EV5A).

-Can the authors perform CHIP with b-catenin or LEF/TCF on the MYBL2 promoter or identify LEF/TCF elements within the MYBL2 promoter.

As the reviewer notes, our data demonstrate that MYBL2 expression is regulated by Wnt/ β -catenin signaling. However, as the reviewer (and also reviewer 3) notes, we did not establish here which sequence elements in the genome are responsible for this. It's somewhat problematic, as β -catenin regulation of genes is very often through distant enhancers rather than through traditional LEF/TCF binding sites in the proximal promoters, c.f., (Nakamura *et al*, 2016; Guenther *et al*, 2014). For example, Guenther et al., showed the important TCF7L2 binding site determining brown versus blond hair is 350 kb distant from the transcriptional start site of the relevant gene. Similarly, when we examined the TCF7L2 ChIPseq data in HCT116 and Panc1 cells (ENCODE Project Consortium, 2012), we found the TCF7L2 binding ~10 Kb upstream of the transcription start site of the MYBL2 gene where it overlapped with the traces for H3K27 acetylation (see figure below). In addition, β -catenin ChIP is significantly more challenging than standard transcription factor ChIP because β -

catenin is not a direct DNA binding protein (Nakamura *et al*, 2016). Therefore, while we did do ChIP with anti-MYBL2 (see below), we did not perform ChIP with TCF or β -catenin.

2. As stated by the authors, PARP inhibitors such as Olaparib, are effective only in the context of HR pathway mutations such as BRCA1. It is therefore surprising that HPAF cells (which presumably do not contain HR pathway mutation) appear to be quite sensitive to Olaparib treatment, both *in vitro* and *in vivo*. What is the explanation for this?

As the reviewer rightly notes, PARP inhibitors are effective in the context of loss-of-function BRCA1 and HR pathway mutations. We note that HPAF-II cells before Wnt inhibition are not intrinsically sensitive to Olaparib as the IC_{50} of Olaparib for *in vitro* analysis is 40 μ M (Table EV1) and the dose for mouse studies is also high (50 mg/kg). We find that treatment with PORCN inhibitors reduces the expression of HR pathway genes including BRCA1 and BRCA2 (Figure 2). Indeed, this is one of the key findings of our manuscript. Wnt inhibition induces a BRCA-deficient state. This coordinate reduction in the expression of DNA repair genes upon Wnt inhibition makes the cells deficient in the HR pathway. Therefore, Wnt inhibition sensitizes cells to the PARP inhibitors.

3. The authors assess the reduction in HR upon treatment with Wnt inhibitors with the DR-GFP assay and state that the direct observation of DNA damage with quantification of foci was impossible. It is understood that the expression of several DNA repair genes normally present at foci is decreased but can a marker such as γ H2AX be used?

We agree that γ H2AX foci formation is the most commonly used method for assessing DNA damage. As the reviewer rightly noted, we are unable to use this marker as the expression of γ H2AX was also reduced upon Wnt inhibition. In response to the reviewer's concern we have now assayed the formation of 53BP1 nuclear foci to assess DNA damage. We find that olaparib treatment induces foci formation that is significantly enhanced in cells treated with the combination of ETC-159 and olaparib. This supports the conclusion that Wnt inhibition impairs DNA repair by downregulating the expression of HR and FA pathway genes inducing a BRCA like state, hence promoting DNA damage (new Figures 6H and I).

4. In figure 1C, presumably the decrease in tumor volume in ETC-159 and Olaparib is statistically significant? This should be indicated. Also, is the combination treatment statistically different from the individual treatments?

As per the reviewer's suggestion, we have now included the p values to indicate the significance of the differences between various treatment groups.

5. The findings that inhibiting tankyrase also synergized with Olaparib represent an important extension of the work given the more frequent mutations of distal Wnt pathway components. It would therefore be important to show that tankyrase treatment in APC mutant tumors also lead to decreased expression of DNA repair genes to validate the mechanism in this context.

We agree with the reviewer and we now show that in these APC mutant colorectal cells, COLO320HSR, Wnt/ β -catenin signaling regulates DNA repair gene expression as both knockdown of β -catenin using siRNA as well as treatment with the tankyrase inhibitor G007LK reduces the expression of DNA repair genes. (new Figure 4D).

****Minor Points:****

1. On line 147 the authors state that the genes were decreased by 8-10 folds. Figure 2D however does not show fold and is expressed as fraction of expression compared to no treatment.

As suggested by the reviewer we have modified the text to improve clarity.

2. In figure 1D, is there an explanation for the absence of difference in tumor weight in the ETC-159 condition? Is this due to differentiation of tumors by ETC-159 leading to mucus loading and hence heavier tumors?

In Figure 1D the mice were intentionally treated with a low dose of ETC-159 (10 mg/kg) so the tumor weight reduction was not as robust as when we use higher (30-75 mg/kg/day) doses.

3. Why the authors performed the DR-GFP assay in *Aspc-1* cells and not in HPAF-II in which most of the work was performed?

We agree that performing the DR-GFP assay in HPAF-II cells would have been ideal. DR-GFP assay requires transient transfection of the I-Sce I expressing plasmid to assess the HR rate. We were limited by the ability to transfect HPAF-II cells using transient transfection methods as the efficiency is less than 1%.

4. The legend for Figure 3 needs to be adjusted to correspond to the panels.

Thank you for pointing this out. We have adjusted the figure legend according to the panels.

Reviewer #2

-My main issue with this manuscript is the inconsistent use of different cell lines. At various points HPAF-II, EGI-1, MCAS, CFPAC-1 and COLO320-HSR are used, without any explanation of why specific lines were used for different experiments. It has a feeling of cherry-picking, and to be truly convinced I would like to see consistency in this aspect. For example, in Fig 4D the COLO320-HSR line is used to show that inhibition of Wnt with G007-LK phenocopies inhibition with ETC-159. However, this is the first time the cell line appears in the manuscript and the effect of ETC-159 on this line is not shown, making it difficult to compare with other experiments.

We apologize if the rationale for selecting various cell lines was not made clear in the manuscript. We used RNF43 mutant HPAF-II cells for most of our initial discovery studies (in vivo and in vitro analysis) as these cells are Wnt-addicted due to loss-of-function RNF43 mutations. We used additional Wnt-addicted cell lines from different tissue of origin and with distinct driver mutations to test generality, i.e., to determine if the reduction of DNA repair gene expression upon Wnt inhibition and the combination of Wnt inhibitor with olaparib is a common phenomenon. For this purpose, we used a cholangiocarcinoma cell line EGI-1 and colorectal patient-derived xenografts (Wnt-addicted due to an R-spondin translocation), the ovarian and pancreatic cancer cell lines MCAS and AsPC-1 respectively (Wnt-addicted due to an RNF43 mutation), and the pancreatic cancer cell line CFPAC-1 (sensitive to Wnt inhibition, mechanism unknown). We used AsPC-1 cells for the DR-GFP assay because these cells are readily transfectable.

To test if other Wnt pathway inhibitor drugs had a similar effect as ETC-159, we used COLO320-HSR cells that have an APC loss-of-function mutation. Hence we used a tankyrase inhibitor, G007LK that prevents Wnt signaling by stabilizing AXIN. This showed that the synergy of Wnt pathway inhibitors with olaparib is a general phenomenon.

-The authors state that the staining for BRCA1, FANCD2 and RAD51 is higher in the stem cells compared to the more proliferative cells higher in the crypt. The images supplied don't really show this however. RAD51 in particular looks to be expressed at a higher level in the upper crypt rather than the lower crypt I would need to see better evidence to agree with the authors' claim.

We apologize for our misstatement regarding the localization of BRCA1, FANCD2, and RAD51 observed in IHC staining and we thank the reviewer for pointing this out. We have modified the text to state: "In agreement with the presence of a Wnt gradient in the intestine, we observed positive nuclear staining for BRCA1, FANCD2 and RAD51 throughout the crypts. Conversely, these proteins were absent in Wnt-low villus epithelial cells (Figures 7A and EV4)"

- Lgr5 positive cells have been extensively transcriptomically characterized, so if they are transcriptionally regulated, as the previous data would suggest, then this will be available in the literature. These datasets will also allow the analysis of Mybl2 in the crypt base.

We thank the reviewer for this helpful suggestion. We have now looked for published RNAseq datasets of intestinal stem cells. Two recent studies were very informative as noted in our response to reviewer 1 as well. A study from the Abud lab (GSE149311) (Jardé *et al*, 2020) online July 2020 performed RNAseq on sorted LGR5 high, mid, low and negative populations from intestinal epithelial cell pools. Looking into their dataset, we find that expression of both *Mybl2* and DNA repair genes increases as LGR5 expression increases, consistent with our model. We have now included this data in our new Figure 7B.

Additionally, we have now examined single cell RNA seq data from the human intestine, also very recently published (Wang *et al*, 2020). There, we also see co-expression of HR and FA genes with *MYBL2* in both stem and transit-amplifying cell clusters (new Figure EV5A). Taken together, this additional data further support the model that the Wnt/ β -catenin working in part through *MYBL2* regulates the expression of HR and FA genes in the human and mouse intestine.

- Furthermore, it is now thought that the *Lgr5* cells at the base of the crypt are a homogenous population, the majority of which are radiosensitive rather than radioresistant (DOI: 10.1038/s41598-020-64987-1). This needs to be acknowledged in the manuscript and the conclusions adjusted accordingly.

We thank the reviewer for pointing out this study, we have now included it in the manuscript and adjusted the conclusions accordingly.

-The data regarding the induction of senescence also needs improvement. Xenograft experiments in Fig 1 are used to show an increase in senescence, which is nicely demonstrated using various markers. The authors are missing a measure of proliferation within these tumours (*Ki67*, *pHH3* or *BrdU* if they have it) in order to show a functional loss of growth potential. The authors further suggest that an accumulation of double strand breaks accelerates the senescence but provide no evidence to show this. Additionally, as above, it would be good to see similar effects in additional cell lines in order to gauge the universality of the finding.

As per the reviewers' suggestion, we have now performed the analysis of multiple cell cycle genes. The expression of the cell cycle genes *MKI67*, *CDKN1*, *CCNE2*, and *AURKB* is reduced upon treatment with ETC-159 and further reduced in tumors treated with a combination of olaparib and ETC-159 (Figure below).

As suggested by the reviewer, we now provide evidence for the accumulation of double-strand DNA breaks using the 53BP1 foci formation and show that olaparib treatment induces 53BP1 nuclear foci formation that is significantly enhanced in cells treated with the combination of ETC-159 and olaparib. This further supports our conclusion that Wnt inhibition impairs DNA repair by downregulating the expression of HR and FA pathway genes inducing a BRCA like state, hence promoting DNA damage (new Figures 6H and I).

- Indeed, senescence is an extremely rare event in the intestine, so it is likely that an alternative mechanism is at play in the data presented in Fig 7. This should be acknowledged (or shown otherwise) and the conclusion drawn should reflect this.

Our data in figure 6 suggests senescence is occurring in the tumors. We agree with the reviewer that senescence does not occur in the intestine. In figure 7 we show that Wnt signaling regulates the expression of HR/FA genes in the intestine.

-The evidence that Mybl2 is binding to the promoters is based on previously published data. It would strengthen the manuscript to that ChipSeq data from the lines used here, potentially with/without Wnt-inhibition.

As suggested by the reviewer, we performed MYBL2 ChIP and confirmed the binding of MYBL2 at the promoters of DNA repair genes. Further, supporting our findings that Wnt/MYBL2 axis regulates the expression of DNA repair genes, we also show reduced MYBL2 binding in the ETC-159 treated cells (new Figure 5G).

-The authors are not consistent with their use of cell lines, using different ones at different points in the manuscript. I have addressed elsewhere in this review.

We have provided an explanation in response to the first question raised by the reviewer.

-The use of EGI-1, MCAS and CFPAC-1 cell lines is used to show that this is a general phenomenon. Considering that colon tissue is the focus of the latter part of the paper, it would make sense to include a colon cancer line here. Indeed, such a line is used (as mentioned elsewhere) but not included here.

As mentioned above we have used several cell lines HPAF-II, AsPC-1, EGI-1, MCAS and CFPAC-1 as well as pancreatic and colorectal patient-derived xenografts with loss-of-function mutations in RNF43 or R-spondin that are Wnt addicted to show changes in gene expression and/or synergistic growth inhibition with a combination of ETC-159 and olaparib treatment.

The colorectal cell line COLO320-HSR, being APC mutant, is sensitive to tankyrase inhibitor G007LK, but not PORCN inhibitors, which act upstream of the mutation. We have therefore tested the combination of PARP inhibitor and G007LK in COLO320-HSR cells (Figure 4). Additionally, the data from the Lowe lab (new figure EV5B) also shows the coordinate regulation of *Mybl2* and HR/FA genes in mouse organoids after APC knockdown.

-The authors show altered HR and FA pathways after Apc deletion in the intestine (and in Wnt-high normal physiology), but do not show that this leads to increased sensitivity to Parp inhibition. Treatment of animals would be needed to support the claim that this is functionally relevant.

We showed (Figure 7C) that APC mutation further increases HR/FA pathway activity in the intestine. This is consistent with the re-analysis of gene expression in organoids after APC knockdown (new Suppl Fig EV5B) as well, where knockdown of APC in the intestinal organoids enhances the expression of DNA repair genes. Therefore Wnt-high tumors (APC adenomas included) should be less, not more sensitive to DNA damage. Therefore, we do not think that additional mouse studies will be useful here.

****Minor comments:****

-There is variability in group size that is strange. In Fig 1D for example, ETC159 and Olaparib groups have 10 animals each, whereas the combination has 7. What is the reason for this?

Usually we start with a group of mice injected with cancer cells, and then assign them to treatment arms to get equal size tumors in each group. Sometimes tumors don't grow, sometimes mice kill each other, or we may make mistakes in dosing and need to censor a mouse. These are the major reasons for the variability in group size.

-In Figure 1, 3 cell lines are used to show the generality of this finding. However, the figure only shows the images for the EGI-1 cell line. The others need to be included (in a supplementary figure perhaps) to prove the finding.

As per the reviewer's suggestion we have included the images from additional cell lines (new supplemental Figures EV1A-B).

-Along the same lines, the immunoblot shown in Fig 2E should be repeated on additional cell lines.

As suggested by the reviewer we have now included the immunoblot for DNA repair genes in colorectal cancer PDX to substantiate the gene expression analysis data (new Figure 2G).

-Fig 4D shows images of a colony formation assay. In the text, these images are quantified, but this quantification is not included in the figure. That should be added.

As per the reviewer's suggestion we have now included the graph showing the combination index values for the COLO320 HSR and Olaparib combination (new Figure 4F).

- Are prior studies referenced appropriately?

-A previous study from the Rudolph lab (DOI: 10.15252/emj.201490700), which was focused on DNA damage and Wnt signalling in the intestinal stem cell, is not in agreement with the results in this manuscript. That paper detailed a well carried out study and is the strongest data we have at present in this regard. The authors really should reference this paper, and comment on the disagreement between their findings.

As requested we have referenced this paper and commented on the disagreement in the discussion.

-A study by Dorsam et al. (DOI: 10.1073/pnas.1712345115) addresses the expression of Parp1 in intestinal cancer. This study needs to be cited and commented on.

We have now included this interesting finding in the discussion.

-Describe the nature and significance of the advance (e.g. conceptual, technical, clinical) for the field.

-The idea that Wnt-inhibition produces a synthetic lethal state with Parp inhibitors is an interesting one. However, the authors stretch the universality of this finding, and would need a lot more evidence to confirm that. There are several studies in the intestine in particular (some of which are referenced above) that indicate a far more complex picture than that presented here. While it is always tempting to think about the clinical significance of findings such as those presented here, I am not convinced that these will translate to the clinic.

We appreciate the reviewer's comment that the combination of Wnt-inhibition with Parp inhibitors is interesting. Wnt signaling has long been associated with drug resistance, but the mechanism we describe here is novel and has both basic and potentially translational impact. Of course additional studies will be needed to test if this will translate to the clinic. Clinical testing is feasible, as olaparib is an FDA approved drug for treatment of BRCA deficient tumors and PORCN inhibitors have advanced to clinical trials, ClinicalTrials.gov-NCT01351103, NCT02278133, NCT02521844, NCT03447470, NCT02675946, and NCT03507998.

Reviewer #3

****Major comments:****

(1) In relation to Fig 1, phenotypic analysis with chemical inhibitors is always concerned with off-target effects. It is better to confirm a synergistic antitumor effect with a combination of at least two independent inhibitors. Synergistic antitumor effects should be confirmed using other compounds.

We agree with the reviewer that independent inhibitors need to be tested to rule out off target effects. We would like to draw the attention of the reviewer to Figure 4 where we have used a different Wnt pathway inhibitor, G007LK, in combination with Olaparib. G007LK has a different mechanism of action than ETC-159. It stabilizes AXIN, hence increasing β -catenin degradation. G007LK and ETC-159 therefore function to inhibit the Wnt pathway by independent mechanisms and we show that both drugs synergize with Olaparib to prevent growth of Wnt high cancers.

(2) In relation to Fig 2E, Immunoblotting of BRCA1, BRCA2, FANCA and FANCD2 in HPAF-II tumors collected 8 and 56 hours after the start of ETC-159 treatment confirmed the down-regulation of these proteins. However, the reduced expression at 8 hours after treatment is unacceptable. The results should be accurately described. In addition, the BRCA1 blot is dirty and unappealing. They should be replaced.

We apologize for any confusion in our description of the immunoblot. We have modified the text to improve clarity as suggested by the reviewer. We are sorry the reviewer finds our blot unappealing. As suggested, we have repeated the BRCA1 blot (new figure EV1D).

(3) It should be determined whether the reduction of HR and FA pathway genes by PORCN inhibitors is dependent on the reduction of Wnt/ β -catenin signaling rather than an indirect effect such as cell proliferation. For example, it is possible to test whether the decrease in gene expression by ETC-159 can be reversed by treatment with a GSK3 inhibitor for a short period.

Our data show clearly that the changes in HR/FA genes are dependent on changes in β -catenin signaling (Figure 4A-B). How much of this change in gene expression is contributed to by changes in cell proliferation is an important question. To address this concern, we would like to draw the reviewers' attention to Suppl Figures EV1E and F. We show that ETC-159 treatment of HPAF-II cells in vitro for 48 h did not significantly reduce the percentage of cells in S-phase, at a time point where there already was a significant reduction in the expression of HR and FA pathway genes.

Additionally, in the DR-GFP assays in AsPC-1 cells (Figure 2J), treatment with ETC-159 for 24 h led to a ~40% reduction in HR-activity upon Wnt inhibition. However, in newly added data (new Figure EV1H), we show that the changes in the S phase cells in ETC-159 treated AsPC-1 cells after 48 h treatment were not significant. Taken together these data show that the reduction of HR and FA pathway genes by PORCN inhibitors is not due to an effect on cell proliferation.

We agree that testing if the decrease in HR/FA gene expression by ETC-159 can be reversed by treatment with a GSK3 inhibitor is informative. We performed this experiment slightly differently as is shown in Figure 4B. We generated xenografts from HPAF-II cells that express stabilized β -catenin and are hence insensitive to CK1 α /GSK3 phosphorylation and subsequent proteasomal degradation. While treatment of HPAF-II xenografts with ETC-159 reduces the expression of HR/FA genes, this is rescued in xenografts from HPAF-II cells with stabilized β -catenin.

(4) *The authors should show direct experimental evidence be provided whether the potentiation of synergistic antitumor effects in combination with PARP inhibitors is dependent on the impairment of the double-stranded DNA repair pathway by PORCN inhibitors.*

As suggested by the reviewer, we now provide evidence for the accumulation of double-strand DNA breaks using the 53BP1 foci formation and show that olaparib treatment induces 53BP1 nuclear foci formation that is significantly enhanced in cells treated with the combination of ETC-159 and olaparib. This supports our conclusion that Wnt inhibition impairs DNA repair by downregulating the expression of HR and FA pathway genes inducing a BRCA like state, hence promoting DNA damage (new Figures 6H and I).

(5) *The results for the synergistic antitumor effect in Fig 3 are impressive. It would be more informative that the authors show one more additional case of Wnt high cells resistant to PORCN inhibition other than PaTu8988T cells. Further, as shown in Fig 1C and D, data from xenograft experiments would be more convincing.*

Per the reviewer's suggestion we have tested an additional Wnt high cell line PL45 that has high abundance of Frizzleds and is resistant to ETC-159 (revised Figure 3A). We now show that similar to *PaTu8988T cells* co-treatment with olaparib sensitizes PL45 cells to ETC-159 treatment (new Figure 3F).

Additionally, as suggested by the reviewer we performed an additional mouse xenograft experiment and now show that similar to the *in vitro* data, the combination of ETC-159 and olaparib is more effective in preventing the growth of PaTu8988T xenografts compared to either drug alone (new Figure 3E).

(6) *For Fig 4B, the authors need to explain the reasons why there is no increased expression of HR/FA genes due to stabilized β -catenin expression compared to control cells. As a control experiment, the authors should show the expression of AXIN2 by stabilized β -catenin expression.*

As per the reviewer's suggestion we have included the data showing the expression of AXIN2 in cells with stabilized β -catenin (modified Figure 4B). We agree with the reviewer that it is interesting that the basal levels of the DNA repair genes do not change in cells with stabilized β -catenin.

To address the reviewer's question on why expression of stabilized β -catenin did not increase the expression of HR/FA genes, we went back to our comprehensive analysis of gene expression in response to Wnt inhibition (Figure EV1C) (Madan *et al*, 2018). We find that there is a class of Wnt/ β -catenin target genes whose expression is robustly upregulated by stabilized β -catenin. These include AXIN2, NKD1 and RNF43. However, there is a second class of Wnt/ β -catenin target genes including MYC, CCND1 and MYBL2 whose expression is NOT increased by stabilized β -catenin. We speculate that expression of this latter class of genes is dependent on multiple upstream factors and that an increase of β -catenin alone is not sufficient to alter their expression.

(7) *Related to Fig 4D, in addition to PORCN inhibitors in Wnt ligand-dependent cancer cells, it is interesting to note the synergistic antitumor effect of PARP inhibitors on G007-LK in the APC mutant colorectal cancer cell line COLO320-HSR cells. The expression of the AXIN2 and HR/FA pathway genes should be confirmed in COLO 320 HSR cells treated with G007-LK.*

Based on the reviewer's suggestions we analysed the gene expression changes in COLO320HSR cells treated with G007-LK. We find that, similar to Wnt inhibition with ETC-159, treatment with G007-LK also reduces expression of DNA repair genes in these

cells. Additionally, we show that knockdown of β -catenin in these cells reduced the expression of DNA repair genes, consistent with the regulation of these genes by Wnt/ β -catenin in part via MYBL2 signaling in these APC mutant cells (new Figures 4D and 5K).

(8) In the APC mutant colon cancer cell line COLO 320 HSR cells, it would be more persuasive if the combined effects of G007-LK and PARP inhibitor could be confirmed in a xenograft experiment.

We have already shown synergy with the PORCN inhibitor *in vitro* and *in vivo* in multiple models, and in addition used the tankyrase inhibitor as a confirmation that it is not a PORCN inhibitor-specific effect. We think an additional xenograft synergy model is time and mouse intensive and we concur with the editor that this will not substantially add to the core conclusion that Wnt signaling regulates BRCA and FA genes.

(9) The Wnt- β -catenin-MYBL2 pathway is an important point in the analysis of the mechanism in the current paper, but the data are somewhat superficial. The following experiments are absolutely required. Determine whether MYBL2 is a direct target gene for the β -catenin/TCF pathway. Does β -catenin knockdown reduce MYBL2 expression in HPAF II cells?

As per the reviewers' suggestion we have tested and now show that *MYBL2* is a direct target of β -catenin and its levels are reduced in cells where β -catenin is knocked down by siRNA (new Figure 5E). We have also added new data from published datasets showing that a) APC knockdown in organoids, activating β -catenin, increases *Mybl2* expression (new figure EV5B) and b) *Mybl2* expression positively correlates with Wnt/ β -catenin target genes *Axin2* and *Lgr5* in the mouse intestine (new Figures 7B and EV5A-B). Taken together, these data support the existence of a Wnt/ β -catenin/*Mybl2* pathway.

Is there a consensus binding sequence for TCF4 in the regulatory region of the MYBL2 gene? The binding of β -catenin/TCF4 to the regulatory element of the MYBL2 gene should be confirmed by ChIP analysis in HPAF II cells.

As noted above and in response to reviewer 1, our data demonstrate that *MYBL2* expression is regulated by Wnt/ β -catenin signaling. However, as the reviewer notes, we did not establish which sequence elements in the genome are responsible for this. It's somewhat problematic, as β -catenin regulation of genes is very often through distant enhancers rather than through traditional LEF/TCF binding sites in the proximal promoters, c.f., (Nakamura *et al*, 2016; Guenther *et al*, 2014). For example, Guenther *et al.*, showed the important *TCF7L2* binding site determining brown versus blond hair is 350 kb distant from the transcriptional start site of the relevant gene. Similarly, when we examined the *TCF7L2* ChIPseq data in HCT116 and Panc1 cells (ENCODE Project Consortium, 2012), we found *TCF7L2* binding ~10 Kb upstream of the transcription start site of the *MYBL2* gene where it overlapped with the traces for H3K27 acetylation (see figure below) and laid on top of a *TCF7L2* motif (indicated with an *) that we found using FIMO motif search tool (Chr 20 + 43657377-7390). In addition, β -catenin ChIP is significantly more challenging than standard transcription factor ChIP because β -catenin is not a direct DNA binding protein (Nakamura *et al*, 2016). Therefore, while we did do ChIP with anti-*MYBL2* (see below), we did not perform ChIP with TCF or β -catenin.

(10) Is reduced expression of MYBL2 also observed with G007-LK treatment in COLO320 HSR cells?

As suggested by the reviewer we now show that the expression of MYBL2 in COLO320HSR cells is reduced by G007LK treatment, supporting the conclusion that MYBL2 is a Wnt/ β -catenin regulated gene in the APC mutant colorectal cells (new Figure 5K)

(11) The authors confirmed the binding of MYBL2 to the promoters of HR/FA pathway genes using a public ChIP sequence data set. However, these data are from human chronic myeloid leukemia cells (K562) cells. The authors should confirm the binding of MYBL2 to the promoter region of the HR/FA pathway genes using Wnt-dependent cells used in this paper.

As suggested by the reviewer, we performed MYBL2 CHIP and confirmed the binding of MYBL2 at the promoters of DNA repair genes in our Wnt-high cell line. Further supporting our findings that the Wnt axis regulates the expression of DNA repair genes in part via MYBL2, we also show reduced MYBL2 binding in the ETC-159 treated cells (new Figures 5G and EV2E).

(12) The public ChIP sequence data should also specify the location of the transcription start site (TTS), the location of the peak from TTS, and the presence or absence of the consensus motif for MYBL2.

Based on the reviewers' suggestion, we have modified the figure to add the location of the MYBL2 peaks with respect to the TSS and putative motif locations. Figure EV2D

(13) The importance of MYBL2 as a downstream mediator of the Wnt/ β -catenin pathway is not clear. Is suppression of PORCN inhibitor-dependent HR/FA pathway genes expression rescued by MYBL2 expression? In addition, can MYBL2 expression rescue the synergistic antitumor effects by a combination of PORCN and PARP inhibitors? These are important data that can also be used as responses to review comments (4).

We appreciate the great interest the reviewer has in further characterization of MYBL2. It is important to point out that suppression of Wnt/ β -catenin signaling leads to changes in many thousands of genes including many transcription factors (Madan *et al*, 2018). We found that MYBL2 is one of the TFs that regulates HR/FA genes, but we do not claim that it is the only TF regulating HR/FA genes. In Figure 5F, MYBL2 knockdown was ~50% as effective as ETC-159 treatment. Other β -catenin regulated transcription factors likely make up the other 50% effect.

Rescue experiments are hard, but as suggested by the reviewer we generated stable cells overexpressing MYBL2 (6 fold) under the control of CMV promoter. As shown below, overexpression of MYBL2 partially (~30%) rescued the repression of DNA repair genes caused by ETC-159 inhibition of Wnt signaling. We assume the rescue is only partial because many other Wnt/ β -catenin factors are also inhibited by ETC-159.

As shown above, MYBL2 is necessary but not sufficient to drive the expression of these DNA repair genes in the Wnt high cells, hence the rescue of cell growth by MYBL2 alone is highly unlikely hence we did not perform this experiment.

(14) Knockdown of MYBL2 and the combination of knockdown of MYBL2 with PARP inhibitors should be tested for their ability to mimic the phenotype of PORCN inhibitors.

As noted above, MYBL2 is part of the story but not the complete story for regulation of HR/FA genes. In addition, haploinsufficiency of *Mybl2* in mice has been shown by others to impair dsDNA break repair (Bayley *et al*, 2018).

(15) Referring to Fig 6B, there is no description concerning the results of the 21-day treatment.

We thank the reviewer for pointing this out. We have now included the information in the Figure legend that the tumors used for analysis of cleaved caspase and PARP (Figure 6B) are from the study shown in Figure 1C-D.

(16) It seems that there is no consistent trend for the expression of p21 in Fig 6D. The results should be accurately described or replaced with consistent data.

We agree with the reviewer and we have now repeated the western blot. Please see the revised Figure 6D.

(17) It would be nice to show that cellular senescence occurs by combined MYBL2 KD and PARP inhibitor treatment. Can MYBL2 expression also rescue cell senescence induced by PORCN and PARP inhibitors treatment? These experiments strengthen the authors' conclusion that the regulation of HR/FA pathway by the Wnt/ β -catenin/MYBL2 axis is involved in cell senescence due to the combination therapy with PORCN and PARP inhibitors.

Again, we appreciate the great interest the reviewer has in further characterization of MYBL2. In several models MYBL2 has been shown to inhibit senescence (Masselink *et al*, 2001; Martinez *et al*, 2011; Mowla *et al*, 2014). It is plausible that the senescence induction upon ETC-159 treatment is also mediated at least in part via MYBL2 inhibition.

Our study does not claim that targeting MYBL2 in combination with PARP inhibitors is a treatment option. Rather, we are showing that the combination of Wnt and PARP inhibitors regulates the expression of DNA repair genes. Our data indicate that MYBL2 is one of the downstream regulators of the Wnt/ β -catenin pathway.

(18) *Enlarged pictures of the villi should be shown for comparison with crypts in Fig 7A.*

We thank the reviewer for the suggestion and as suggested we have included the enlarged images of the villi and additional images to show the difference in expression of DNA repair genes in crypts vs. villi (new Figure EV4).

(19) *The authors should show the expression of MYBL2 in normal intestine and intestinal tumor legions of APC min mouse immunohistochemically.*

As per our response to the reviewers' comment #9, above we have tested and now show that *MYBL2* is a direct target of β -catenin and its levels are reduced in cells where β -catenin is knocked down using a siRNA (new Figure 5E). As noted above, we have also added new data from published datasets showing a) APC knockdown in intestinal organoids increases *Mybl2* (new Figure EV5B). and b) *Mybl2* expression positively correlates with *Lgr5* expression (new Figures 7B and EV5A). Taken together, these data support the existence of a Wnt/ β -catenin/*Mybl2* pathway. It's not clear to us that MYBL2 IHC would add much except a nice visual to the existing data.

REFERENCES:

- Bayley R, Blakemore D, Cancian L, Dumon S, Volpe G, Ward C, Almaghrabi R, Gujar J, Reeve N, Raghavan M, Higgs MR, Stewart GS, Petermann E & García P (2018) MYBL2 Supports DNA Double Strand Break Repair in Hematopoietic Stem Cells. *Cancer Res.* **78**: 5767–5779
- Dow LE, O'Rourke KP, Simon J, Tschaharganeh DF, van Es JH, Clevers H & Lowe SW (2015) Apc Restoration Promotes Cellular Differentiation and Reestablishes Crypt Homeostasis in Colorectal Cancer. *Cell* **161**: 1539–1552
- ENCODE Project Consortium (2012) An integrated encyclopedia of DNA elements in the human genome. *Nature* **489**: 57–74
- Guenther CA, Tasic B, Luo L, Bedell MA & Kingsley DM (2014) A molecular basis for classic blond hair color in Europeans. *Nat Genet* **46**: 748–752 Available at: <http://www.nature.com/articles/ng.2991>
- Jardé T, Chan WH, Rossello FJ, Kaur Kahlon T, Theocharous M, Kurian Arackal T, Flores T, Giraud M, Richards E, Chan E, Kerr G, Engel RM, Prasko M, Donoghue JF, Abe S-I, Pheffe TJ, Nefzger CM, McMurrick PJ, Powell DR, Daly RJ, et al (2020) Mesenchymal Niche-Derived Neuregulin-1 Drives Intestinal Stem Cell Proliferation and Regeneration of Damaged Epithelium. *Cell Stem Cell* **27**: 646–662.e7
- Madan B, Harmston N, Nallan G, Montoya A, Faull P, Petretto E & Virshup DM (2018) Temporal dynamics of Wnt-dependent transcriptome reveal an oncogenic Wnt/MYC/ribosome axis. *J. Clin. Invest.* **128**: 5620–5633
- Martinez I, Cazalla D, Almstead LL, Steitz JA & DiMaio D (2011) miR-29 and miR-30 regulate B-Myb expression during cellular senescence. *Proc. Natl. Acad. Sci. U.S.A.* **108**: 522–527
- Masselink H, Vastenhouw N & Bernards R (2001) B-myb rescues ras-induced premature senescence, which requires its transactivation domain. *Cancer Lett.* **171**: 87–101
- Mowla SN, Lam EW-F & Jat PS (2014) Cellular senescence and aging: the role of B-MYB. *Aging Cell* **13**: 773–779
- Nakamura Y, de Paiva Alves E, Veenstra GJC & Hoppler S (2016) Tissue- and stage-specific Wnt target gene expression is controlled subsequent to β -catenin recruitment to cis-regulatory modules. *Development* **143**: 1914–1925 Available at: <http://dev.biologists.org/lookup/doi/10.1242/dev.131664>
- Wang Y, Song W, Wang J, Wang T, Xiong X, Qi Z, Fu W, Yang X & Chen Y-G (2020) Single-cell transcriptome analysis reveals differential nutrient absorption functions in human intestine. *J Exp Med* **217**: 357

22nd Dec 2020

Dear Dr. Madan,

Thank you for the submission of your revised manuscript to EMBO Molecular Medicine. We have now received the enclosed report from the three referees who had reviewed your original manuscript. As you will see, while referees #1 and #2 are satisfied with the revisions and are supporting publication of the manuscript, referee #3 still raises some concerns that should be addressed in a minor revision of the present manuscript:

Referee #3 comments:

- #1: Please test the combined effects of another PARP inhibitor and a PORCN inhibitor in vitro.
- #2: The blot you have included in the EV1D is enough and you do not need to repeat the experiment.
- #3: Please address this comment in writing. If you do have data at hand, we will be happy for you to include it, however it will not be required.
- #9: Please perform a ChIP experiment as required by the referee.
- #14: Please perform the experiment required by the referee with MYBL2 knockdown combined with PARP inhibitor.
- #17: Please address this comment in writing. If you do have data at hand, we will be happy for you to include it, however it will not be required.
- #19: Please address this comment in writing. If you do have data at hand, we will be happy for you to include it, however it will not be required.

Furthermore, please address the following editorial issues:

1) Main manuscript text:

- Please answer/correct the changes suggested by our data editors in the main manuscript file (in track changes mode). This file will be sent to you in the next few days. Please use this file for any further modification.
- Please remove the red text.
- We can accommodate a maximum of 5 keywords, please adjust accordingly.
- References: please adjust format so as to have 10 authors listed before et al.
- Authors contribution: please use initials instead of full names.
- Please replace "Declaration of Interests" by "Conflict of Interest"
- "Data not shown": remove "data not shown" (p.38) As per our guidelines, on "Unpublished Data" the journal does not permit citation of "Data not shown". All data referred to in the paper should be displayed in the main or Expanded View figures. "Unpublished observations" may be referred to in exceptional cases, where these are data peripheral to the major message of the paper and are intended to form part of a future or separate study, the names of the persons that reported the observation should be listed in brackets.
- Material and methods:
Mice: please indicate the strain, gender, and age of the mice, as well as the frequency of treatment.
- Thank you for including exact p values. Please also indicate the exact p values for non-significant p value (i.e. Fig. 3E), along with the statistical test used.
- Please include a Data Availability Section:
Before submitting your revision, primary datasets produced in this study such as RNAseq data need

to be deposited in an appropriate public database (see <https://www.embopress.org/page/journal/17574684/authorguide#dataavailability>). The accession numbers and database should be listed in a formal "Data Availability" section (placed after Materials & Method). Please note that the Data Availability Section is restricted to new primary data that are part of this study. If not applicable, please indicate: "This study includes no data deposited in external repositories"

2) Figures:

- Figure 7C: please indicate in the upper picture the box of origin for the lower pictures.

3) Checklist:

Please fill in the section F/18: Data deposition in a public repository is mandatory for protein, DNA and RNA sequences, Macromolecular structures, Crystallographic data for small molecules, Functional genomics data, Proteomics and molecular interactions.

4) We would also encourage you to include the source data for figure panels that show essential data. Numerical data should be provided as individual .xls or .csv files (including a tab describing the data). For blots or microscopy, uncropped images should be submitted (using a zip archive if multiple images need to be supplied for one panel). Additional information on source data and instruction on how to label the files are available at .

5) Thank you for providing a synopsis text. Please remove it from the manuscript and upload it as a separate document. I added minor modifications to fit our style and format, please let me know if you agree with the following:

"This study identifies that Wnt/ β -catenin signaling regulates homologous recombination and Fanconi anaemia DNA repair pathways in Wnt-high cancers and intestinal stem cells. Wnt signaling inhibition induces a BRCA-like state, and Wnt and PARP inhibitors synergize to inhibit Wnt-addicted cancers.

- Wnt signaling regulated the expression of a broad set of genes involved in repairing DNA double-strand breaks. This was mediated in part via the transcription factor MYBL2.
- Wnt inhibition created a BRCA-like state by inhibiting the expression of genes in the homologous recombination and Fanconi anemia DNA repair pathway.
- Wnt inhibition synergized with PARP inhibitors to more effectively treat multiple Wnt-addicted cancers.
- Wnt inhibition caused homologous recombination deficiency that, when combined with blockage of PARP-mediated ssDNA repair, led to an accumulation of dsDNA breaks to enhance senescence.
- Wnt signaling also regulated Homologous recombination and Fanconi Anemia pathway genes in intestinal stem and transit amplifying cells."

6) As part of the EMBO Publications transparent editorial process initiative (see our Editorial at <http://embomolmed.embopress.org/content/2/9/329>), EMBO Molecular Medicine will publish online a Review Process File (RPF) to accompany accepted manuscripts.

In the event of acceptance, this file will be published in conjunction with your paper and will include the anonymous referee reports, your point-by-point response and all pertinent correspondence relating to the manuscript. Let us know whether you agree with the publication of the RPF and as here, IF YOU WANT TO REMOVE OR NOT ANY FIGURES from it prior to publication.

I look forward to receiving your revised manuscript.

Yours sincerely,

Lise Roth

Lise Roth, PhD
Editor
EMBO Molecular Medicine

To submit your revised manuscript, please follow this link:

Link Not Available

***** Reviewer's comments *****

Referee #1 (Remarks for Author):

The authors have taken my previous reviews seriously and have done a thorough job addressing the comments from all reviewers. The manuscript is strengthened and represents an innovative body of work, which could lead to potential new therapeutic strategies.

Referee #2 (Remarks for Author):

The authors have done a good job in replying to my previous comments, and I appreciate the additional work that has been carried out. In particular, the ChIP experiments and the analysis of published crypt base data has significantly improved the work. I am satisfied with the adjustments they have made, and compliment the authors on their efforts. I am happy to recommend this manuscript for publication.

Referee #3 (Comments on Novelty/Model System for Author):

Since the authors did not address some of my critiques directly, I do not believe that the present results support the authors' conclusion. Whether further revision is acceptable or not is due to the Editor's decision.

The authors did not address some of my critiques. They proposed the model that the Wnt/beta-catenin/MYBL2 axis regulates expression of the FA/HR genes, resulting in cell proliferation of Wnt-high cancers. This reviewer requested direct evidence.

The following numbers indicate the original comments.

#1: To address to the reviewer's comment, the authors tested G007LK in addition to ETC-159, but they did not test another PARP inhibitor. The combined effects of another PARP inhibitor and a PORNC inhibitor should be tested.

#2: This comment means that the quality of the blot in Fig 2E (BRCA1) should be improved. The bands in lanes 3 and 4 are messy.

#3: The important conclusion of this study is that expression of the FA/HR genes is dependent on Wnt/beta-catenin signaling. Therefore, the authors need to prove that the short time treatment (within 10~20 hours) with Wnt activators, such as CHIR, restores the transcription of the FA/HR genes, even though MYBL2 mediated this reaction. Secondary effects cannot be ruled out in xenograft experiments.

#9: The ChIP analysis using HPAFII cells is essential for this study to show the presence of the presence of the Wnt/beta-catenin/MYBL2 axis. ChIP-qPCR analysis of TCF4 is a very common experiment and seems to be easy for the authors.

#14: The experiments which the reviewer requested are not difficult, thus the authors should perform them. The haploinsufficiency mouse data of MYBL2 is not related with authors' conclusion. If suppression of MYBL2 expression cannot reproduce the phenotypes induced by the PORCN inhibitor, even partially, then the authors' claim may be an exaggeration.

#17: The authors did not address to the reviewer's comment. This reviewer requested the results showing the association of cellular senescence and with the Wnt/beta-catenin/MYBL2 axis.

#19: The spatial information of MYBL2 expression in the normal intestine reinforces the authors' model, because the results would prove the association of the MYBL2 expression region with the Wnt signaling activity and the HR/FA expression region.

Rev_Com_number: RC-2020-00342

New_manu_number: EMM-2020-13349-V2

Corr_author: Madan

Title: WNT inhibition creates a BRCA-like state in Wnt-addicted cancer

12th Jan 21

Dear Dr. Roth

Thanks for considering our manuscript for publication in EMBO Molecular Medicine. We are pleased that referees #1 and #2 appreciate our extensive efforts and find the manuscript now suitable for publication. We would like to appeal the request by referee #3 for additional experimentation. Please find attached our response to the specific comments raised by the referee.

Best regards

Babita Madan

Dear Dr. Roth,

We are pleased that referees #1 and #2 appreciate our extensive efforts and find the manuscript now suitable for publication. We would like to appeal the request by referee #3 for additional experimentation. We believe that we have already fully addressed their comments, as described in more detail below.

Specific issues

1. Point #1: To address to the reviewer's comment, the authors tested G007LK in addition to ETC-159, but they did not test another PARP inhibitor. The combined effects of another PARP inhibitor and a PORCN inhibitor should be tested.

In the initial review, referee 3 noted “phenotypic analysis with chemical inhibitors is always concerned with off-target effects. It is better to confirm a synergistic antitumor effect with a combination of at least two independent inhibitors”. This new comment is a different question than the first review. This is also a very time-consuming request, since these assays must be done in several rounds in soft agar, taking several weeks each round. The senior authors have many years of experience in drug development and are keenly aware of the possibility of off-target effects. Therefore, in the initial manuscript we addressed this point robustly. We used two classes of Wnt pathway inhibitors as well as siRNA assays to show that the effect is not specific to any one small molecule. Additionally, we performed genetic assays to show that stabilization of β -catenin reverses the effect of ETC-159 (Figure 4). We also showed that in a cell line that is not Wnt high (Panc 08.13) (Figure 3) the expression of DNA repair genes is not regulated by Wnt inhibition and correspondingly there was no sensitivity to Wnt inhibitor and Olaparib combination. Taken together these data robustly address the issue of off-target effects, and confirm that we are studying a Wnt-regulated phenomenon. We feel this establishes the point very clearly, and that testing yet another drug will not add significantly to this conclusion.

2. Point #2: This comment means that the quality of the blot in Fig 2E (BRCA1) should be improved. The bands in lanes 3 and 4 are messy.

Perhaps the reviewer missed this new panel, but as suggested, we repeated the BRCA1 blot and included it in the revised manuscript as a new figure EV1D.

3. Point #3: The important conclusion of this study is that expression of the FA/HR genes is dependent on Wnt/beta-catenin signaling. Therefore, the authors need to prove that the short time treatment (within 10–20 hours) with Wnt activators, such as CHIR, restores the transcription of the FA/HR genes, even though MYBL2 mediated this reaction. Secondary effects cannot be ruled out in xenograft experiments.

As previously stated, we agree that testing if the decrease in HR/FA gene expression by ETC-159 can be reversed by treatment with a GSK3 inhibitor could be informative. We performed this experiment slightly differently. Instead of using a drug, we used a genetic approach. We generated xenografts from HPAF-II cells that express stabilized β -catenin and are hence insensitive to CK1 α /GSK3 phosphorylation and subsequent proteasomal degradation. While treatment of HPAF-II xenografts with ETC-159 reduces the expression of HR/FA genes, this is rescued in xenografts from HPAF-II cells with stabilized β -catenin (Figure 5D).

4. Point #9: *The ChIP analysis using HPAFII cells is essential for this study to show the presence of the presence of the Wnt/beta-catenin/MYBL2 axis. ChIP-qPCR analysis of TCF4 is a very common experiment and seems to be easy for the authors.*

In our response to the reviews, we provided a detailed explanation why TCF4 ChIP experiments do not provide the kind of information referee 3 seeks. This explanation satisfied referee 1, who had also requested a similar experiment. Without repeating our previous reply and results here, the key points are i) critical TCF4 sites are often in distant enhancers rather than in promoters, and ii) TCF4 binding to DNA is not a Wnt-regulated phenomenon, so the binding of TCF4 does not equate with regulation by Wnt/ β -catenin. We clearly showed by multiple approaches that expression of MYBL2 is regulated by Wnt/ β -catenin signaling. TCF4 ChIP would not add to this conclusion.

5. Point #14: *The experiments, which the reviewer requested are not difficult, thus the authors should perform them. The haploinsufficiency mouse data of MYBL2 is not related with authors' conclusion. If suppression of MYBL2 expression cannot reproduce the phenotypes induced by the PORCN inhibitor, even partially, then the authors' claim may be an exaggeration.*

We believe that the requested additional study of MYBL2 is beyond the scope of this manuscript. We already confirmed by multiple approaches the published results of others that MYBL2 is a regulator of DNA repair. As requested by the reviewer, we also performed an additional experiment to show that overexpression of MYBL2 partially rescues the effect of ETC-159. To perform a combination of PARP inhibitor on MYBL2 KO cells (which is not the major point of our paper) we would need to generate CRISPR KO cells for MYBL2 and then perform soft agar assays. These are not simple 3 day growth assays in 2D cell culture, and so it would take several months to perform this study. We believe that any new information would not significantly add to or alter our already well-supported conclusions.

6. Point #17: *The authors did not address to the reviewer's comment. This reviewer requested the results showing the association of cellular senescence and with the Wnt/beta-catenin/MYBL2 axis.*

In our initial response to the reviewer's comment that "*it would be nice to show...*", we addressed this concern of the reviewer. It is plausible that the senescence induction upon ETC-159 treatment is also mediated at least in part via MYBL2 inhibition. We have found that for Wnt-regulated tumors, senescence assays are much more informative in *in vivo* models. These are time-consuming experiments that we think are beyond the scope of this manuscript.

7. Point #19: *The spatial information of MYBL2 expression in the normal intestine reinforces the authors' model, because the results would prove the association of the MYBL2 expression region with the Wnt signaling activity and the HR/FA expression region.*

We addressed this point in our previous response by including in the revision the re-analysis of published scRNAseq data to specifically address if *Mybl2* is co-expressed with HR/FA genes in Wnt-active cells. We presented this data showing that *Mybl2* expression positively correlates both with the Wnt target gene *Lgr5* as well as with HR/FA gene expression in the new Figure EV5A. Taken together, these data support the existence of a Wnt/ β -catenin/*Mybl2* pathway in the same cells, addressing the reviewer's question.

Thank you for your consideration of our appeal.

Respectfully

David and Babita

19th Jan 21

Dear Dr. Madan,

Thank you for your email asking us to reconsider the request for additional experimentation, following the evaluation of your revised manuscript by referee #3. I communicated your letter to referee #1 and have now received his/her feedback. This referee agrees that your initial revisions satisfactorily addressed the points from the three reviewers, and thinks that while additional experimentation could potentially strengthen the findings, it would require a lot of work and would not substantially add to the conclusions of the manuscript.

Therefore, having discussed this matter one more time with my colleagues, we agree that only the editorial changes indicated in my decision letter should be addressed in this last round of revisions.

Looking forward to receiving your revised manuscript,

With my best wishes,

Lise

The authors performed the requested editorial changes.

YOU MUST COMPLETE ALL CELLS WITH A PINK BACKGROUND ↓
PLEASE NOTE THAT THIS CHECKLIST WILL BE PUBLISHED ALONGSIDE YOUR PAPER

Corresponding Author Name: Babita Madan
Journal Submitted to: EMBO Molecular Medicine
Manuscript Number: EMM-2020-13349